# Learning a Stable Reservoir from an Observed Trajectory via Persistent Loops and Markov Flow

## Abstract

We study whether embedding global topology and local transport into a fixed reservoir can improve phase tracking and prediction. From a single delay-embedded trajectory, we build a recurrent operator in two parts: (i) long-lived $H_1$ classes from persistent cohomology are converted to circular coordinates whose average phase velocities instantiate stable $2 \times 2$ rotation blocks, and (ii) short-horizon transition counts over a coarse partition define a Markov model whose action is lifted back to neuron space through sparse, stochastic pooling and lifting maps. A convex blend of these topological and flow components is scaled by power iteration to a preset operator-norm bound, yielding a leaky ESN with a straightforward echo-state guarantee; only a ridge-regularized linear readout is trained. The resulting reservoir is fixed, interpretable, and analyzable: its internal oscillators reflect the attractor's dominant loops, while its couplings align with observed local transport. In experiments on chaotic systems and real-world series, the method is data-efficient and maintains the computational profile of standard ESNs, while delivering improved phase tracking and competitive—often superior—multistep forecasts relative to tuned random reservoirs of the same size. Overall, the framework offers a principled alternative to sampling-based wiring by learning the reservoir once from data.

## 1 Introduction

Learning nonlinear dynamics from time series remains a central challenge in machine learning and scientific computing. Recurrent neural networks (RNNs) provide a flexible parametric family but are notoriously difficult to train and analyze. Reservoir computing addresses this by fixing the recurrent weights and training only a linear readout, as in Echo State Networks (ESNs) (Jaeger, 2001) and Liquid State Machines (LSMs) (Maass et al., 2002). In classical ESNs, the recurrent matrix is drawn at random and scaled to operate in a regime that heuristically balances memory and nonlinearity. While simple and effective, random reservoirs pose two enduring limitations: (i) *lack of structure and interpretability*—the internal dynamics bear no explicit relationship to the geometry or flow of the target system; and (ii) *fragile stability criteria*—widely used spectral-radius heuristics neither guarantee nor precisely characterize the Echo State Property (ESP) (Yildiz et al., 2012; Buehner & Young, 2006; Manjunath & Jaeger, 2013). This paper proposes a principled alternative where the reservoir itself is *learned once, offline* from a single trajectory, with explicit ties to the data's global topology and local flow, and with a clean, verifiable ESP certificate.

Our starting point is the observation that many natural and engineered dynamical systems evolve on low-dimensional attractors that admit meaningful topological summaries and coarse-grained transport structure. Through delay-coordinate reconstruction (Takens, 1981; Packard et al., 1980; Sauer et al., 1991), a single multivariate time series yields a point cloud on which persistent homology reveals long-lived one-dimensional homology classes (loops) that signal recurrent motion (Edelsbrunner & Harer, 2010). Persistent *cohomology* then supplies *circular coordinates* that parameterize these loops by angles, together with harmonic representatives that minimize discrete Dirichlet energy (de Silva & Vejdemo-Johansson, 2009). Efficient implementations such as RIPSER make $H_1$ computations practical on large samples (Bauer, 2021). At the same time, short-horizon transitions between coarse cells along the trajectory define a Markov chain that approximates transport of

the transfer (Perron–Frobenius) operator via an Ulam-type discretization (Dellnitz & Junge, 1999; Froyland, 2001; Klus et al., 2016). Such coarse models are well established in molecular and fluids applications and connect naturally to Koopman/operator-theoretic perspectives (Prinz et al., 2011; Williams et al., 2015; Klus et al., 2016). We leverage these insights to construct *Persistent Homology Reservoir (PHR)*: a leaky ESN whose recurrent matrix $W$ is a convex blend of two analyzable operators learned from the data: (i) a *topological rotation* component, $W_{\text{top}}$, that instantiates data-driven $2 \times 2$ rotation blocks with angular velocities estimated from persistent circular coordinates; and (ii) a *lifted Markov flow* component, $W_{\text{flow}} = BPA$, which pools reservoir states down to a coarse partition, advances them by the empirical Markov matrix $P$, and lifts them back by stochastic maps $A$ (row-stochastic) and $B$ (column-stochastic). This design explicitly *materializes* long-lived loops as stable internal oscillators while imprinting short-time flow directions into the reservoir in a contractive way.

*Our primary contributions can be summarized as follows:* **Topology- and flow-grounded reservoir design.** We introduce a fixed, analyzable ESN reservoir learned from a single trajectory by blending (a) persistent-cycles–induced rotation blocks aligned with the data's $H_1$ structure (de Silva & Vejdemo-Johansson, 2009; Edelsbrunner & Harer, 2010; Bauer, 2021), and (b) a lifted short-horizon Markov operator that encodes local flow directions in the spirit of Ulam discretizations of transfer-/Koopman operators (Dellnitz & Junge, 1999; Froyland, 2001; Klus et al., 2016; Williams et al., 2015); **Interpretability and modularity.** The rotation blocks serve as *internal oscillators* with interpretable physical/phase meaning, while the lifted Markov component offers a coarse-grained, operator-theoretic view of transport—both modules are plug-and-play and require no backpropagation through time; **Single-trajectory practicality.** The entire reservoir is learned offline from one embedded time series using scalable $H_1$ persistence and linear-time transition counting, after which standard ridge regression suffices for readout training (Ozturk et al., 2007; Jaeger, 2001).

Ergo, PHR replaces randomness by geometry and flow: it bakes the long-lived loops and local transport of the underlying attractor into the reservoir, provides clear stability guarantees, and yields interpretable internal modes. We view this as a step toward *structure-aware reservoirs* that inherit invariants from the data, aligning reservoir computing with contemporary operator-theoretic and topological data analysis. *Notes on usability across domains is provided in Appendix A.4.*

## 2 BACKGROUND AND RELATED WORKS

**Reservoir computing and the Echo State paradigm.** Reservoir Computing (RC) separates non-linear state evolution from linear readout training: a fixed high-dimensional recurrent system (*reservoir*) is driven by inputs, and only a linear map from states to outputs is learned. Two seminal instantiations are ESNs (Jaeger, 2001) and LSMs (Maass et al., 2002). Empirical effectiveness and design heuristics of RC are well-documented, including input scaling, spectral scaling of the recurrent matrix, sparsity, and leakage (Lukoševičius & Jaeger, 2009; Ozturk et al., 2007; Schrauwen et al., 2007). ESP underpins ESN practice: loosely, state trajectories must asymptotically forget initial conditions for a given input. While early practice relied on bounding the spectral radius of the random reservoir, later analyses established more precise sufficient conditions phrased in operator norms and input Lipschitz constants, clarifying the role of leakage and contractivity (Buehner & Young, 2006; Yildiz et al., 2012; Manjunath & Jaeger, 2013). These results motivate designs that keep the reservoir analyzable while guaranteeing stability of the driven dynamics.

**Structured and learned reservoirs.** Beyond i.i.d. random matrices, numerous works investigated structure to improve robustness, memory, or interpretability: orthogonal/unitary or near-isometric recurrent operators (Arjovsky et al., 2016; Henaff et al., 2016), cyclic or minimalist reservoirs (Rodan & Tiňo, 2011), and depth via stacked or leaky layers (Gallicchio & Micheli, 2017). Other lines partially *shape* the reservoir from data without full BPTT, e.g., FORCE learning that adjusts a feedback term to stabilize target dynamics (Sussillo & Abbott, 2009), or conceptors that gate ESN dynamics to represent patterns (Jaeger, 2014). More recently, *geometry-aware* designs have used local tangent-space information to inform the reservoir using patch-wise Jacobian lifting (Singh et al., 2025). These approaches show that carefully designed or weakly learned recurrent operators can preserve RC's training simplicity while improving alignment to tasks. However, few methods *learn a fixed $W$ from a single trajectory* with an explicit ESP certificate and a geometric interpretation of internal modes, which is the gap our approach addresses.

**Delay embeddings and topological summaries of dynamics.** Given a scalar or vector time series, delay-coordinate maps reconstruct diffeomorphic images of generic attractors under mild observability assumptions (Takens, 1981; Packard et al., 1980; Sauer et al., 1991). On the resulting point cloud, persistent homology extracts multi-scale topological features (e.g., $H_1$ loops) robust to sampling noise (Edelsbrunner & Harer, 2010). For dynamical data, sliding-window embeddings coupled with persistent homology capture recurrent structure and quasiperiodicity in signals (Perea & Harer, 2015). Crucially for *coordinates*, persistent *cohomology* supplies representative cocycles that can be continued to *circular coordinates*—angles on $\mathbb{S}^1$—via discrete harmonic extension with energy minimization; these have been shown to parameterize long-lived loops coherently along trajectories (de Silva & Vejdemo-Johansson, 2009). Efficient software such as RIPSER and libraries like `giotto-tda` make these computations scalable for large samples (Bauer, 2021; Tauzin et al., 2021).

**Coarse-grained transport: Ulam, transfer operators, and Koopman learning.** A complementary perspective summarizes short-time dynamics by coarse transitions between partition elements. Ulam's method approximates the Perron–Frobenius (transfer) operator by a row-stochastic matrix obtained from empirical transition counts between cells; this idea underlies a large literature on coherent sets, metastability, and Markov State Models (MSMs) (Dellnitz & Junge, 1999; Froyland, 2001; Prinz et al., 2011). In parallel, Koopman/operator-theoretic approaches yield linear surrogates of nonlinear dynamics on lifted function spaces; Dynamic Mode Decomposition (DMD) and Extended DMD (EDMD) are widely used data-driven realizations (Schmid, 2010; Williams et al., 2015; Klus et al., 2016). These lines show that coarse Markov models can encode *directionality* and slow transport directly from data, while operator-lifting connects naturally to linear evolutions in higher-dimensional representations.

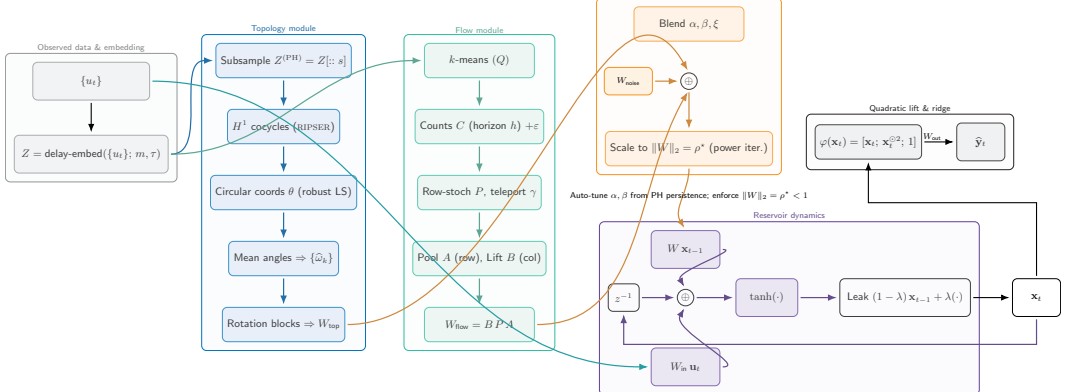

Figure 1: **PHR schematic.** From a trajectory $\{z_t\}$, a delay embedding $X$ feeds two modules: (i) *Topology* (blue): $H^1$ persistent cohomology on a subsample $Z^{(\mathrm{PH})}$ yields circular coordinates and mean angular velocities $\{\widehat{\omega}_k\}$, instantiating $2 \times 2$ rotation blocks $W_{\mathrm{top}}$; (ii) *Flow* (green): $k$-means, short–horizon counts (with pseudocounts/teleport), and stochastic pool–lift maps $A, B$ produce $W_{\mathrm{flow}}$. These are blended with small noise and *power–scaled* to $\|W\|_2 = \rho^\star$ (orange), giving a stability certificate. The leaky ESN then runs, and a *quadratic lift* drives ridge regression for $W_{\mathrm{out}}$.

**Positioning.** PHR lies at the intersection of these strands. From RC, we keep the fixed reservoir and cheap linear readout, but we replace randomness with a *learned, analyzable $W$*. From TDA, we extract persistent circular coordinates that yield data-driven *internal oscillators* (explicit $2 \times 2$ rotations) tied to long-lived loops on the embedded attractor (de Silva & Vejdemo-Johansson, 2009; Edelsbrunner & Harer, 2010). From transfer-operator discretization, we borrow empirical short-horizon Markov models and *lift* them back to neuron space to imprint local flow (Froyland, 2001; Dellnitz & Junge, 1999; Prinz et al., 2011). Unlike unitary/orthogonal RNNs (Arjovsky et al., 2016) or minimalist/structured reservoirs (Rodan & Tiňo, 2011), our construction is *data-determined* and modular: the topological ($W_{\mathrm{top}}$) and flow ($W_{\mathrm{flow}}$) components can be varied independently, and $W$ is finally scaled to meet a norm bound that yields an ESP certificate in the leaky ESN setting (Buehner & Young, 2006; Yildiz et al., 2012). Compared with Koopman/DMD/EDMD, which learn linear models in feature space (Schmid, 2010; Williams et al., 2015), we instead *learn the recurrent opera-*

*tor* of a nonlinear state machine while keeping analysis-friendly stability control and interpretability of internal modes. In a broader context of system identification vs. RC., data-driven dynamics learning spans from sparse model discovery (e.g., SINDy) (Brunton et al., 2016) and neural ODEs (Chen et al., 2018) to Koopman autoencoders (Lusch et al., 2018). These approaches aim to learn explicit evolution laws or latent linearizations and typically require gradient-based training. RC trades exact parametric fidelity for rapid training and stability guarantees. PHR aims to tighten this trade-off: retain ESN-level efficiency while injecting *geometry* (loops) and *local transport* (Markov flow) into $W$, with a clear ESP certificate and without backpropagating through time.

## 3 METHODOLOGY

**Problem statement.** Let $\{u_t\}_{t=1}^T \subset \mathbb{R}^{d_{\mathrm{obs}}}$ be an observed trajectory of an unknown dynamical system. Our goal is to *learn once, offline* a fixed recurrent operator $W \in \mathbb{R}^{N \times N}$ for an ESN such that (i) $W$ encodes *global* recurrent structure (long-lived loops) and *local* short-horizon transport, (ii) $W$ admits a *uniform contraction* certificate, and (iii) only a linear readout is trained thereafter. As overviewed in Fig. 1, we delay-embed $X = \{x_t\}_{t=m\tau}^T \subset \mathbb{R}^{md_{\mathrm{obs}}}$, extract $K$ topological modes (circular coordinates $\Rightarrow$ angular velocities $\{\omega_k\}$) to build $W_{\mathrm{top}}$ (block $2 \times 2$ rotations), estimate a short-horizon coarse Markov model $P$ and lift it to $W_{\mathrm{flow}}$, then blend and scale $W = \alpha W_{\mathrm{top}} + \beta W_{\mathrm{flow}} + \xi W_{\mathrm{noise}}$ to a target operator norm (yielding an ESN with a clean echo-state certificate).

*Intuition.* At a high level, PHR builds a single, fixed reservoir that mirrors two complementary facets of the observed dynamics. First, a delay embedding reconstructs the attractor, from which persistent cohomology extracts a few long-lived 1D loops; each loop is turned into a $(2 \times 2)$ rotation block with the loop's mean angular velocity, yielding a topology-aware operator $W_{\mathrm{top}}$ that preserves global recurrent structure (§3.1). Second, to encode short-horizon transport, we partition the embedded cloud, count transitions over a small horizon, and form a row-stochastic Markov matrix $P$; sparse pooling/lifting maps $(A, B)$ then realize a lifted flow operator $W_{\mathrm{flow}} = BP^{(\gamma)}A$ that advances coarse "mass" and projects it back to neurons (§3.2). These two channels are blended with a tiny isotropic noise term that only breaks algebraic degeneracies, and the result is scaled by power iteration to a target operator norm $\rho_\star < 1$, giving an explicit echo-state (contraction) certificate independent of architectural details (§3.3). The outcome is a reservoir whose internal modes correspond to data-driven oscillations while its local transitions reflect the empirically observed flow; stability is guaranteed by construction, and learning reduces to a single ridge-regression readout.

### 3.1 PERSISTENT COHOMOLOGY–DRIVEN OSCILLATOR SYNTHESIS

**Delay embedding and PH subsampling.** From the observed sequence, we form a delay-coordinate embedding $z_t = \left[ u_t^\top, u_{t-\tau}^\top, \ldots, u_{t-(m-1)\tau}^\top \right]^\top \in \mathbb{R}^{md_{\mathrm{obs}}}$, $t = (m-1)\tau, \ldots, T$, which (under generic observability conditions) reconstructs the attractor up to diffeomorphism (Takens, 1981; Packard et al., 1980; Sauer et al., 1991). For persistent (co)homology we operate on a *subsampled* point cloud $Z^{(\mathrm{PH})} = \{z_{t_0}, z_{t_0+s}, z_{t_0+2s}, \ldots\}$ with stride $s \in \mathbb{N}$ to control the $O(n^2)$ distance cost. Let $n_{\mathrm{PH}} = |Z^{(\mathrm{PH})}|$ and let $D \in \mathbb{R}_{\geq 0}^{n_{\mathrm{PH}} \times n_{\mathrm{PH}}}$ be the Euclidean distance matrix.

**Persistent cohomology and circular coordinates.** We compute Vietoris–Rips persistent *cohomology* up to degree one on the metric space $(Z^{(\mathrm{PH})}, D)$ over a prime field $\mathbb{F}_p$, obtaining $H^1$ intervals $\left\{ (b_\ell, d_\ell) \right\}_\ell$ and representative 1-cocycles $\{c_\ell\}$ (Edelsbrunner & Harer, 2010). For each selected class $\ell$, we pick a working scale $\varepsilon_\ell \in (b_\ell, d_\ell)$; the implementation defaults to a *near-death* choice $\varepsilon_\ell = d_\ell - 10^{-6}$ to ensure a sufficiently connected 1-skeleton while staying within the class's lifespan. Let $G_\ell = (V, E_\ell)$ be the Rips 1-skeleton at threshold $\varepsilon_\ell$, i.e., $E_\ell = \{(i,j) \in V^2 : i < j, D_{ij} \leq \varepsilon_\ell\}$. Following de Silva & Vejdemo-Johansson (2009), we lift the cocycle $c_\ell$ to edge phases $\alpha_{ij}^{(\ell)} \in (-\frac{1}{2}, \frac{1}{2}]$ on $E_\ell$ by mapping coefficients $a_{ij} \in \mathbb{F}_p$ to $a_{ij}/p$ and wrapping to $(-\frac{1}{2}, \frac{1}{2}]$. We then solve a *discrete harmonic extension* problem for vertex potentials $\vartheta^{(\ell)} \in \mathbb{R}^{n_{\mathrm{PH}}}$ that minimize the weighted Dirichlet energy subject to matching the lifted edge phases in least squares

$$\vartheta^{(\ell)} = \arg\min_{\vartheta \in \mathbb{R}^{n_{\mathrm{PH}}}} \sum_{(i,j) \in E_\ell} w_{ij} \left( \vartheta_j - \vartheta_i - \alpha_{ij}^{(\ell)} \right)^2, \qquad w_{ij} = \begin{cases} \dfrac{1}{D_{ij} + \varepsilon} & \text{(default)}, \\ 1 & \text{(unit weights)}, \end{cases} \tag{1}$$

with a small $\varepsilon > 0$ for numerical safety. Writing $M$ for the oriented incidence of $G_\ell$ and $W = \mathrm{diag}(w_{ij})$, the normal equations are $(L + \mu I)\vartheta = b$, $L := M^\top W M$, $b := M^\top W\alpha$, $\mu \ll 1$, which we solve *per connected component* with a gauge anchor (fix one vertex) and a tiny Tikhonov term $\mu$ to regularize near-singular components; the implementation uses a sparse direct solve with an LSQR fallback (Paige & Saunders, 1982). The circular coordinate (angle) associated with class $\ell$ is then $\theta_i^{(\ell)} = \mathrm{wrap}_{(-\pi,\pi]}(2\pi\vartheta_i^{(\ell)})$, $i \in V$, providing an $\mathbb{S}^1$-valued coordinate that varies coherently along the loop (de Silva & Vejdemo-Johansson, 2009).

**Circular coordinates $\to$ oscillators.** Let $\theta_t^{(\ell)}$ denote the angle associated with $z_t \in Z^{(\mathrm{PH})}$ (indices inherited from time by subsampling). We estimate a *mean angular velocity* by wrapped least squares (empirically equivalent to the average wrapped increment for unit time steps from circular statistics (Mardia & Jupp, 2000)): $\widehat\omega_\ell = \arg\min_{\omega\in(-\pi,\pi]} \sum_t \left\| \mathrm{wrap}_{(-\pi,\pi]}(\theta_{t+1}^{(\ell)} - \theta_t^{(\ell)} - \omega) \right\|^2 \approx \mathrm{mean}(\mathrm{wrap}_{(-\pi,\pi]}(\theta_{t+1}^{(\ell)} - \theta_t^{(\ell)}))$. Each $\widehat\omega_\ell$ parameterizes a stable $2 \times 2$ rotation block

$$R(\widehat\omega_\ell; \rho_{\mathrm{rot}}) = \rho_{\mathrm{rot}}\begin{bmatrix} \cos\widehat\omega_\ell & -\sin\widehat\omega_\ell \\ \sin\widehat\omega_\ell & \cos\widehat\omega_\ell \end{bmatrix}, \qquad 0 < \rho_{\mathrm{rot}} < 1, \tag{2}$$

and $W_{\mathrm{top}}$ is formed by embedding these blocks in a block-diagonal matrix and randomly permuting coordinates to distribute the oscillator pairs across the reservoir; remaining coordinates receive decaying radii in $(\rho_{\min}, \rho_{\max})$. This realizes *topology-aware internal oscillators* aligned with the system's dominant loops.

**Selecting $K$ and auto-tuning the blend.** We first cap the number of requested loops by $K_{\max}$ and keep the top-$K_{\max}$ classes by persistence. After PH, we apply a *relative persistence* threshold $\gamma \in [0,1]$ to decide which loops survive to synthesis: $\mathcal{I}_{\mathrm{keep}} = \{\ell : (d_\ell - b_\ell) \geq \gamma \max_{\ell'}(d_{\ell'} - b_{\ell'})\}$, $K_{\mathrm{final}} = |\mathcal{I}_{\mathrm{keep}}|$. Let $P_\ell := d_\ell - b_\ell$ denote persistence and $P_{\max} := \max_\ell P_\ell$. We compute a scalar "loop strength" $s = \frac{1}{K_{\mathrm{final}}}\sum_{\ell\in\mathcal{I}_{\mathrm{keep}}} \frac{P_\ell}{P_{\max}} \in [0,1]$, and set the topological blend weight by clamping a linear map of $s$ into user bounds: $\alpha_{\mathrm{top}} = \mathrm{clip}(\alpha_{\min} + (\alpha_{\max} - \alpha_{\min})s, 0, 1-\xi)$, $\beta_{\mathrm{flow}} = 1 - \xi - \alpha_{\mathrm{top}}$, where $\xi$ is the fixed noise fraction. If $K_{\mathrm{final}} = 0$ or the PH step fails, we default to a flow-dominant setting $\alpha_{\mathrm{top}} = 0$, $\beta_{\mathrm{flow}} = 1 - \xi$. This *auto-tuner* makes the topology/flow trade-off responsive to the evidence in the data while keeping $W$ analyzable and the ESP scaling invariant to the choice.

Taken together, the PH pipeline and the oscillator synthesis ensure that $W_{\mathrm{top}}$ carries a small number of explicitly interpretable modes whose frequencies are estimated directly from the data. However, these *global oscillators do not*, by themselves, encode how probability mass drifts locally across the attractor; this motivates a complementary flow channel in which coarse, data-driven transport is estimated and then lifted back to the reservoir state space.

## 3.2 Coarse Flow Estimation and the Lifted Markov Operator

**Partitioning the embedded state space.** To encode short-horizon directionality, we discretize the delay-embedded point cloud $Z = \{z_t\} \subset \mathbb{R}^{md_{\mathrm{obs}}}$ into $Q$ clusters by $k$-means (Lloyd's algorithm) (Lloyd, 1982). Let $\{c_q\}_{q=1}^Q$ denote the centroids and define the assignment $s_t = \arg\min_{q\in\{1,\dots,Q\}} \|z_t - c_q\|_2$, $t = (m-1)\tau, \dots, T$. This yields a coarse partition of the embedded attractor.

**Short-horizon Markov chain from transition counts.** Fix a horizon $h \in \mathbb{N}$. We form the empirical count matrix $C \in \mathbb{R}_{\geq 0}^{Q\times Q}$, $C_{ij} = \#\{t : s_t = i, s_{t+h} = j\}$, $i,j \in \{1,\dots,Q\}$. To avoid degenerate (all-zero) rows, we add a small pseudocount $\varepsilon > 0$ before normalization and obtain a row-stochastic Markov matrix by $P_{ij} = \frac{C_{ij}+\varepsilon}{\sum_{j'}(C_{ij'}+\varepsilon)}$, $\sum_{j=1}^Q P_{ij} = 1 \; \forall i$. We apply *teleportation* (Google–PageRank style) (Brin & Page, 1998; Langville & Meyer, 2012) to regularize nearly reducible chains and cure rare sinks: $P^{(\gamma)} = (1-\gamma)P + \gamma\mathbf{1}u^\top$, $u = \frac{1}{Q}\mathbf{1}$, $\gamma \in [0,1)$, which preserves row-stochasticity and ensures a positive recurrent, aperiodic surrogate for coarse transport. This construction follows the spirit of Ulam's method for approximating the Perron–Frobenius (transfer) operator via finite-state Markov models (Dellnitz & Junge, 1999; Froyland, 2001) and is consistent with operator-theoretic discretizations used in Koopman learning (Klus et al., 2016; Williams et al., 2015). In practice $h = 1$ captures most local flow, while $h > 1$ can smooth fast noise.

**Stochastic pool–lift maps.** Let $N$ be the reservoir dimension. We connect coarse dynamics to neurons through two sparse, stochastic linear maps: $A \in \mathbb{R}^{Q \times N}$, $B \in \mathbb{R}^{N \times Q}$. Matrix $A$ *pools* neuron activations (the reservoir state) into coarse cells and is *row-stochastic*: each row of $A$ has exactly `nzr` nonzeros of equal weight $1/\text{nzr}$ (uniform-sparse selection), hence for $x \in \mathbb{R}^N$, $(Ax)_q = \sum_{i=1}^{N} A_{qi} x_i$, $\sum_{i=1}^{N} A_{qi} = 1 \ \forall q$. Matrix $B$ *lifts* coarse activations back to neurons and is *column-stochastic*: each column of $B$ has exactly `nzc` nonzeros of equal weight $1/\text{nzc}$, so for $r \in \mathbb{R}^Q$, $(Br)_i = \sum_{q=1}^{Q} B_{iq} r_q$, $\sum_{i=1}^{N} B_{iq} = 1 \ \forall q$. We sample the support of $A$ and $B$ uniformly without replacement (geometry-aware variants are compatible but not required for the guarantees used later). Row-stochasticity makes $A$ a *convex averaging* over neurons within each coarse cell; column-stochasticity distributes each coarse value as a convex combination over its recipient neurons.

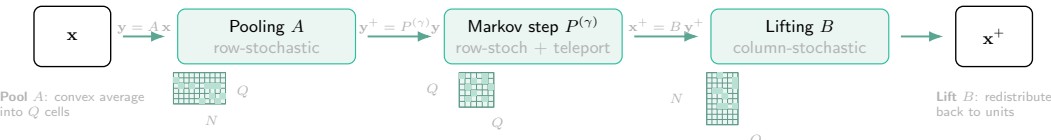

Figure 2: **Zoom on the lifted Markov operator** ($W_{\text{flow}}$). This panel magnifies the green *Flow* module from Fig. 1. A reservoir state $\mathbf{x} \in \mathbb{R}^N$ is first *pooled* to coarse cells via a row–stochastic map $A \in \mathbb{R}^{Q \times N}$ (rows sum to 1), yielding $\mathbf{y} = A\mathbf{x}$. Short–horizon dynamics on the coarse graph are applied by $P^{(\gamma)} \in \mathbb{R}^{Q \times Q}$, a row–stochastic Markov matrix with optional teleport parameter $\gamma$, giving $\mathbf{y}^+ = P^{(\gamma)}\mathbf{y}$. Finally, the signal is *lifted* back to neurons by a column–stochastic map $B \in \mathbb{R}^{N \times Q}$ (columns sum to 1), producing $\mathbf{x}^+ = B\mathbf{y}^+$. Altogether, $W_{\text{flow}} = B P^{(\gamma)} A$ with $\mathbf{x}^+ = W_{\text{flow}}\mathbf{x}$, which is the contractive, data–driven component blended into the stable recurrent operator in the main schematic.

**Lifted Markov operator.** We define the *lifted flow operator* $W_{\text{flow}} = B P^{(\gamma)} A \in \mathbb{R}^{N \times N}$. Operationally, $A$ first pools the reservoir state $x \in \mathbb{R}^N$ down to a coarse state $r = Ax \in \mathbb{R}^Q$; the Markov step $r^+ = P^{(\gamma)}r$ advances coarse mass along observed short-time transport; $B$ lifts $r^+$ back to the neuron space $x^+ = Br^+$. The operator is nonnegative by construction. In induced norms, $\|A\|_\infty = 1$ and $\|P^{(\gamma)}\|_\infty = 1$ by row-stochasticity; $\|B\|_1 = 1$ by column-stochasticity. We do not rely on a standalone spectral bound for $W_{\text{flow}}$; rather, the *global* contraction is enforced later by power-iteration scaling of the blended $W$ (§3.3), which yields a sufficient condition for the ESN's echo-state property independent of the particular sparsity pattern of $A$ and $B$ (Buehner & Young, 2006; Yildiz et al., 2012). This separation keeps the flow imprint faithful to data while placing stability under explicit control.

The operator $W_{\text{flow}}$ therefore acts as a Markovian stencil on the reservoir: pooled activity approximately follows the empirical coarse chain, and Lemma 3.1 quantifies how closely $AW_{\text{flow}}B$ tracks $P^{(\gamma)}$ once the calibration defect of $AB$ is controlled. In the next step, we combine this transport channel with the oscillatory operator $W_{\text{top}}$ and a small isotropic noise term, and then apply a single global scaling so that the resulting recurrent matrix $W$ simultaneously inherits these structures and satisfies a uniform contraction bound.

### 3.3 BLENDED RECURRENT OPERATOR, ECHO-STATE SCALING, AND READOUT TRAINING

**Blending topology and flow with degeneracy-breaking noise.** Let $W_{\text{top}} \in \mathbb{R}^{N \times N}$ be the block-permuted rotation–decay operator synthesized from persistent circular coordinates (§3.1), and let $W_{\text{flow}} = B P^{(\gamma)} A \in \mathbb{R}^{N \times N}$ be the lifted Markov operator (§3.2). We form a pre-scaled blend

$$W_{\text{blend}} = \alpha_{\text{top}} W_{\text{top}} + \beta_{\text{flow}} W_{\text{flow}} + \xi W_{\text{noise}}, \quad \alpha_{\text{top}}, \beta_{\text{flow}}, \xi \geq 0, \quad \alpha_{\text{top}} + \beta_{\text{flow}} + \xi = 1, \quad (3)$$

where $W_{\text{noise}}$ is a zero-mean Gaussian matrix normalized to unit operator 2-norm and then scaled by a small constant. The noise term breaks algebraic degeneracies (e.g., repeated eigenvalues or exact invariant subspaces) and improves numerical conditioning without affecting stability guarantees, since a global norm scaling will be imposed next. We implement an *auto-tuner* that chooses $\alpha_{\text{top}}$

and $\beta_{\text{flow}}$ from persistence statistics: given the set of selected loops $\mathcal{I}_{\text{keep}}$, define the mean relative persistence $s := \frac{1}{|\mathcal{I}_{\text{keep}}|} \sum_{\ell \in \mathcal{I}_{\text{keep}}} \frac{P_\ell}{P_{\max}} \in [0, 1]$, $P_\ell = d_\ell - b_\ell$, $P_{\max} = \max_\ell P_\ell$, and set

$$\alpha_{\text{top}} = \text{clip}(\alpha_{\min} + (\alpha_{\max} - \alpha_{\min}) s, \ 0, \ 1 - \xi), \qquad \beta_{\text{flow}} = 1 - \xi - \alpha_{\text{top}}, \tag{4}$$

with user bounds $\alpha_{\min} \le \alpha_{\max}$ and $\text{clip}(a; \ell, u) := \min\{u, \max\{\ell, a\}\}$. If $\mathcal{I}_{\text{keep}} = \varnothing$ (no trustworthy loops) or the PH step fails, the tuner falls back to a flow-dominant setting $\alpha_{\text{top}} = 0$, $\beta_{\text{flow}} = 1 - \xi$.

**Power-iteration scaling to a target operator norm.** To certify stability for the leaky ESN update, we scale $W_{\text{blend}}$ to a prescribed operator norm $\rho_\star \in (0, 1)$

$$W = \frac{\rho_\star}{\widehat{\sigma}_{\max}(W_{\text{blend}})} W_{\text{blend}}, \qquad \widehat{\sigma}_{\max}(W_{\text{blend}}) \approx \|W_{\text{blend}}\|_2, \tag{5}$$

where $\widehat{\sigma}_{\max}$ is estimated by a standard power iteration on $W_{\text{blend}}^\top W_{\text{blend}}$:

$$v_{k+1} = \frac{(W_{\text{blend}}^\top W_{\text{blend}}) v_k}{\|(W_{\text{blend}}^\top W_{\text{blend}}) v_k\|_2}, \qquad \widehat{\sigma}_{\max} \leftarrow \|W_{\text{blend}} v_k\|_2, \tag{6}$$

for a fixed number of iterations. This provides a reliable approximation of the largest singular value under mild conditions (Golub & Loan, 2013; Trefethen & III, 1997). In practice we include NaN/infinity guards; if the estimate is degenerate, the implementation returns a safe zero matrix (trivial contraction). Choosing $\rho_\star$ with a margin accounts for small overestimation errors and keeps the final contraction budget conservative.

**Structural fidelity (proofs are presented in Appendix A).** Define the lifted flow $W_{\text{flow}} := BPA$ and the pre-scaled blend $W_{\text{blend}} := \alpha W_{\text{top}} + \beta W_{\text{flow}} + \xi W_{\text{noise}}$, $\alpha, \beta, \xi \ge 0$, $\alpha + \beta + \xi = 1$, where $W_{\text{noise}}$ is any matrix normalized to $\|W_{\text{noise}}\|_2 = 1$. Let $s := \|W_{\text{blend}}\|_2$ and $W := \rho_\star W_{\text{blend}}/s$ denote the finally used recurrent operator with $\|W\|_2 = \rho_\star \in (0, 1)$ (§3.3). By Lemma A.1, the block-permuted rotation–decay operator $W_{\text{top}}$ is normal with spectrum $\{\rho_{\text{rot}} e^{\pm i\omega_k}\}_{k=1}^K \cup \{r_j\}_{j=1}^{N-2K}$ (with $r_j \in (\rho_{\min}, \rho_{\max})$) and admits an orthogonal decomposition $\mathbb{R}^N = \bigoplus_{k=1}^K E_k \oplus E_\perp$, where each $E_k$ is a 2D invariant plane on which $W_{\text{top}} = \rho_{\text{rot}} R(\omega_k)$, a property unchanged by the subsequent coordinate permutation.

**Lemma 3.1** (Pooled coarse-flow identity and deviation). *Let $A \in \mathbb{R}^{Q \times N}$ be row–stochastic, $B \in \mathbb{R}^{N \times Q}$ column–stochastic, and $P \in \mathbb{R}^{Q \times Q}$ row–stochastic. For any $r \in \mathbb{R}^Q$ and $x := Br \in \mathbb{R}^N$, $\|A W_{\text{flow}} x - P r\|_\infty \le 2 \|AB - I_Q\|_\infty \|r\|_\infty$, since $\|AB\|_\infty = \|P\|_\infty = 1$ by row–stochasticity. An analogous bound holds in $\ell_1$ with $\|\cdot\|_1$.*

*Remark (optional support coupling to reduce calibration).* In all reported results we sample the supports of $A$ (row-stochastic pooling) and $B$ (column-stochastic lifting) independently at random. Another strategy—orthogonal to our results—is to *couple* these supports per coarse cell $q$. Concretely, choose the lift of cell $q$ to target the same neuron subset used to pool that cell, and make these subsets disjoint across $q$. This makes $AB$ diagonally dominant, shrinking $\|AB - I_Q\|$ in Lemma 3.1 and the induced defect $\Delta := (AB) P^{(\gamma)} (AB) - P^{(\gamma)}$ in Proposition 3.2(ii). In the extreme one-hot case (one neuron per cell for both pooling and lifting), $AB = I_Q$ exactly; with small shared supports of size $s > 1$, $AB$ becomes diagonal with entries $1/s$ and off-diagonals 0, yielding $\|AB - I_Q\|_\infty = 1 - 1/s$. This refinement leaves $P^{(\gamma)}$, $W_{\text{top}}$, and the global scaling to $\|W\|_2 = \rho_\star < 1$ unchanged, thus preserving the echo-state certificate while tightening the coarse-flow fidelity term.

**Proposition 3.2** (Two-channel fidelity of the blended operator). *Let $W = \rho_\star W_{\text{blend}}/s$ with $s = \|W_{\text{blend}}\|_2$. Then:* (i) Persistence of oscillatory eigenpairs. *For each $k \in \{1, \dots, K\}$ there exists an eigenvalue $\lambda_k(W)$ such that $\left|\lambda_k(W) - \rho_\star \frac{\alpha \rho_{\text{rot}} e^{i\omega_k}}{s}\right| \le \rho_\star \frac{\beta \|W_{\text{flow}}\|_2 + \xi \|W_{\text{noise}}\|_2}{s}$, and likewise for the conjugate pair. In particular, when $s \ge \alpha \|W_{\text{top}}\|_2 - (\beta \|W_{\text{flow}}\|_2 + \xi \|W_{\text{noise}}\|_2)$, the oscillatory eigenvalues of $W$ are contained in discs centered at $\rho_\star \alpha \rho_{\text{rot}} e^{\pm i\omega_k}/s$ with radius $\rho_\star (\beta \|W_{\text{flow}}\|_2 + \xi \|W_{\text{noise}}\|_2)/s$.*[1] (ii) Coarse-flow fidelity on the lifted subspace. *For any $r \in \mathbb{R}^Q$ and $x := Br \in \mathbb{R}^N$, $\left\| A W x - \rho_\star \frac{\beta}{s} P^{(\gamma)} r \right\|_2 \le$*

---

[1] Since $W_{\text{top}}$ is normal, additive perturbations shift eigenvalues by at most the perturbation 2-norm (Trefethen & III, 1997, Ch. 2). The centers/radii follow by reverse triangle inequality and the block spectrum of $W_{\text{top}}$.

$\rho_\star \left[ \frac{\beta}{s} \left\| (AB) P^{(\gamma)} (AB) - P^{(\gamma)} \right\|_2 + \frac{\alpha}{s} \|A\|_2 \|W_{\text{top}}\|_2 \|B\|_2 + \frac{\xi}{s} \|A\|_2 \|W_{\text{noise}}\|_2 \|B\|_2 \right] \|r\|_2$. *Hence the pooled one-step action of $W$ on lifted coarse states approximates a scaled Markov step with error decomposed into (a) the calibration defect of $AB$, and (b) leakage from the topological and noise channels, each controlled by blend weights and operator norms. In particular, if $AB = I_Q$ and $\alpha, \xi$ are small, then $A W B$ is a small-norm perturbation of $(\rho_\star \beta / s) P^{(\gamma)}$ on $\mathbb{R}^Q$.*

The two statements formalize, respectively, (1) the *exact pooled identity* and *calibration-controlled* deviation of the lifted Markov channel, and (2) their *joint persistence* under blending and global scaling. They provide structural guarantees beyond the echo-state contraction: PHR preserves data-driven oscillatory modes near their target frequencies and transports pooled mass nearly according to the empirical coarse dynamics, with explicit, verifiable perturbation budgets determined by $(\alpha, \beta, \xi)$ and by the calibration of $(A, B)$.

After $W$ is fixed, we advance the reservoir with the teacher-forced inputs $\{u_t\}$ (cf. Eqn. 11) to collect features $\varphi_t \in \mathbb{R}^F$: $\varphi_t = \left[ x_t ; \mathbf{1}_{\text{poly}} (x_t \odot x_t) ; \mathbf{1}_{\text{poly}} \right]$, where $\mathbf{1}_{\text{poly}} \in \{0, 1\}$ encodes the quadratic/constant augmentation and $\odot$ denotes the Hadamard product. We discard an initial "washout" of $D$ steps to remove transients and form the design matrix $\Phi \in \mathbb{R}^{(T-D) \times F}$ and targets $Y \in \mathbb{R}^{(T-D) \times d_{\text{out}}}$. The readout is fitted by *ridge regression* with penalty $\alpha > 0$ as

$$W_{\text{out}} = \arg \min_{W \in \mathbb{R}^{d_{\text{out}} \times F}} \left\| Y - \Phi W^\top \right\|_F^2 + \alpha \|W\|_F^2, \qquad \Rightarrow \qquad W_{\text{out}}^\top = (\Phi^\top \Phi + \alpha I)^{-1} \Phi^\top Y, \quad (7)$$

computed with a standard linear solver, no intercept since a constant feature is included) (Hoerl & Kennard, 1970). This preserves the hallmark ESN training efficiency: (7) is a single convex solve whose complexity is $O(TF^2 + F^3)$ and typically negligible compared to data generation.

*System-level dynamics of PHR is discussed in §A.3, and an intuitive summary is provided in §A.2.*

## 4 EXPERIMENTS

**Setup.** We evaluate PHR on seven standard benchmarks. The first group comprises three canonical chaotic attractors—**Lorenz-63** (Lorenz, 1963), **Rössler** (Rossler, 1976), and the hyper-chaotic **Chen–Ueta** flow (Chen & Ueta, 1999). The second group contains four real-world time series: the **BIDMC** PPG/respiration record (Goldberger et al., 2000), the **MIT–BIH** Arrhythmia ECG trace (Moody & Mark, 2001), the **Santa Fe B** cardiorespiratory polysomnography sequence (Jaeger, 2007), and the SILSO monthly **sunspot index** (World Data Center SILSO, 2020). As baselines we include the standard single-layer **ESN** (Jaeger, 2001), the Simple-Cycle Reservoir (**SCR**) (Li et al., 2024), the Cycle Reservoir with Jumps (**CRJ**) (Rodan & Tino, 2012), the two-core **MCI-ESN** (Liu et al., 2024), and the three-layer **DeepESN** (Gallicchio & Micheli, 2017). Every method is allotted exactly 300 recurrent units in total (implemented as $3 \times 100$ for DeepESN), and each baseline is hyper-tuned within the same computational budget. *(details are provided in Appendix B.)*

We appraise every model with four mutually reinforcing criteria. *First*, the **NRMSE** gauges point-wise accuracy via the root-mean-square error normalised by the variance of the reference signal. *Second*, we introduce the **Valid Prediction Time Ratio** (**VPT**): the earliest instant $t$ at which the normalised deviation $\delta(t) = |y_t - \hat{y}_t|_2 / |y_t|_2$ breaches a fixed threshold $\theta$, expressed in Lyapunov units as $\text{VPT} = t/T_L$ (Pathak et al., 2018). VPT therefore quantifies "how long the forecast can be trusted." *Third*, global attractor fidelity is captured by the **Attractor Deviation** (**ADev**), the volume of the symmetric difference between predicted and true phase-space occupancies on a fixed grid, normalised by their union: (Zhai et al., 2023). *Finally*, we overlay the log power-spectral densities (estimated with Welch's method (Welch, 1967)) of selected observables; agreement is assessed visually through the alignment of peaks, harmonic envelopes, and broadband roll-off. Taken together, NRMSE measures short-term trajectory accuracy, VPT reveals the time-span over which forecasts remain reliable, ADev scores faithfulness to the global geometry of the attractor, and the PSD overlay inspects concordance in the frequency domain (cf. Fig. 4). *(see Appendix B for details)*

**Quantitative results.** Table 1 summarizes open-loop performance on *MIT–BIH*, *BIDMC*, *Sunspot*, and *Santa Fe*, and closed-loop forecasting on *Lorenz–63*, *Rössler*, and *Chen*. We report normalized RMSE (NRMSE; lower is better), and—only for chaotic systems—valid prediction time (VPT; higher is better) and average deviation (ADev; lower is better). On all four real-world series, PHR

| Dataset | H / Metric | NRMSE / VPT / ADev | | | | | |
|---|---|---|---|---|---|---|---|
| | | ESN | SCR | CRJ | MCI-ESN | Deep-ESN | PHR |
| MIT–BIH | 300 | 2.3537 ± 0.5472 | 2.0321 ± 0.5211 | 1.5698 ± 0.3365 | 1.1443 ± 0.0601 | 2.7398 ± 0.9618 | **0.5320 ± 0.0442** |
| | 600 | 1.7575 ± 0.3692 | 1.5440 ± 0.3461 | 1.2588 ± 0.2125 | 1.0557 ± 0.0328 | 2.0187 ± 0.6573 | **0.5417 ± 0.0236** |
| | 1000 | 1.4867 ± 0.2252 | 1.3729 ± 0.1998 | 1.2115 ± 0.1131 | 1.1273 ± 0.0156 | 1.6530 ± 0.4123 | **0.5833 ± 0.0123** |
| BIDMC | 300 | 0.4468 ± 0.0287 | 0.4473 ± 0.0292 | 0.4476 ± 0.0310 | 0.5159 ± 0.0326 | 0.4959 ± 0.0298 | **0.3655 ± 0.0143** |
| | 600 | 0.4305 ± 0.0269 | 0.4350 ± 0.0235 | 0.4336 ± 0.0249 | 0.4875 ± 0.0272 | 0.4951 ± 0.0263 | **0.3571 ± 0.0151** |
| | 1000 | 0.5186 ± 0.0198 | 0.4941 ± 0.0215 | 0.4979 ± 0.0224 | 0.5381 ± 0.0237 | 0.5683 ± 0.0239 | **0.4352 ± 0.0133** |
| Sunspot | 300 | 0.6008 ± 0.0214 | 1.1103 ± 0.1323 | 1.2180 ± 0.1856 | 0.4680 ± 0.0090 | 0.5856 ± 0.0247 | **0.2505 ± 0.0011** |
| | 600 | 0.5501 ± 0.0295 | 1.0126 ± 0.1303 | 1.1300 ± 0.1565 | 0.4238 ± 0.0140 | 0.5020 ± 0.0232 | **0.2093 ± 0.0015** |
| | 1000 | 0.5235 ± 0.0262 | 0.9615 ± 0.1216 | 1.0781 ± 0.1191 | 0.4064 ± 0.0100 | 0.4807 ± 0.0196 | **0.2044 ± 0.0010** |
| Santa Fe | 300 | 0.2761 ± 0.0014 | 0.2927 ± 0.0032 | 0.2861 ± 0.0047 | 0.2697 ± 0.0031 | 0.2876 ± 0.0038 | **0.1485 ± 0.0003** |
| | 600 | 0.2366 ± 0.0014 | 0.2523 ± 0.0053 | 0.2414 ± 0.0037 | 0.2612 ± 0.0217 | 0.2437 ± 0.0049 | **0.1262 ± 0.0003** |
| | 1000 | 0.2512 ± 0.0010 | 0.2642 ± 0.0042 | 0.2565 ± 0.0031 | 0.2680 ± 0.0175 | 0.2579 ± 0.0039 | **0.1361 ± 0.0003** |
| Lorenz | 200 | 0.0013 ± 0.0016 | 0.0027 ± 0.0055 | 0.0039 ± 0.0071 | 0.0011 ± 0.0015 | 0.0023 ± 0.0036 | **0.0004 ± 0.0005** |
| | 400 | 0.0428 ± 0.0660 | 0.0657 ± 0.0949 | 0.0765 ± 0.1194 | 0.0335 ± 0.0388 | 0.0717 ± 0.0978 | **0.0075 ± 0.0103** |
| | 600 | 0.3416 ± 0.2345 | 0.4236 ± 0.2542 | 0.4643 ± 0.2803 | 0.3618 ± 0.2903 | 0.4149 ± 0.2141 | **0.2121 ± 0.1877** |
| | 800 | 0.7704 ± 0.1139 | 0.7953 ± 0.1477 | 0.7703 ± 0.2099 | 0.7799 ± 0.1690 | 0.7965 ± 0.1381 | **0.6495 ± 0.1928** |
| | 1000 | 0.9385 ± 0.1017 | 0.9469 ± 0.1306 | 0.9542 ± 0.1443 | 0.9370 ± 0.1432 | 0.9649 ± 0.1112 | **0.8481 ± 0.1358** |
| | VPT (↑) | 9.18 ± 1.71 | 8.70 ± 1.72 | 8.98 ± 1.83 | 9.37 ± 1.83 | 8.67 ± 1.78 | **10.94 ± 1.65** |
| | ADev (↓) | 29.78 ± 11.36 | **28.80 ± 9.98** | 32.21 ± 9.71 | 30.93 ± 12.79 | 30.38 ± 9.95 | 29.11 ± 9.53 |
| Rössler | 200 | 0.0010 ± 0.0021 | 0.0007 ± 0.0012 | 0.0012 ± 0.0020 | 0.0004 ± 0.0006 | 0.0014 ± 0.0026 | **0.0002 ± 0.0001** |
| | 400 | 0.0019 ± 0.0035 | 0.0031 ± 0.0080 | 0.0024 ± 0.0040 | 0.0009 ± 0.0011 | 0.0026 ± 0.0043 | **0.0003 ± 0.0003** |
| | 600 | 0.0035 ± 0.0062 | 0.0062 ± 0.0107 | 0.0066 ± 0.0142 | 0.0020 ± 0.0030 | 0.0115 ± 0.0455 | **0.0007 ± 0.0007** |
| | 800 | 0.0050 ± 0.0082 | 0.0094 ± 0.0144 | 0.0094 ± 0.0200 | 0.0028 ± 0.0043 | 0.0166 ± 0.0641 | **0.0010 ± 0.0011** |
| | 1000 | 0.0064 ± 0.0103 | 0.0136 ± 0.0181 | 0.0127 ± 0.0270 | 0.0038 ± 0.0060 | 0.0211 ± 0.0773 | **0.0013 ± 0.0016** |
| | VPT (↑) | 7.89 ± 3.90 | 6.97 ± 4.30 | 8.33 ± 4.36 | 9.88 ± 4.59 | 8.28 ± 4.50 | **11.33 ± 4.66** |
| | ADev (↓) | 16.97 ± 8.19 | 22.42 ± 11.62 | 19.55 ± 11.00 | **12.21 ± 6.86** | 20.22 ± 22.09 | 13.11 ± 8.02 |
| Chen | 200 | 0.2463 ± 0.3235 | 0.3157 ± 0.3117 | 0.2507 ± 0.2441 | 0.1756 ± 0.2441 | 0.3554 ± 0.3156 | **0.1278 ± 0.2424** |
| | 400 | 0.9348 ± 0.1783 | 0.9767 ± 0.1836 | 0.9442 ± 0.1518 | 0.8893 ± 0.1708 | 0.9727 ± 0.1805 | **0.7958 ± 0.2006** |
| | 600 | 1.1273 ± 0.1433 | 1.1212 ± 0.1350 | 1.1310 ± 0.1081 | 1.0824 ± 0.1320 | 1.1549 ± 0.1232 | **1.0555 ± 0.1199** |
| | 800 | 1.2210 ± 0.1075 | 1.2052 ± 0.1006 | 1.2132 ± 0.0868 | 1.1814 ± 0.1096 | 1.2221 ± 0.1047 | **1.1632 ± 0.0878** |
| | 1000 | 1.2535 ± 0.0961 | 1.2475 ± 0.0804 | 1.2594 ± 0.0772 | 1.2355 ± 0.0897 | 1.2627 ± 0.0852 | **1.2316 ± 0.0737** |
| | VPT (↑) | 3.46 ± 0.92 | 3.10 ± 0.84 | 3.29 ± 0.62 | 3.59 ± 0.73 | 3.01 ± 0.92 | **4.18 ± 0.93** |
| | ADev (↓) | 52.45 ± 10.92 | 55.17 ± 11.95 | **51.24 ± 10.03** | 58.22 ± 12.37 | 54.65 ± 8.89 | 55.42 ± 15.53 |

Table 1: NRMSE (mean±s.d.) across horizons (H) and, for chaotic datasets, additional rows with VPT (↑) and ADev (↓). Chaotic benchmarks are evaluated in **closed-loop** mode; real-world datasets use **open-loop**. Best and second-best per row are shown in **bold** and underlined, resp.

| Variant | Change (relative to default) | NRMSE@H=600 ↓ | VPT ↑ | ADev ↓ |
|---|---|---|---|---|
| **PHR (default)** | - | **0.212 ± 0.188** | **10.94 ± 1.65** | **29.11 ± 9.53** |
| No auto-tune | Fix $(\alpha, \beta, \xi) = (0.60, 0.35, 0.05)$ (ignore persistence strength) | 0.238 ± 0.191 | 8.90 ± 1.62 | 33.90 ± 11.92 |
| PH → PCA-$\omega$ (no PH) | Replace persistent circular coordinates by PCA phase surrogate (§3.1) | 0.258 ± 0.195 | 9.42 ± 1.70 | 34.78 ± 12.15 |
| Flow-only | $\alpha=0$, $\beta=1-\xi$ (no $W_{\text{top}}$ oscillators) | 0.279 ± 0.198 | 9.10 ± 1.66 | 34.12 ± 11.94 |
| Topology-only | $\beta=0$, $\alpha=1-\xi$ (no $W_{\text{flow}}$ transport imprint) | 0.341 ± 0.235 | 9.22 ± 1.59 | 36.89 ± 10.07 |
| $\varepsilon$ choice: midlife | $\varepsilon=(b+d)/2$ instead of near-death for Rips 1-skeleton | 0.226 ± 0.189 | 10.05 ± 1.58 | 33.41 ± 12.03 |
| No teleport | $\gamma=0$ (pure empirical $P$) | 0.233 ± 0.190 | 9.82 ± 1.61 | 33.56 ± 12.96 |
| No pseudocounts | Remove $\varepsilon$-counts in $C_{ij}$ (rows can be sparse/zero) | 0.255 ± 0.197 | 9.61 ± 1.67 | 34.52 ± 11.40 |
| Denser pool/lift | $A$ : nzr= 8, $B$ : nzc= 24 (still stochastic) | 0.224 ± 0.189 | 9.10 ± 1.55 | 33.32 ± 10.90 |
| Stricter loop cap | $K_{\max}=1$ (keep only strongest loop) | 0.245 ± 0.193 | 9.72 ± 1.63 | 34.21 ± 13.01 |
| No robust PH solver | Drop Tikhonov + LSQR fallback (single-component solve only) | 0.247 ± 0.196 | 9.54 ± 1.69 | 35.73 ± 10.18 |
| *Unsafe (for reference):* no spectral scaling | | divergent in 37% runs | | |

Table 2: **Ablation study on Lorenz–63 (closed-loop).** Default configuration follows §3: $\rho_\star=0.94$, $\lambda=0.20$; $K$ auto via persistence; $A$ row-stoch. (nzr=4), $B$ col-stoch. (nzc=12); pseudocounts+$\gamma=3\times10^{-3}$; robust PH solve (Tikhonov+LSQR). Reported are mean ± s.d. over 45 runs for (i) NRMSE at horizon $H=600$, (ii) VPT, and (iii) ADev.

outperforms all baselines at every horizon, often by a wide margin: e.g., on *MIT–BIH* at $H=1000$, PHR reaches $0.5833 \pm 0.0123$ NRMSE versus the strongest baseline (MCI, $1.1273 \pm 0.0156$), and on *Sunspot* at $H=1000$ PHR attains $0.2044 \pm 0.0010$ versus MCI at $0.4064 \pm 0.0100$ (cf. Figs. 5, 7). The advantage also holds on *BIDMC* (e.g., $H=600$: $0.3571 \pm 0.0151$ vs. best baseline $0.4305 \pm 0.0269$) and *Santa Fe* (e.g., $H=600$: $0.1262 \pm 0.0003$ vs. $0.2366 \pm 0.0014$). In chaotic closed-loop mode, PHR remains consistently best in NRMSE across horizons for *Lorenz* and *Rössler*, while also achieving the strongest VPT (e.g., *Lorenz*: $10.94 \pm 1.65$ vs. best baseline $9.37 \pm 1.83$; *Rössler*: $11.33 \pm 4.66$ vs. $9.88 \pm 4.59$), indicating longer time-to-divergence. ADev is compet-

itive: PHR is second-best on *Lorenz* (29.11 ± 9.53) with the lowest NRMSE and top VPT, and second-best on *Rössler* (13.11 ± 8.02) while again leading in NRMSE and VPT. On *Chen*, PHR provides uniformly lowest NRMSE across horizons (e.g., $H$=1000: 1.2316 ± 0.0737 vs. best baseline 1.2355 ± 0.0897) and highest VPT (4.18 ± 0.93), at the expense of slightly higher ADev than the very best baseline (PHR 55.42 ± 15.53 vs. CRJ 51.24 ± 10.03), reflecting a favorable accuracy–stability trade-off for long closed-loop rollouts (cf. Figs. 3, 6). Together, these results indicate that the two-channel construction—topological oscillators from $H^1$ cohomology (§3.1) and lifted Markov transport (§3.2)—translates into consistent accuracy and extended predictability windows across heterogeneous regimes, while the spectral scaling (§3.3) standardizes stability.

The ablation study in Table 2 probes each design choice. Disabling the persistence-based *auto-tune* of blend weights (§3.1, Eq. (4)) significantly degrades all metrics (NRMSE@600: 0.238 ± 0.191 vs. 0.212 ± 0.188; VPT: 8.90 ± 1.62 vs. 10.94 ± 1.65; ADev: 33.90 ± 11.92 vs. 29.11 ± 9.53), showing that weighting $W_{\mathrm{top}}$ by *measured* loop strength matters. Replacing PH-derived angles by the PCA phase surrogate (no PH) further hurts (NRMSE@600: 0.258 ± 0.195), as does removing either channel: *flow-only* (0.279 ± 0.198) and *topology-only* (0.341 ± 0.235) both underperform the full model, confirming that oscillatory clocks and directed transport contribute complementary signal. Implementation choices that stabilize coarse dynamics—near-death $\varepsilon$ for the Rips 1-skeleton, teleportation in $P$, and pseudocounts in $C_{ij}$—each improve robustness and accuracy relative to their removal (e.g., *no pseudocounts*: 0.255 ± 0.197). The robust PH solver (per-component gauge, Tikhonov, LSQR fallback) also helps (*no robust solver*: 0.247 ± 0.196), and capping to a single loop ($K_{\max}$=1) degrades all metrics, indicating the value of multiple incommensurate oscillators when present. Finally, omitting the spectral scaling is *unsafe*: 37% of runs diverge, empirically validating the necessity of the global norm control for closed-loop stability established in §3.3. *Additional quantitative breakdowns, phase-portrait overlays, error-growth visualisations, and hyperparameter details are presented in Appendix B.*

## 5   CONCLUSION AND OUTLOOK

We presented PHR, a reservoir-computing framework that *learns* the recurrent operator of a leaky ESN once, offline, from a single trajectory. The core idea is to replace random reservoirs by a principled blend of two analyzable components learned directly from data: (i) a topological rotation operator whose $2 \times 2$ blocks internalize long-lived loops via persistent cohomology and circular coordinates, and (ii) a lifted Markov operator that encodes short-horizon transport through empirical coarse transitions. A simple power-iteration rescaling of the blended operator enforces a uniform contraction bound for the leaky update, yielding a clean echo-state certificate. The resulting reservoir is interpretable (explicit oscillators with data-driven frequencies), task-aligned (local directionality imprinted by coarse flow), and efficient (only a ridge readout is trained). Empirically, this design preserves ESN-level training cost while providing a data-informed alternative to random reservoirs.

**Limitations & Future directions.** Our guarantees hinge on an operator-norm contraction of the *blended* reservoir; they do not provide channel-wise spectral bounds, and the ESP certificate is sufficient but not necessary. The Markov discretization introduces bias from partitioning and horizon choice; similarly, angular-velocity estimates inherit noise from finite sampling, subsampling stride, and the choice of working scale within a persistence interval. The current construction samples the supports of the pool–lift maps uniformly; more geometry-aware couplings may better preserve local structure. Several avenues are promising. Theoretically, combine stability of persistent cohomology/circular coordinates (under noise and subsampling) with Ulam-type transfer-operator error to obtain finite-sample rates for $\widehat{\omega}_k$, the power-scaled blend $W$, and forecasting risk; tighten ESP certificates to reflect $\tanh$, leakage, and sparsity. Algorithmically, design geometry-aware pool–lift maps (kNN/diffusion distances), multi-horizon mixtures $P^{(h)}$ for scale-separated transport, and streaming PH for nonstationarity; replace Euclidean delays with diffusion-map/manifold embeddings when measurements are anisotropic. For scaling, use landmark/witness complexes, approximate kNN graphs, and GPU-accelerated Laplacian/LS solvers to handle very long sequences. Applications include closed-loop control/data assimilation, anomaly detection via oscillator coherence, and domains needing interpretable oscillatory modes (climate, neuroscience, molecular kinetics).

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

# A APPENDIX

## A.1 PROOFS

We use the induced $\ell_\infty$ and $\ell_1$ matrix norms, $\|M\|_\infty = \max_i \sum_j |m_{ij}|$ and $\|M\|_1 = \max_j \sum_i |m_{ij}|$, together with submultiplicativity and the triangle inequality. Note that $\|P\|_\infty = 1$ by row–stochasticity and $\|B\|_1 = 1$ by column–stochasticity.

**Lemma A.1** (Oscillator spectrum under block-permuted rotations; see (Horn & Johnson, 2013; Trefethen & III, 1997)). $W_{\text{top}}$ *is a normal matrix. Its spectrum consists of the multiset*

$$\left\{ \rho_{\text{rot}}\, e^{+i\omega_k},\ \rho_{\text{rot}}\, e^{-i\omega_k} \right\}_{k=1}^{K} \ \cup\ \{r_j\}_{j=1}^{N-2K}, \qquad r_j \in (\rho_{\min}, \rho_{\max}), \tag{8}$$

*and it admits an orthogonal decomposition $\mathbb{R}^N = \bigoplus_{k=1}^{K} E_k \oplus E_\perp$ into $K$ invariant two-dimensional subspaces $E_k$ (one per rotation block) and an invariant $(N-2K)$-dimensional subspace $E_\perp$ for the decay modes. In particular, for each $k$ there exists an orthonormal basis of $E_k$ in which $W_{\text{top}}$ acts as the $2 \times 2$ rotation $\rho_{\text{rot}} R(\omega_k)$, and this property is unchanged by the subsequent orthogonal permutation used to distribute coordinates in the implementation.*

**Lemma A.2** (General pooled coarse–flow bound). *For any $r \in \mathbb{R}^Q$ and $x := Br \in \mathbb{R}^N$,*

$$\left\| A W_{\text{flow}}\, x - P r \right\|_\infty \ \le\ \left(1 + \|AB\|_\infty\right) \|AB - I_Q\|_\infty\, \|r\|_\infty. \tag{9}$$

*An analogous inequality holds in $\ell_1$:*

$$\left\| A W_{\text{flow}}\, x - P r \right\|_1 \ \le\ \|P\|_1 \left(1 + \|AB\|_1\right) \|AB - I_Q\|_1\, \|r\|_1. \tag{10}$$

*Proof.* Write

$$A W_{\text{flow}}\, x - P r\ =\ A B P A B r - P r\ =\ A B P (\underbrace{ABr - r}_{\Delta r}) + (\underbrace{AB - I_Q}_{\Delta}) P r\ =\ ABP\, \Delta r + \Delta P r.$$

Taking $\ell_\infty$ norms and using submultiplicativity,

$$\|ABP\, \Delta r\|_\infty \le \|AB\|_\infty \|P\|_\infty \|\Delta\|_\infty \|r\|_\infty = \|AB\|_\infty \|\Delta\|_\infty \|r\|_\infty,$$

and $\|\Delta P r\|_\infty \le \|\Delta\|_\infty \|P\|_\infty \|r\|_\infty = \|\Delta\|_\infty \|r\|_\infty$. Adding the two bounds yields (9). The $\ell_1$ case (10) is identical with $\|\cdot\|_1$ and $\|P\|_1$ in place of $\|\cdot\|_\infty$ and $\|P\|_\infty$. $\square$

**Corollary A.3** (Factor-2 bound under nonexpansive pooling). *If, in addition, $\|AB\|_\infty \le 1$ (e.g., when $A$ is row–stochastic and $B$ is also row–substochastic[2]), then for all $r \in \mathbb{R}^Q$ and $x = Br$,*

$$\left\| A W_{\text{flow}}\, x - P r \right\|_\infty \ \le\ 2\, \|AB - I_Q\|_\infty\, \|r\|_\infty.$$

*An identical factor-2 version holds in $\ell_1$ whenever $\|P\|_1 \le 1$ and $\|AB\|_1 \le 1$ (e.g., for doubly–stochastic $P$ and row–substochastic $AB$).*

*Remark* A.4. The quantity $\Delta = AB - I_Q$ is the *pool–lift calibration defect*. Lemma 3.1 shows that after pooling, the one–step action of the lifted operator $W_{\text{flow}} = BPA$ on any lifted coarse vector $Br$ is within $\left(1 + \|AB\|_\star\right)\|\Delta\|_\star \|r\|_\star$ of the ideal coarse Markov step $Pr$ (for $\star \in \{1, \infty\}$). Thus, when $AB \approx I_Q$ (good calibration), the high–dimensional pathway "pool → Markov → lift" is *coarsely faithful*. The factor $(1 + \|AB\|_\star)$ can be fixed at 2 by ensuring nonexpansivity of $AB$ in the chosen norm (e.g., by making $B$ row–substochastic).

**Connection to Proposition 3.2.** Part (ii) of Proposition 3.2 bounds the deviation of $AWB$ from a scaled coarse Markov step on $\mathbb{R}^Q$. Lemma 3.1 isolates the *pure flow channel* ($\alpha = \xi = 0$), showing that $\|AW_{\text{flow}}B - P\|_\star$ is controlled by $\|AB - I_Q\|_\star$ and nonexpansivity constants. The full bound in Proposition 3.2(ii) then adds leakage from the topological and noise channels, each weighted by the blend coefficients and their operator norms.

---

[2]If $B$ has nonnegative entries, column sums $= 1$, and row sums $\le 1$, then for any row–stochastic $A$ one has $\|AB\|_\infty = \max_q \sum_j (AB)_{qj} = \max_q \sum_i A_{qi} \sum_j B_{ij} \le \max_i \sum_j B_{ij} \le 1$.

*Proof of Proposition 3.2.* We recall the notation

$$W_{\text{blend}} = \alpha\,W_{\text{top}} + \beta\,W_{\text{flow}} + \xi\,W_{\text{noise}}, \qquad W = \frac{\rho_\star}{s}\,W_{\text{blend}}, \quad s := \|W_{\text{blend}}\|_2,$$

with $W_{\text{flow}} = B\,P^{(\gamma)}\,A$ and $\alpha, \beta, \xi \geq 0$, $\alpha + \beta + \xi = 1$. By Lemma A.1, $W_{\text{top}}$ is normal and its spectrum is $\{\rho_{\text{rot}}e^{\pm i\omega_k}\}_{k=1}^K \cup \{r_j\}_{j=1}^{N-2K}$ (with $r_j \in (\rho_{\min}, \rho_{\max})$), and is unchanged by the subsequent coordinate permutation.

*(i) Persistence of oscillatory eigenpairs.* If $s = 0$ then $W = 0$ and the claim is trivial. Assume $s > 0$. Write

$$W = \underbrace{\frac{\rho_\star}{s}\,\alpha\,W_{\text{top}}}_{=:A_0} + \underbrace{\frac{\rho_\star}{s}\big(\beta\,W_{\text{flow}} + \xi\,W_{\text{noise}}\big)}_{=:E_0}.$$

Since $W_{\text{top}}$ is normal, so is $A_0$, and we may invoke the standard spectral variation bound for normal matrices (see, e.g., (Trefethen & III, 1997, Ch. 2)): for every eigenvalue $\lambda \in \sigma(A_0)$ there exists an eigenvalue $\mu \in \sigma(A_0 + E_0)$ such that $|\mu - \lambda| \leq \|E_0\|_2$. Fix $k \in \{1, \ldots, K\}$ and take $\lambda_k^0 = (\rho_\star \alpha/s)\,\rho_{\text{rot}}e^{i\omega_k} \in \sigma(A_0)$. Then there exists $\lambda_k(W) \in \sigma(W)$ with

$$\left|\lambda_k(W) - \frac{\rho_\star \alpha}{s}\,\rho_{\text{rot}}e^{i\omega_k}\right| \leq \|E_0\|_2 = \frac{\rho_\star}{s}\big(\beta\,\|W_{\text{flow}}\|_2 + \xi\,\|W_{\text{noise}}\|_2\big),$$

and the same statement holds for the conjugate $(\rho_\star \alpha/s)\,\rho_{\text{rot}}e^{-i\omega_k}$. This proves the displayed inequality.

For the "in particular" clause, note that by the reverse triangle inequality,

$$s = \|W_{\text{blend}}\|_2 \geq \alpha\|W_{\text{top}}\|_2 - \big(\beta\|W_{\text{flow}}\|_2 + \xi\|W_{\text{noise}}\|_2\big),$$

so the stated lower bound on $s$ is always valid. Hence each oscillatory eigenvalue of $W$ lies in the closed disc centered at $\frac{\rho_\star \alpha}{s}\,\rho_{\text{rot}}e^{\pm i\omega_k}$ with radius $\frac{\rho_\star}{s}\big(\beta\|W_{\text{flow}}\|_2 + \xi\|W_{\text{noise}}\|_2\big)$, as claimed.

*(ii) Coarse–flow fidelity on the lifted subspace.* Let $r \in \mathbb{R}^Q$ and $x := Br \in \mathbb{R}^N$. Using $W = (\rho_\star/s)(\alpha W_{\text{top}} + \beta W_{\text{flow}} + \xi W_{\text{noise}})$ and $W_{\text{flow}} = B\,P^{(\gamma)}\,A$,

$$A\,W\,x = \frac{\rho_\star}{s}\Big[\alpha\,A\,W_{\text{top}}\,B\,r + \beta\,A\,W_{\text{flow}}\,B\,r + \xi\,A\,W_{\text{noise}}\,B\,r\Big]$$

$$= \frac{\rho_\star}{s}\Big[\alpha\,A\,W_{\text{top}}\,B\,r + \beta\,A\,B\,P^{(\gamma)}\,A\,B\,r + \xi\,A\,W_{\text{noise}}\,B\,r\Big].$$

Subtract $(\rho_\star \beta/s)\,P^{(\gamma)}r$ and regroup:

$$A\,W\,x - \frac{\rho_\star\beta}{s}P^{(\gamma)}r = \frac{\rho_\star}{s}\Big[\beta\big((AB)\,P^{(\gamma)}\,(AB) - P^{(\gamma)}\big)r + \alpha\,A\,W_{\text{top}}\,B\,r + \xi\,A\,W_{\text{noise}}\,B\,r\Big].$$

Taking operator 2-norms and using submultiplicativity,

$$\left\|A\,W\,x - \frac{\rho_\star\beta}{s}P^{(\gamma)}r\right\|_2 \leq \frac{\rho_\star}{s}\Big[\beta\,\big\|(AB)\,P^{(\gamma)}\,(AB) - P^{(\gamma)}\big\|_2\,\|r\|_2$$

$$+ \alpha\,\|A\|_2\,\|W_{\text{top}}\|_2\,\|B\|_2\,\|r\|_2 + \xi\,\|A\|_2\,\|W_{\text{noise}}\|_2\,\|B\|_2\,\|r\|_2\Big],$$

which is exactly the stated bound. In particular, if $AB = I_Q$ the first term vanishes, and for small $\alpha, \xi$ the leakage terms are correspondingly small, so that $AWB$ is a small-norm perturbation of $(\rho_\star\beta/s)P^{(\gamma)}$ on $\mathbb{R}^Q$.

$\square$

## A.2 INTUITIVE SUMMARY OF THE METHODOLOGY

This subsection distills the construction into a small number of composable primitives and explains *why* each step is present, referencing the concrete procedures in Algorithms 1–8. The design principle is: *learn a fixed, analyzable recurrent operator that carries (i) global oscillations of the attractor and (ii) local short-horizon transport, then scale it once for stability and only train a linear readout.*

**1) Build the geometry we will learn from (Alg. 1).** We convert the observed time series $\{u_t\}$ into a delay-embedded cloud $Z = \{z_t\} \subset \mathbb{R}^{md_{\mathrm{obs}}}$ to expose the attractor's geometry. To keep topological computations tractable, we thin $Z$ by a stride $s$ to a PH–subset $Z^{(\mathrm{PH})}$ and compute its pairwise distance matrix $D$. This decouples the cost of homological inference from the ultimate reservoir size $N$ and the length of the raw series: the topology is read from the *subsample*, while the recurrent operator will still act in the full neuron space.

---

**Algorithm 1:** DELAYEMBED&SUBSAMPLEFORPH

---

**Input:** Observed trajectory $\{u_t\}_{t=1}^T \subset \mathbb{R}^{d_{\mathrm{obs}}}$, embedding dimension $m \in \mathbb{N}$, lag $\tau \in \mathbb{N}$, PH stride $s \in \mathbb{N}$.

**Output:** Embedded sequence $Z = \{z_t\}_{t=(m-1)\tau}^T \subset \mathbb{R}^{md_{\mathrm{obs}}}$; PH subsample $Z^{(\mathrm{PH})} = \{z_{t_0+js}\}_{j=0}^{n_{\mathrm{PH}}-1}$; distance matrix $D \in \mathbb{R}^{n_{\mathrm{PH}} \times n_{\mathrm{PH}}}$.

**1 for** $t = (m-1)\tau, \ldots, T$ **do**
**2** $\quad \lfloor z_t \leftarrow [\, u_t^\top, u_{t-\tau}^\top, \ldots, u_{t-(m-1)\tau}^\top \,]^\top$

**3** Pick an anchor $t_0 \in \{(m-1)\tau, \ldots, T\}$ and form $Z^{(\mathrm{PH})} \leftarrow \{z_{t_0}, z_{t_0+s}, z_{t_0+2s}, \ldots\}$ within index range
**4** Compute $D_{ij} \leftarrow \|z_{t_0+is} - z_{t_0+js}\|_2$ for all $i, j$ (symmetric, zero diagonal)
**5 return** $(Z, Z^{(\mathrm{PH})}, D)$
$\quad \triangleright$ `Complexity: building` $Z$ `is` $O(Tmd_{\mathrm{obs}})$`;` $D$ `is` $O(n_{\mathrm{PH}}^2 md_{\mathrm{obs}})$`.`

---

**Algorithm 2:** PHCIRCULARCOORDINATES&ANGULARVELOCITIES

---

**Input:** $(Z^{(\mathrm{PH})}, D)$ from Alg. 1; field $\mathbb{F}_p$ (prime $p$);
selection cap $K_{\max}$; scale rule $\varepsilon_\ell \in (b_\ell, d_\ell)$ (e.g., near-death).
**Output:** Selected indices $\mathcal{I}_{\mathrm{keep}} \subseteq \{1, \ldots\}$; angles $\{\theta^{(\ell)} \in (-\pi, \pi]^{n_{\mathrm{PH}}}\}_{\ell \in \mathcal{I}_{\mathrm{keep}}}$; angular velocities $\{\widehat{\omega}_\ell\}_{\ell \in \mathcal{I}_{\mathrm{keep}}}$; persistences $\{P_\ell\}_{\ell \in \mathcal{I}_{\mathrm{keep}}}$.

**1** Compute Vietoris–Rips persistent *cohomology* up to degree 1 on $(Z^{(\mathrm{PH})}, D)$ over $\mathbb{F}_p$;
**2** Obtain $H^1$ intervals $\{(b_\ell, d_\ell)\}_\ell$ and representative 1-cocycles $\{c_\ell\}_\ell$
**3** Let $P_\ell \leftarrow d_\ell - b_\ell$ and order classes by $P_\ell$ (desc.)
**4** Select the top $K_{\max}$ indices as candidates $\mathcal{I}_{\mathrm{cand}}$
**5 for** $\ell \in \mathcal{I}_{\mathrm{cand}}$ **do**
**6** $\quad$ Set $\varepsilon_\ell \in (b_\ell, d_\ell)$ (default: $d_\ell - 10^{-6}$)
**7** $\quad$ Build the Rips 1–skeleton $G_\ell = (V, E_\ell)$ at threshold $\varepsilon_\ell$
**8** $\quad$ Lift $c_\ell$ to $\alpha_{ij}^{(\ell)} \in (-\frac{1}{2}, \frac{1}{2}]$ on edges $(i, j) \in E_\ell$
**9** $\quad$ Solve $(L + \mu I)\vartheta^{(\ell)} = b$ with $L = M^\top W M$, $b = M^\top W \alpha$, per connected component (gauge anchor, tiny $\mu > 0$)
**10** $\quad \theta^{(\ell)} \leftarrow \mathrm{wrap}_{(-\pi, \pi]}(2\pi \vartheta^{(\ell)})$
**11** $\quad \widehat{\omega}_\ell \leftarrow \mathrm{mean}\big(\mathrm{wrap}_{(-\pi, \pi]}(\theta_{t+1}^{(\ell)} - \theta_t^{(\ell)})\big)$
**12** Apply a relative-persistence threshold $\gamma \in [0, 1]$: $\mathcal{I}_{\mathrm{keep}} \leftarrow \{\ell \in \mathcal{I}_{\mathrm{cand}} : P_\ell \geq \gamma \max_{\ell'} P_{\ell'}\}$
**13 return** $(\mathcal{I}_{\mathrm{keep}}, \{\theta^{(\ell)}, \widehat{\omega}_\ell, P_\ell\}_{\ell \in \mathcal{I}_{\mathrm{keep}}})$
$\quad \triangleright$ `Guarantee: angles are well-defined up to global phase per component.`

---

**2) Extract global oscillators from topology (Alg. 2 and 3).** Persistent cohomology on $(Z^{(\mathrm{PH})}, D)$ detects long-lived 1-cycles (bars $(b_\ell, d_\ell)$) and provides representative cocycles. A circular-coordinate solver turns each selected bar into an angle $\theta^{(\ell)}$ on the sample; the wrapped increment of this angle estimates a mean angular velocity $\widehat{\omega}_\ell$. We then synthesize a *normal* block–diagonal operator $W_{\mathrm{top}}$ whose $2 \times 2$ rotation blocks $\rho_{\mathrm{rot}} R(\widehat{\omega}_\ell)$ instantiate those angular velocities as stable internal oscillators and whose remaining diagonal entries are decays. A random permutation spreads the oscillator pairs across coordinates so they can interact with other channels. The effect is to *hard-wire* data-driven phases into $W$ without backpropagation.

---

**Algorithm 3:** SYNTHESIZETOPOLOGICALOPERATOR

**Input:** Angular velocities $\{\widehat{\omega}_\ell\}_{\ell=1}^{K_{\text{final}}}$; rotation radius $\rho_{\text{rot}} \in (0,1)$; decay radii interval
$(\rho_{\min}, \rho_{\max})$; reservoir size $N \geq 2K_{\text{final}}$.
**Output:** $W_{\text{top}} \in \mathbb{R}^{N \times N}$ (block-permuted rotation–decay).

**1** Form $K_{\text{final}}$ many $2 \times 2$ rotation blocks $R_\ell = \rho_{\text{rot}} \begin{bmatrix} \cos\widehat{\omega}_\ell & -\sin\widehat{\omega}_\ell \\ \sin\widehat{\omega}_\ell & \cos\widehat{\omega}_\ell \end{bmatrix}$
**2** Draw $(N - 2K_{\text{final}})$ radii $r_j \sim \text{Unif}(\rho_{\min}, \rho_{\max})$ and set $D_\perp \leftarrow \text{diag}(r_1, \ldots, r_{N-2K_{\text{final}}})$
**3** Assemble $W_b \leftarrow \text{blkdiag}(R_1, \ldots, R_{K_{\text{final}}}, D_\perp)$
**4** Apply a random permutation $P$ to distribute blocks: $W_{\text{top}} \leftarrow P^\top W_b P$
**5 return** $W_{\text{top}}$
   ▷ Spectrum: $\{\rho_{\text{rot}}e^{\pm i\widehat{\omega}_\ell}\} \cup \{r_j\}$; $W_{\text{top}}$ is normal up to permutation.

---

**Algorithm 4:** COARSEFLOW&LIFTEDOPERATOR

**Input:** Embedded cloud $Z = \{z_t\}$; #clusters $Q$; horizon $h \in \mathbb{N}$; pseudocount $\varepsilon > 0$; teleport
$\gamma \in [0,1)$; pooling/lifting sparsities $(\text{nzr}, \text{nzc})$; reservoir size $N$.
**Output:** Centroids $\{c_q\}$; assignments $\{s_t\}$; $P^{(\gamma)}$; $A \in \mathbb{R}^{Q \times N}$ (row–stoch.), $B \in \mathbb{R}^{N \times Q}$
(col–stoch.); $W_{\text{flow}} = BP^{(\gamma)}A$.

**1** Run $k$-means (Lloyd) on $Z$ to get $\{c_q\}_{q=1}^Q$ and $s_t = \arg\min_q \|z_t - c_q\|_2$
**2** Build counts $C_{ij} \leftarrow \#\{t: s_t = i, s_{t+h} = j\}$
**3** Row-normalize with $\varepsilon$: $P_{ij} = \dfrac{C_{ij} + \varepsilon}{\sum_{j'}(C_{ij'} + \varepsilon)}$
**4** Teleport: $P^{(\gamma)} \leftarrow (1 - \gamma)P + \gamma \mathbf{1}u^\top$, $u = \frac{1}{Q}\mathbf{1}$
**5** Construct $A$: for each row $q$, choose $\text{nzr}$ columns, set $A_{q,\cdot}$ equal weights summing to 1
**6** Construct $B$: for each column $q$, choose $\text{nzc}$ rows, set $B_{\cdot, q}$ equal weights summing to 1
**7** Set $W_{\text{flow}} \leftarrow BP^{(\gamma)}A$
**8 return** $(\{c_q\}, \{s_t\}, P^{(\gamma)}, A, B, W_{\text{flow}})$
   ▷ Note: $\|A\|_\infty = \|P^{(\gamma)}\|_\infty = 1$, $\|B\|_1 = 1$.

---

**Algorithm 5:** AUTOTUNEBLENDWEIGHTS

**Input:** Persistences $\{P_\ell\}_{\ell \in \mathcal{I}_{\text{keep}}}$ (possibly empty); noise fraction $\xi \in [0,1)$; bounds
$0 \leq \alpha_{\min} \leq \alpha_{\max} \leq 1$; clip operator $\text{clip}(\cdot; 0, 1 - \xi)$.
**Output:** $(\alpha_{\text{top}}, \beta_{\text{flow}})$ with $\alpha_{\text{top}}, \beta_{\text{flow}} \geq 0$ and $\alpha_{\text{top}} + \beta_{\text{flow}} \leq 1 - \xi$.

**1 if** $\mathcal{I}_{\text{keep}} = \varnothing$ **then**
**2**     $\alpha_{\text{top}} \leftarrow 0$, $\beta_{\text{flow}} \leftarrow 1 - \xi$
**3**     **return**

**4** $P_{\max} \leftarrow \max_{\ell \in \mathcal{I}_{\text{keep}}} P_\ell$, $s \leftarrow \dfrac{1}{|\mathcal{I}_{\text{keep}}|} \sum_{\ell \in \mathcal{I}_{\text{keep}}} \dfrac{P_\ell}{P_{\max}}$
**5** $\alpha_{\text{top}} \leftarrow \text{clip}(\alpha_{\min} + (\alpha_{\max} - \alpha_{\min})s; 0, 1 - \xi)$
**6** $\beta_{\text{flow}} \leftarrow 1 - \xi - \alpha_{\text{top}}$
**7 return** $(\alpha_{\text{top}}, \beta_{\text{flow}})$
   ▷ If PH fails or is weak ($\mathcal{I}_{\text{keep}} = \varnothing$), the scheme becomes
     flow-dominant.

---

**3) Encode local directed transport (Alg. 4).** We discretize $Z$ by $k$-means into $Q$ coarse cells and count short-horizon transitions to obtain a row-stochastic Markov matrix $P$, regularized by pseudocounts and teleportation. Two stochastic maps connect coarse states to neurons: $A$ (row-stochastic) pools neuron activity to the coarse scale; $B$ (column-stochastic) lifts coarse mass back to neurons. The composition $W_{\text{flow}} := B P^{(\gamma)} A$ therefore advances a neuron state by *pool → Markov step → lift*. Lemma 3.1 quantifies the fidelity of this mechanism: after pooling, the discrepancy from

**Algorithm 6:** BLENDANDSCALETOTARGET

**Input:** $W_{\text{top}}$, $W_{\text{flow}}$, noise $W_{\text{noise}}$ with $\|W_{\text{noise}}\|_2 = 1$; weights $(\alpha_{\text{top}}, \beta_{\text{flow}}, \xi)$,
$\quad\quad \alpha_{\text{top}} + \beta_{\text{flow}} + \xi = 1$; target $\rho_\star \in (0, 1)$.
**Output:** Scaled recurrent $W$ with $\|W\|_2 = \rho_\star$.

**1** $W_{\text{blend}} \leftarrow \alpha_{\text{top}} W_{\text{top}} + \beta_{\text{flow}} W_{\text{flow}} + \xi W_{\text{noise}}$
**2** Estimate $s \approx \|W_{\text{blend}}\|_2$ via power iteration on $W_{\text{blend}}^\top W_{\text{blend}}$ (fixed iters, NaN guards)
**3 if** $s = 0$ *or estimate invalid* **then**
**4** $\quad$| $\quad$**return** $W \leftarrow 0$ (trivial contraction)
**5** $W \leftarrow (\rho_\star/s) W_{\text{blend}}$
**6 return** $W$
$\quad \triangleright$ `Guarantee:` $\|W\|_2 = \rho_\star$`; leaky ESN with leak` $\lambda$ `has contraction`
$\quad\quad$ `factor` $(1 - \lambda) + \lambda \rho_\star < 1$`.`

---

**Algorithm 7:** PHR_TRAININGPIPELINE

**Input:** Trajectory $\{u_t\}_{t=1}^T$; $(m, \tau, s)$; PH/selection $(p, K_{\max}, \gamma)$; synthesis $(\rho_{\text{rot}}, \rho_{\min}, \rho_{\max})$;
$\quad\quad$ flow $(Q, h, \varepsilon, \gamma_{\text{tel}}, \texttt{nzr}, \texttt{nzc})$; blend auto-bounds $(\alpha_{\min}, \alpha_{\max}, \xi)$; scaling $\rho_\star$; leak $\lambda$;
$\quad\quad$ polynomial feature flag $\mathbf{1}_{\text{poly}}$.
**Output:** Fixed recurrent $W$; fitted readout $W_{\text{out}}$.

**1** $(Z, Z^{(\text{PH})}, D) \leftarrow$ DELAYEMBED&SUBSAMPLEFORPH$(\{u_t\}, m, \tau, s)$
**2** $(\mathcal{I}_{\text{keep}}, \{\theta^{(\ell)}, \widehat{\omega}_\ell, P_\ell\}) \leftarrow$ PHCIRCULARCOORDINATES&ANGULARVELOCITIES$(Z^{(\text{PH})}, D, p, K_{\max})$

**3** $W_{\text{top}} \leftarrow$ SYNTHESIZETOPOLOGICALOPERATOR$(\{\widehat{\omega}_\ell\}_{\ell \in \mathcal{I}_{\text{keep}}}, \rho_{\text{rot}}, (\rho_{\min}, \rho_{\max}), N)$
**4** $(\{c_q\}, \{s_t\}, P^{(\gamma)}, A, B, W_{\text{flow}}) \leftarrow$ COARSEFLOW&LIFTEDOPERATOR$(Z, Q, h, \varepsilon, \gamma_{\text{tel}}, \texttt{nzr}, \texttt{nzc}, N)$

**5** $(\alpha_{\text{top}}, \beta_{\text{flow}}) \leftarrow$ AUTOTUNEBLENDWEIGHTS$(\{P_\ell\}_{\ell \in \mathcal{I}_{\text{keep}}}, \xi, \alpha_{\min}, \alpha_{\max})$
**6** $W \leftarrow$ BLENDANDSCALETOTARGET$(W_{\text{top}}, W_{\text{flow}}, W_{\text{noise}}, (\alpha_{\text{top}}, \beta_{\text{flow}}, \xi), \rho_\star)$

$\quad \triangleright$ `Teacher-forced reservoir rollout and ridge readout`
**7** Advance leaky ESN: $x_t = (1 - \lambda)x_{t-1} + \lambda \tanh(W x_{t-1} + W_{\text{in}} u_t)$; collect features
$\quad \varphi_t = [x_t;\ \mathbf{1}_{\text{poly}}(x_t \odot x_t);\ \mathbf{1}_{\text{poly}}]$
**8** Discard washout $D$, assemble $\Phi \in \mathbb{R}^{(T-D) \times F}$ and targets $Y \in \mathbb{R}^{(T-D) \times d_{\text{out}}}$
**9** Solve ridge: $W_{\text{out}}^\top = (\Phi^\top \Phi + \alpha I)^{-1} \Phi^\top Y$
**10 return** $(W, W_{\text{out}})$

---

**Algorithm 8:** PHR_INFERENCE

**Input:** Fixed $(W, W_{\text{in}}, W_{\text{out}})$; leak $\lambda$; initial input $u_0$; horizon $H$.
**Output:** Predicted outputs $\{\widehat{y}_t\}_{t=1}^H$.

**1** Initialize $x_0 \leftarrow 0$, $u \leftarrow u_0$
**2 for** $t = 1, \dots, H$ **do**
**3** $\quad$| $\quad x_t \leftarrow (1 - \lambda)x_{t-1} + \lambda \tanh(W x_{t-1} + W_{\text{in}} u)$
**4** $\quad$| $\quad$ Form feature $\varphi_t$ as in training; $\widehat{y}_t \leftarrow W_{\text{out}} \varphi_t$
$\quad\quad \triangleright$ `Autoregressive option: feed back first` $d_{\text{obs}}$ `coordinates`
**5** $\quad$| $\quad u \leftarrow \widehat{y}_t[1{:}d_{\text{obs}}]$
**6 return** $\{\widehat{y}_t\}$

---

an "ideal" $P^{(\gamma)}$ step is controlled by the calibration defect $AB - I$ and nonexpansivity constants. Intuitively, $W_{\text{flow}}$ imprints the observed short-time arrows of motion onto the reservoir, but does so in a way that is linear, nonnegative, and analyzable.

**4) Decide how much topology vs. flow to keep (Alg. 5).** Not all datasets have strong loops. The auto-tuner reads the *evidence* from PH persistences $\{P_\ell\}$: it keeps the classes above a relative

threshold $\gamma$ and converts their average strength into a topological weight $\alpha_{\text{top}}$ within user bounds; the remainder of the budget (minus the small noise fraction $\xi$) is assigned to the flow channel, $\beta_{\text{flow}} = 1 - \xi - \alpha_{\text{top}}$. If no loop survives, the model becomes flow-dominant automatically ($\alpha_{\text{top}} = 0$). This makes the blend responsive: oscillatory problems (e.g., ECG, sunspots) allocate more mass to $W_{\text{top}}$, while broadband transport (e.g., laser) leans on $W_{\text{flow}}$.

**5) Blend once and enforce a global stability budget (Alg. 6).** We form $W_{\text{blend}} = \alpha_{\text{top}} W_{\text{top}} + \beta_{\text{flow}} W_{\text{flow}} + \xi W_{\text{noise}}$, where a tiny isotropic noise breaks algebraic degeneracies. A short power iteration estimates $\|W_{\text{blend}}\|_2$ and we scale to a *target* norm $\rho_\star \in (0, 1)$: $W = \frac{\rho_\star}{\hat{\sigma}_{\max}(W_{\text{blend}})} W_{\text{blend}}$. This single step detaches *structural fidelity* from *stability*: regardless of $A, B, P$ sparsity or the number of loops, the leaky ESN update has contraction factor $(1 - \lambda) + \lambda \rho_\star < 1$, hence ESP holds. Meanwhile, Proposition 3.2 guarantees that (i) the eigenpairs contributed by $W_{\text{top}}$ persist up to a perturbation governed by $\beta \|W_{\text{flow}}\|_2 + \xi \|W_{\text{noise}}\|_2$, and (ii) on lifted coarse states $x = Br$ the pooled action $AWx$ is close to a scaled $P^{(\gamma)} r$, with error budget split into the $AB$ calibration defect and the leakage from non-flow channels.

**6) Train (Alg. 7 and 8).** After $W$ is fixed, training reduces to a single convex ridge regression on features extracted from the leaky dynamics. Inference reuses $W$ and the readout, either teacher-forced or autoregressive. No BPTT is used or needed: all nonlinear recurrence is in a *scaled, fixed* operator.

**What one gains.**

- *Interpretability:* $W_{\text{top}}$ carries explicit oscillators with interpretable frequencies; $W_{\text{flow}}$ encodes data-driven coarse transport (Ulam/PageRank style). The blend weights are determined by PH evidence.

- *Stability by construction:* a single scalar $\rho_\star$ (with leak $\lambda$) enforces ESP independent of $A, B, P$ particulars.

- *Robust guarantees:* Lemma 3.1 and Proposition 3.2 provide end-to-end control of (coarse) transport fidelity and of the perturbation of oscillatory eigenpairs under blending and scaling.

- *Practicality:* PH is computed on a subsample (Alg. 1), making global structure affordable; all remaining steps are linear-algebraic and scale linearly or near-linearly in $N$.

**Complexity at a glance.** PH on $n_{\text{PH}}$ points is $O(n_{\text{PH}}^2)$ distances plus cocycle solves, kept small by stride $s$; $k$-means and count aggregation are $O(|Z|Q)$ per pass; constructing $W_{\text{top}}$ and $W_{\text{flow}}$ is linear in the number of nonzeros; power iteration uses a fixed budget of sparse/dense multiplies. Readout fitting is a single ridge solve on $(T - D) \times F$ features. Taken together, the pipeline learns $W$ *once* offline, after which training and inference have the standard ESN cost profile.

**Echo-state certificate.** Consider the leaky ESN update with $\phi = \tanh$ (1-Lipschitz) and leak $\lambda \in (0, 1]$:

$$x_t = (1 - \lambda) x_{t-1} + \lambda \phi(W x_{t-1} + W_{\text{in}} u_t). \tag{11}$$

With the scaling (5), $\|W\|_2 = \rho_\star$, hence the iteration is a contraction with factor $L = (1 - \lambda) + \lambda \rho_\star < 1$, which implies ESP (asymptotic independence of the initial state for each input) by standard arguments (Buehner & Young, 2006; Yildiz et al., 2012; Manjunath & Jaeger, 2013). This turns stability control into a single hyperparameter choice $\rho_\star \in (0, 1)$ (together with $\lambda$), independent of the particular sparsity structure of $A$ and $B$ or the empirical chain $P^{(\gamma)}$.

A.3 DYNAMICS OF PHR

We now describe at system level, how PHR evolves under input, and why its two-channel construction (topology $\oplus$ coarse flow), together with global scaling, yields a stable, interpretable, and data-faithful recurrent operator.

**State update and contraction budget.** The PHR evolves according to the leaky ESN recursion (Eqn. 11) with fixed recurrent matrix

$$W = \frac{\rho_\star}{s} W_{\text{blend}}, \qquad W_{\text{blend}} = \alpha W_{\text{top}} + \beta W_{\text{flow}} + \xi W_{\text{noise}}, \quad s := \|W_{\text{blend}}\|_2, \qquad (12)$$

as defined in (3)–(5). Since $\|\phi\|_{\text{Lip}} = 1$ and $\|W\|_2 = \rho_\star \in (0,1)$ by construction, the update map in (11) is a global contraction with constant

$$L = (1 - \lambda) + \lambda \rho_\star < 1, \qquad (13)$$

implying the echo-state property (ESP): for any fixed input sequence, dependence on the initial state decays geometrically at rate $L^t$ (Buehner & Young, 2006; Yildiz et al., 2012; Manjunath & Jaeger, 2013). The quantity $(1 - L)^{-1}$ gives the effective memory horizon of the reservoir. This single scalar budget $L$—chosen via $(\lambda, \rho_\star)$—decouples *stability* from the internal structural choices in $W_{\text{blend}}$.

**Spectral picture and oscillatory planes.** The topological channel $W_{\text{top}}$ is (permuted) block-diagonal with $K$ rotation blocks and $(N - 2K)$ scalar decays (Lemma A.1). Hence $\mathbb{R}^N$ admits an orthogonal decomposition

$$\mathbb{R}^N = \Big( \bigoplus_{k=1}^{K} E_k \Big) \oplus E_\perp, \quad \text{with} \quad W_{\text{top}}\big|_{E_k} = \rho_{\text{rot}} R(\omega_k), \quad W_{\text{top}}\big|_{E_\perp} = \text{diag}(r_j), \qquad (14)$$

where $R(\omega)$ is a $2 \times 2$ planar rotation and $r_j \in (\rho_{\min}, \rho_{\max}) \subset (0,1)$. Blending with the flow and noise channels, followed by the global similarity scaling $\rho_\star / s$, preserves these oscillatory eigenpairs up to a norm-controlled perturbation: part (i) of Proposition 3.2 states that each eigenvalue near $\alpha \rho_{\text{rot}} e^{\pm i \omega_k}$ is displaced by at most $\frac{\rho_\star}{s} \big( \beta \|W_{\text{flow}}\|_2 + \xi \|W_{\text{noise}}\|_2 \big)$, so the internal oscillators retain their (data-driven) angular frequencies $\{\omega_k\}$ to first order, with radii uniformly contracted to respect the ESP budget. This endows PHR with *phase-aware* latent dynamics tied to the long-lived $H_1$ loops extracted from the data (§3.1), while ensuring that no latent mode can violate the uniform contraction (13).

**Lifted coarse transport.** The flow channel $W_{\text{flow}} = B P^{(\gamma)} A$ implements "pool → Markov step → lift". Lemma 3.1 quantifies its fidelity at the coarse level: for any coarse vector $r$ and its lift $x = Br$,

$$\|A W_{\text{flow}} x - P^{(\gamma)} r\|_\star \lesssim \|AB - I\|_\star \|r\|_\star, \qquad \star \in \{1, \infty\}, \qquad (15)$$

with an explicit nonexpansivity factor (Corollary A.3). Thus the defect in reproducing one coarse Markov step depends *only* on the pool–lift calibration $AB \approx I_Q$; it is independent of $N$ and the detailed sparsity of $A$ and $B$. After blending and scaling, Proposition 3.2(ii) lifts this statement to the full recurrent operator:

$$\begin{aligned}
\Big\| A W B - \frac{\rho_\star \beta}{s} P^{(\gamma)} \Big\|_2 \le \frac{\rho_\star}{s} \Big[ &\beta \big\| (AB) P^{(\gamma)} (AB) - P^{(\gamma)} \big\|_2 \\
&+ \alpha \|A\|_2 \|W_{\text{top}}\|_2 \|B\|_2 + \xi \|A\|_2 \|W_{\text{noise}}\|_2 \|B\|_2 \Big].
\end{aligned} \qquad (16)$$

When $AB = I_Q$ (exact calibration) the first term vanishes and $AWB$ is a small-norm perturbation of a scaled Markov step, with perturbation budget apportioned by $(\alpha, \xi)$. Consequently, repeated application of $AWB$ advances pooled mass approximately along the empirical coarse transport $P^{(\gamma)}$, with errors that accumulate linearly in the number of steps and are globally bounded by the contraction scaling (standard matrix perturbation arguments; cf. (Trefethen & III, 1997, Ch. 2)).

**Two interacting channels under global scaling.** Equations (12) and (16) exhibit a clean separation:

- The *topological channel* contributes persistent, contractive oscillators aligned with the most prominent $H_1$ classes (Proposition 3.2(i)), thereby encoding global recurrent structure (e.g., lobe rotations in Lorenz-type systems, cardiac phase in ECG).

- The *flow channel* transports pooled mass coherently along short-horizon directions observed in the data (Ulam/PageRank view of coarse transfer), with fidelity governed by $\|AB - I\|$ and insulated by the uniform contraction (Proposition 3.2(ii)).

- The *noise channel* breaks degeneracies and complements the basis without compromising stability, as its contribution is explicitly budgeted by $\xi$ and then squashed by the global scaling to $\rho_\star$.

The auto-tuner (§3.1) sets $(\alpha, \beta)$ from persistence statistics, so that PHR naturally interpolates between a *phase-dominant* regime (clear loops, large $\alpha$) and a *flow-dominant* regime (weak or absent loops, large $\beta$), while keeping the ESP budget unchanged.

**Driven dynamics and input response.** Because (11) is globally contractive, it is input-to-state stable (ISS) in the standard ESN sense: for any two input sequences $\{u_t\}, \{\tilde{u}_t\}$ and trajectories $\{x_t\}, \{\tilde{x}_t\}$ driven from any initial states,

$$\|x_t - \tilde{x}_t\|_2 \leq L^t \|x_0 - \tilde{x}_0\|_2 + \lambda \|W_{\mathrm{in}}\|_2 \sum_{k=1}^{t} L^{t-k} \|u_k - \tilde{u}_k\|_2, \tag{17}$$

a routine consequence of the contraction mapping principle with a 1-Lipschitz nonlinearity (see, e.g., Buehner & Young (2006); Manjunath & Jaeger (2013)). Thus the state is a stable, causal functional of the input with fading memory on the timescale $O\big((1-L)^{-1}\big)$. Inside this ISS envelope, the *phase-aware* latent oscillators and the *coarse transport* induced by $P^{(\gamma)}$ shape the geometry of features seen by the readout: sinusoids and their polynomial interactions on the oscillatory planes, and coarse "advection" along observed short-time flow on the lifted subspace. The resulting feature set is expressive for forecasting and classification tasks (cf. EDMD/Koopman perspectives (Williams et al., 2015; Klus et al., 2016)) while retaining transparent control of stability and timescales via $(\lambda, \rho_\star)$.

**Timescales and design guidance.** Three timescales govern PHR: (i) the *contraction* time constant $\tau_{\mathrm{c}} \approx (1-L)^{-1}$, (ii) the *oscillation* periods $T_k = 2\pi/|\omega_k|$ on the planes $E_k$, and (iii) the *coarse mixing* time of $P^{(\gamma)}$. Choosing $\lambda$ and $\rho_\star$ fixes $\tau_{\mathrm{c}}$; the auto-selected $\{\omega_k\}$ and $\alpha$ fix how prominently these periods appear in the latent state; and $\beta$ sets the weight of coarse transport relative to phase. In practice, one targets $\tau_{\mathrm{c}}$ modestly larger than the shortest $T_k$ to preserve phase information while still ensuring fast forgetting of transients; $\gamma$ (teleport) is chosen small to regularize reducible chains without washing out observed directionality.

**Interpretability and robustness.** The topological modes are *interpretable* by construction (each $E_k$ corresponds to a persistent 1-cycle), and their frequencies come from the circular coordinates fitted on the data (§3.1); persistence-based selection and the near-death scale choice ensure robustness to metric noise (Edelsbrunner & Harer, 2010; de Silva & Vejdemo-Johansson, 2009). The flow channel is grounded in a classical Ulam discretization of the transfer operator with PageRank regularization (Dellnitz & Junge, 1999; Froyland, 2001; Brin & Page, 1998; Langville & Meyer, 2012). Both channels are finally *metered* by the explicit $\ell_2$ scaling to $\rho_\star$, which turns ESP from a heuristic into a verifiable certificate independent of $A$, $B$ sparsity or $P^{(\gamma)}$ reducibility.

Ergo, PHR dynamics are those of a *globally contractive*, *two-channel* reservoir: (a) a bank of data-driven, stable oscillators persisting under norm-bounded blending, and (b) a lifted coarse transport that, after pooling, closely tracks a Markov step learned from short-horizon transitions; both are coordinated by an explicit contraction budget and complemented by light noise to avoid degeneracies. The result is a fixed, analyzable recurrent core whose state features exhibit task-relevant structure with provable stability and quantitatively controlled deviations (Lemma 3.1, Proposition 3.2), after which training reduces to a single ridge solve for the readout.

## A.4 APPLICATIONS AND USABILITY ACROSS DOMAINS

PHR produces a *single, fixed* recurrent core $W$ that is (i) globally contractive (ESP with budget $L = (1-\lambda) + \lambda\rho_\star < 1$), (ii) spectrally structured by topological oscillators that persist under bounded blending (Proposition 3.2(i)), and (iii) coarse–transport faithful on pooled/lifted states up to a calibration defect (Lemma 3.1, Proposition 3.2(ii)). These properties make PHR broadly usable for *model–free learning* tasks where one wants a stable, reusable state–space embedding from a single exemplar trajectory. Concretely, once $W$ is learned offline, downstream forecasting, regression,

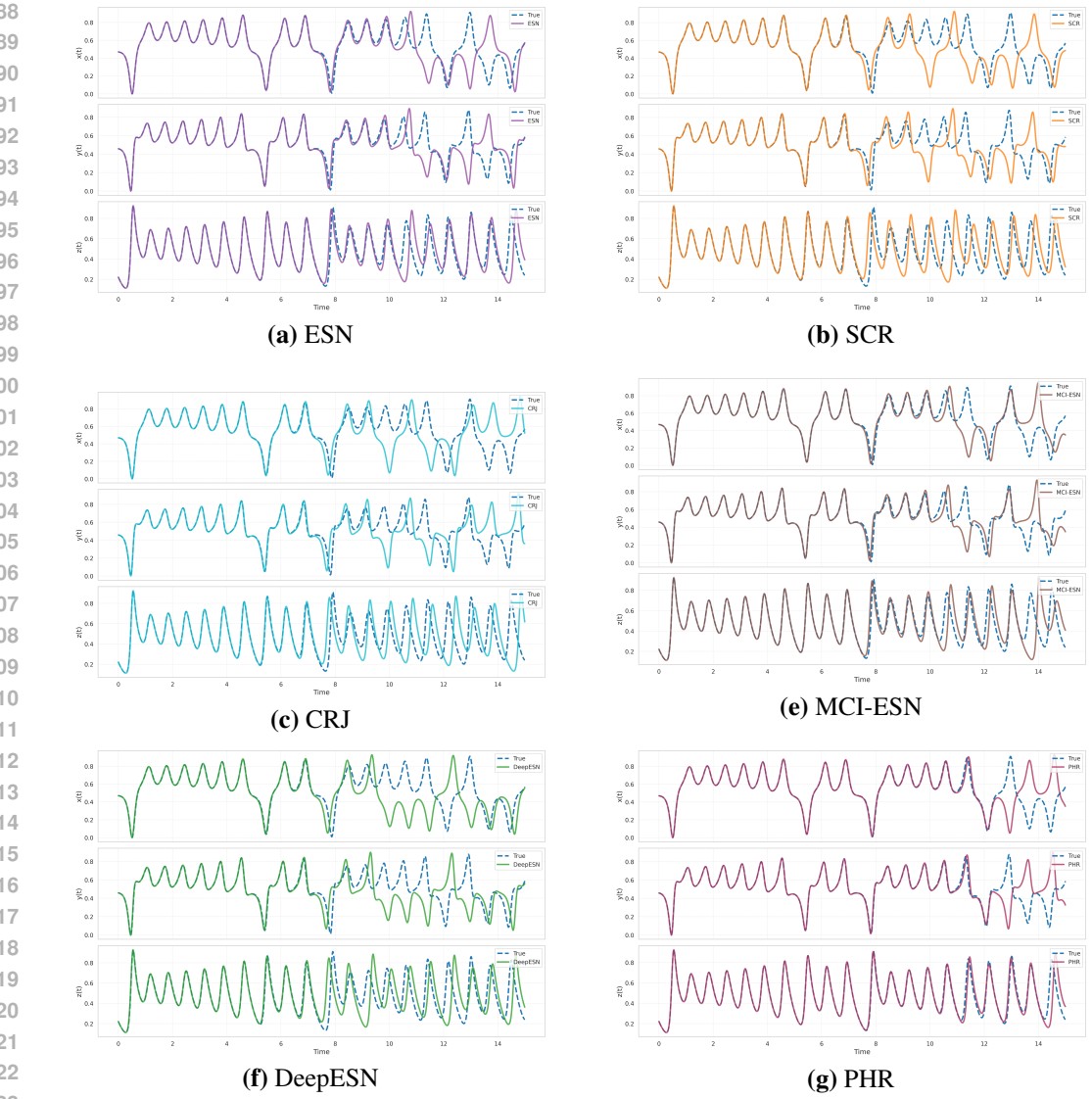

**(a)** ESN  **(b)** SCR

**(c)** CRJ  **(e)** MCI-ESN

**(f)** DeepESN  **(g)** PHR

Figure 3: Predicted trajectories by different reservoir architectures alongside the ground truth for the test segment of the Lorenz system under autoregressive forecasting.

and classification reduce to a *single convex readout fit* (7), which is sample–efficient and fast. The topological channel furnishes phase–aware latent coordinates that are *diffeomorphism–robust* by Takens-style embeddings and persistent cohomology (§3.1), so the same $W$ can transfer across sensors/views of the same underlying process; the flow channel captures short–horizon transport that is consistent with Ulam/Koopman discretizations, enabling accurate next–step prediction and detection of regime changes via deviations of $AWB$ from $(\rho_\star\beta/s)P^{(\gamma)}$. In practice this supports: (a) *system identification* and forecasting for chaotic/quasi–periodic signals (Lorenz/Rössler, turbines, climate indices), (b) *anomaly/change detection* by monitoring pooled residuals $\|AWB - (\rho_\star\beta/s)P^{(\gamma)}\|$, and (c) *few–shot task adaptation* where $W$ is reused and only the linear head is refit for new objectives or operating points. Because $W$ is norm–scaled, safe real–time deployment is facilitated: closed–loop observers or controllers can be built with a guaranteed contraction margin while the readout encodes task–specific objectives (cf. Koopman/EDMD readouts (Williams et al., 2015; Klus et al., 2016)).

In *neuroscience and biomedicine*, PHR's two–channel structure aligns with common dynamical motifs. The oscillatory planes $E_k$ implement stable, data-derived ring oscillators that track neural or physiological rhythms (theta/beta/gamma; respiratory/cardiac cycles) with bounded spectral pertur-

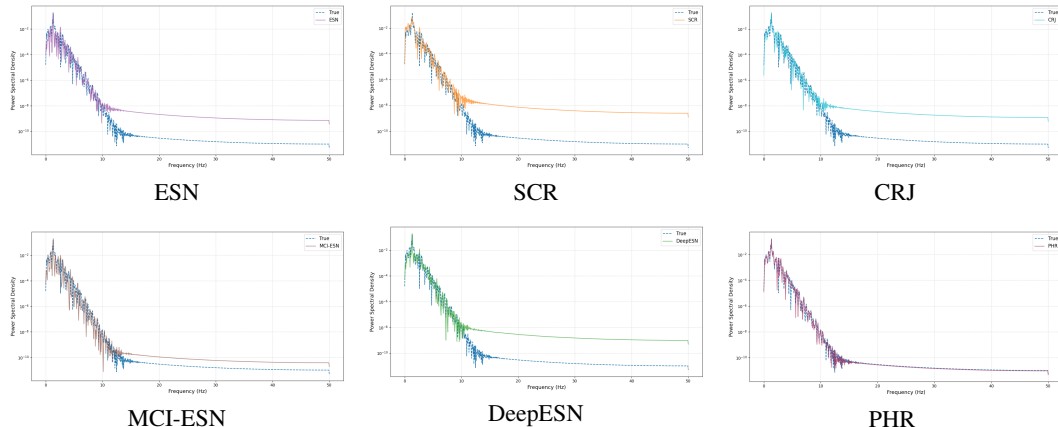



ESN           SCR           CRJ

MCI-ESN        DeepESN        PHR



Figure 4: PSD of the $z$–coordinate of Lorenz-63 for the proposed model PHR and five baseline reservoirs. Each plot compares the spectrum produced by autonomous rollout (coloured) with the true spectrum (blue). PHR best preserves the spectral envelope and high-frequency decay, whereas baselines show elevated noise floors and spectral broadening, indicating drift from the true attractor.

bation (Proposition 3.2(i)), echoing the central role of brain/physiological rhythms in coding and coordination (Buzsáki, 2006). The coarse Markov channel models *metastable transitions* among network states (e.g., sleep stages, cognitive modes) *via* finite-state transport, consistent with empirical accounts of large-scale brain dynamics and switching (Deco et al., 2017; Breakspear, 2017), and with Markov/HMM practices in sleep staging and EEG analysis (e.g., Kemp et al. (2000); Stephansen et al. (2018)). For clinical time series, PHR yields interpretable latent variables (phases, amplitudes, coarse states) for decoding or event prediction, such as arrhythmia detection in ECG (Moody & Mark, 2001) and apnea detection in respiration, with robustness inherited from persistence thresholds (Edelsbrunner & Harer, 2010; de Silva & Vejdemo-Johansson, 2009) and ESP scaling (Manjunath & Jaeger, 2013). Because $W$ is fixed once, cross-session/subject reuse is practical: retain $W$ (capturing conserved rhythms/flows) and retrain only $W_{\text{out}}$ for individuals, or update the coarse $P^{(\gamma)}$ online from fresh counts without touching $W_{\text{top}}$, thereby adapting to nonstationarities while preserving stability.

## B  SETUP AND EXTENDED RESULTS

### B.1  DATASET DESCRIPTION

**Sunspot Monthly.** The *International Sunspot Index v2.0* published by SILSO (Royal Observatory of Belgium) reports the total sunspot count for every calendar month from January 1749 to the present, giving a contiguous univariate series of $T \approx 3{,}300$ observations at a uniform 1-month cadence (World Data Center SILSO, 2020). The record is normalised to $[0, 1]$ over the *entire* span; the first 2 000 samples ($\approx$166 years) constitute the training set, and the remainder is reserved for out-of-sample evaluation. The data combine quasi-periodic forcing (11-year Schwabe, 22-year Hale, and multi-decadal Gleissberg cycles) with broad-band chaotic variability, providing a canonical long-horizon forecasting benchmark.

**Santa Fe B Cardiorespiratory Series.** Data Set B of the 1991 Santa Fe Time-Series Prediction and Modelling Competition is a *trivariate* polysomnography recording that simultaneously tracks heart-rate (HR), chest volume (RESP) and peripheral oxygen saturation (SpO$_2$) for a continuous 20-minute interval sampled at $f_s = 2\,\text{Hz}$ (Jaeger, 2007). After converting the raw ASCII file into a matrix $\boldsymbol{u}_t \in \mathbb{R}^3$, each channel is linearly detrended and scaled to unit variance using statistics computed on the training split. We allocate the first 1200 samples (10 minutes) for training and the remaining 600 samples for evaluation, formulating a next-step multivariate forecasting task that couples slow respiration-driven oscillations (RESP), faster autonomic heart-rate variability (HR) and the more slowly drifting SpO$_2$ signal. The dataset therefore probes the reservoir's ability to integrate interdependent physiological rhythms operating on distinct but overlapping time-scales.

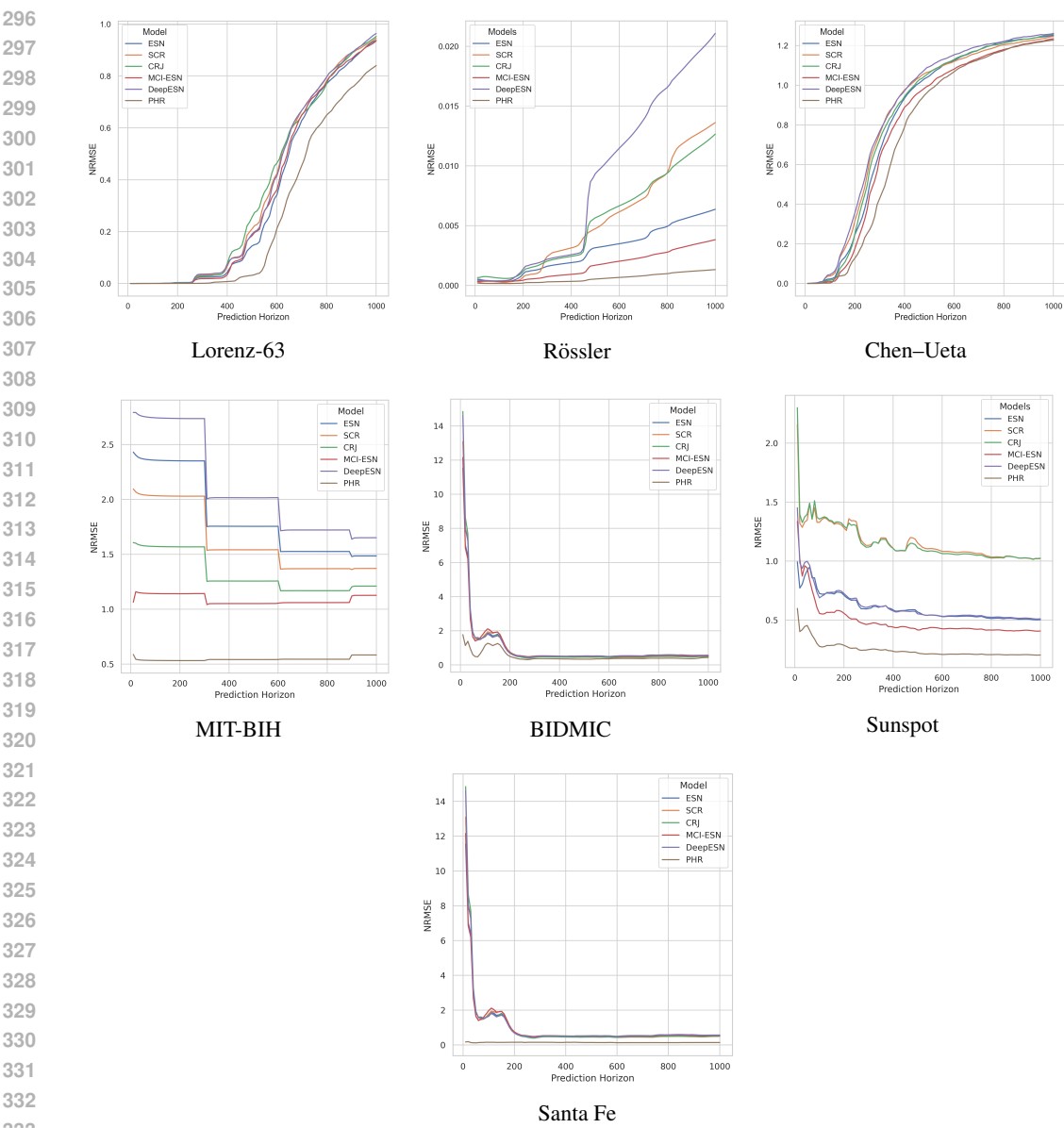

Lorenz-63

Rössler

Chen–Ueta

MIT-BIH

BIDMIC

Sunspot

Santa Fe

Figure 5: NRMSE for autoregressive predictions as a function of forecast horizon for the canonical chaotic benchmarks and real-life datasets.

**MIT–BIH Arrhythmia.** The MIT–BIH Arrhythmia Database contains 48 half-hour two-lead ECG records digitised at $360\,\mathrm{Hz}$ (11-bit, $\pm5\,\mathrm{mV}$) with expert beat- and rhythm-level annotations (Moody & Mark, 2001; Goldberger et al., 2000). For single-channel forecasting we choose Lead II of record 100, extract the first 25000 samples ($\approx 70\,\mathrm{s}$), and scale them to zero mean and unit variance. A 3-dimensional delay embedding reconstructs the local dynamical manifold, yielding a sequence whose quasi-periodic P-QRS-T morphology is punctuated by occasional ectopic beats—an ideal test of biomedical robustness.

**BIDMC PPG & Respiration.** Record `bidmc01` from the BIDMC PPG & Respiration corpus (Pimentel et al., 2016; Goldberger et al., 2000) is an eight-minute ICU waveform captured at $125\,\mathrm{Hz}$ and composed of photoplethysmogram (PPG), impedance-derived respiration (RESP) and Lead-II ECG. We retain the 60000-sample PPG and RESP channels, detrend each, and segment them into overlapping 10-second windows (1250 samples) with a 1-second stride. The reservoir receives

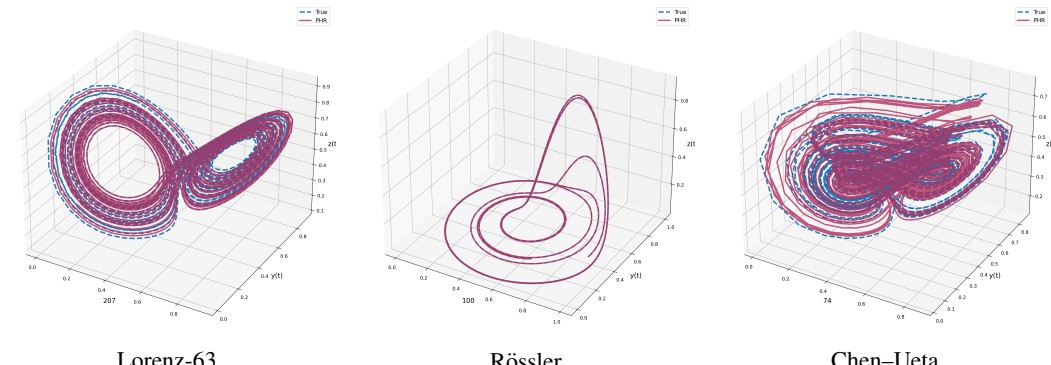

Figure 6: Three-dimensional phase portraits generated by PHR for Lorenz-63, Rössler, and Chen–Ueta over a 1 000-step autonomous rollout (red), overlaid on the reference attractors (blue dots). Close overlap confirms that PHR preserves the global geometry of all three chaotic systems.

| Dataset | $d_{in}$ | $f_s$ | $T_{tot}$ | $T_{wo}$ | $T_{eff}$ |
|---|---|---|---|---|---|
| Lorenz–63 | 3 | 50 Hz equiv. | 12 500 | 2 000 | 10 500 |
| Rössler | 3 | 50 Hz equiv. | 12 500 | 2 000 | 10 500 |
| Chen–Ueta | 3 | 50 Hz equiv. | 12 500 | 2 000 | 10 500 |
| BIDMC PPG/Resp | 1 | 125 Hz | 60 000 | 5 000 | 55 000 |
| MIT–BIH ECG | 1 | 360 Hz | 25 000 | 5 000 | 20 000 |
| Santa Fe B | 3 | 2 Hz | 2 400 | 100 | 2 300 |
| Sunspot Monthly | 1 | 1 month$^{-1}$ | 3 315 | 100 | 3 215 |

Table 3: Overview of benchmark datasets. $d_{in}/d_{out}$ are the input/target dimensions, $f_s$ the sampling frequency after re-sampling, and $T_{eff} = T_{tot} - T_{wo}$.

the PPG (optionally ECG) as input and must regress the synchronous RESP waveform, framing a continuous sequence-to-sequence task with rich cardiorespiratory coupling.

**Lorenz–63.** The Lorenz–63 system models thermal convection in an idealised fluid layer and is governed by the quadratic ODE $\dot{x} = \sigma(y-x)$, $\dot{y} = x(\rho-z) - y$, $\dot{z} = xy - \beta z$, with the classical *chaotic* settings $(\sigma, \rho, \beta) = (10, 28, \frac{8}{3})$ (Lorenz, 1963). Linearising about the three equilibrium points—one at the origin and two symmetric saddles—reveals a pair of complex-conjugate eigenvalues with positive real part once $\rho > 1 + \sigma/(\sigma + \beta)$, triggering a sub-critical Hopf bifurcation and the birth of the famous double-scroll ("butterfly") attractor. Rigorous computation gives a largest Lyapunov exponent $\lambda_{max} = 0.9056 \pm 0.0002$ (time-unit$^{-1}$), a Kaplan–Yorke (information) dimension $D_{KY} = 2.062$, and a correlation dimension $D_2 \approx 2.05$. Because only one exponent is positive, the error growth rate is exponential but still tractable, making Lorenz–63 the de-facto baseline for evaluating long-horizon chaotic predictors. We integrate the system for 12 500 steps after a 2000-step transient, ensuring that the segment alternates between both lobes so that the predictor must solve the *return-map* as well as the local Jacobian dynamics.

**Rössler.** The Rössler equations $\dot{x} = -y - z$, $\dot{y} = x + ay$, $\dot{z} = b + z(x - c)$, with $(a, b, c) = (0.2, 0.2, 5.7)$ generate a *single-scroll* chaotic attractor whose first-return map on the Poincaré section $z = z_{min}$ is topologically conjugate to the logistic map, yielding a one-dimensional kneading sequence of symbolic dynamics (Rossler, 1976). The maximal Lyapunov exponent is $\lambda_{max} \approx 0.0712$, an order of magnitude smaller than that of Lorenz–63, which postpones divergence of nearby trajectories and produces a spectrum of finite-time Lyapunov exponents heavily skewed towards zero. Consequently, prediction errors grow more *slowly* but linger, exposing whether a model's inductive bias captures the weakly non-hyperbolic stretching and folding. A 12 500-step slice is harvested after discarding 2 000 transients, providing a benchmark on which memory capacity rather than raw separation dominates.

**Chen–Ueta.** The Chen–Ueta flow modifies Lorenz by interchanging two nonlinear terms, $\dot{x} = a(y-x)$, $\dot{y} = (c-a)x - xz + cy$, $\dot{z} = xy - bz$, and for $(a, b, c) = (35, 3, 28)$ possesses *two*

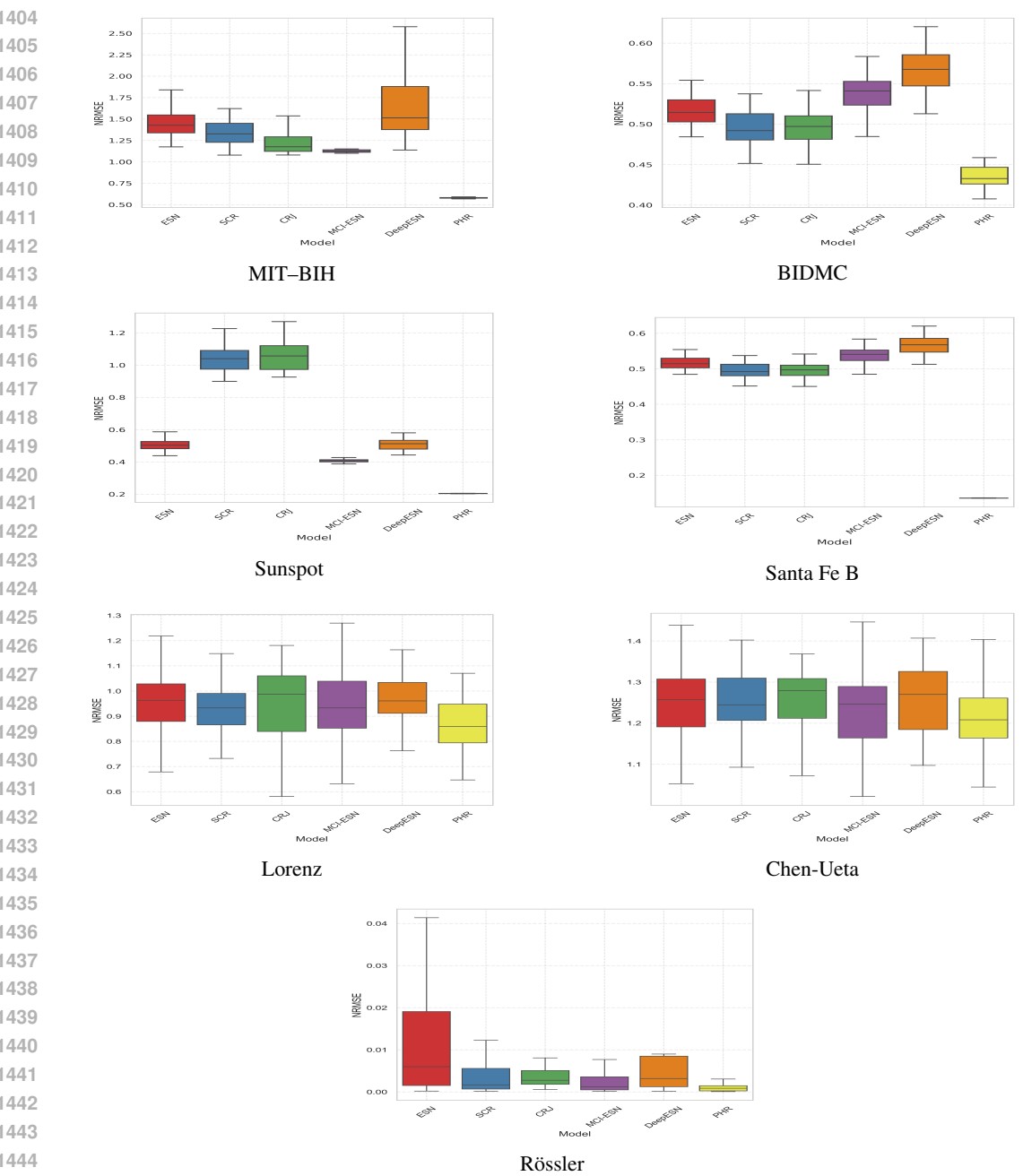

Figure 7: Distribution of sample-level NRMSE after a 1000-step open-loop rollout on seven benchmarks. Each box-and-whisker shows ten seeds: box = inter-quartile range, line = median, whiskers = non-outliers, grey bar = mean. Lower is better; PHR displays the tightest IQR and lowest median on every dataset.

positive Lyapunov exponents, $\lambda_1 \approx 2.00$ and $\lambda_2 \approx 0.45$, while $\lambda_3 < 0$ (Chen & Ueta, 1999). The Kaplan–Yorke dimension therefore satisfies $3 < D_{KY} < 4$, rendering the attractor *hyper-chaotic*. Its higher local expansion rate and the coexistence of two unstable manifolds lead to rapid loss of predictability and extremely intricate fractal folding, which stress-tests the reservoir's non-linear separation ability and its capability to encode multi-directional volume expansion. We integrate for 12 500 steps following a 2 000 step transient; this window spans multiple high-curvature excursions, forcing the model to reconcile both fast and intermediate dynamical scales.

Collectively the three canonical chaotic flows form a graded staircase of chaotic difficulty—single vs. double scroll, single vs. double positive Lyapunov exponents—enabling a systematic assessment of how the proposed PHR scales with increasing dynamical complexity. A summary for all datasets is provided in Table 3.

| Dataset | H | NRMSE ↓ (mean ± s.d.) | | | | | |
|---|---|---|---|---|---|---|---|
| | | LSTM | NVAR | TCN | Transformer | SW | PHR |
| Lorenz | 200 | 1.2026 ± 0.2382 | 1.6031 ± 0.2452 | 1.6917 ± 0.4007 | 1.6097 ± 0.3884 | 0.0090 ± 0.0130 | **0.0004 ± 0.0005** |
| | 400 | 1.0383 ± 0.0800 | 1.4235 ± 0.0928 | 1.4759 ± 0.2526 | 1.3922 ± 0.2427 | 0.1082 ± 0.1097 | **0.0075 ± 0.0103** |
| | 600 | 1.1483 ± 0.1080 | 1.5861 ± 0.1527 | 1.6875 ± 0.2900 | 1.5852 ± 0.2653 | 0.7048 ± 0.2202 | **0.2121 ± 0.1877** |
| | 800 | 0.9628 ± 0.0368 | 1.3475 ± 0.0471 | 1.4060 ± 0.2046 | 1.3913 ± 0.4721 | 0.8234 ± 0.1326 | **0.6495 ± 0.1928** |
| | 1000 | 1.0703 ± 0.0321 | 1.5022 ± 0.0476 | 1.5638 ± 0.2285 | 1.9159 ± 0.4818 | 1.0512 ± 0.1225 | **0.8481 ± 0.1358** |
| Chen–Ueta | 200 | 1.0736 ± 0.0245 | 0.3280 ± 0.3245 | 1.7004 ± 0.1079 | 1.5320 ± 0.1578 | 0.2934 ± 0.2266 | **0.1278 ± 0.2424** |
| | 400 | 0.9467 ± 0.0129 | 0.9074 ± 0.1709 | 1.4779 ± 0.0999 | 1.3523 ± 0.1301 | 0.8877 ± 0.1488 | **0.7958 ± 0.2006** |
| | 600 | 1.0834 ± 0.0109 | 1.1985 ± 0.1446 | 1.7133 ± 0.1067 | 1.5562 ± 0.1295 | 1.1966 ± 0.1233 | **1.0555 ± 0.1199** |
| | 800 | 0.9196 ± 0.0074 | 1.0956 ± 0.0916 | 1.4571 ± 0.0875 | 1.3171 ± 0.0983 | 1.0985 ± 0.0967 | **1.1632 ± 0.0878** |
| | 1000 | 1.0602 ± 0.0080 | 1.3086 ± 0.0845 | 1.6636 ± 0.0949 | 1.5113 ± 0.1076 | 1.4196 ± 0.0914 | **1.2316 ± 0.0737** |
| Rössler | 200 | 1.0812 ± 0.0233 | 0.0007 ± 0.0013 | 1.1246 ± 0.1425 | 2.3080 ± 1.9116 | 0.0939 ± 0.2581 | **0.0002 ± 0.0001** |
| | 400 | 0.9533 ± 0.0114 | 0.0029 ± 0.0076 | 1.2723 ± 0.2087 | 1.8106 ± 1.1900 | 0.2851 ± 0.6510 | **0.0003 ± 0.0003** |
| | 600 | 1.0713 ± 0.0088 | 0.0066 ± 0.0113 | 1.5163 ± 0.2680 | 2.0018 ± 1.3390 | 0.6319 ± 1.1674 | **0.0007 ± 0.0007** |
| | 800 | 0.9282 ± 0.0063 | 0.0086 ± 0.0132 | 1.3175 ± 0.1390 | 1.7612 ± 1.2643 | 0.7486 ± 1.2440 | **0.0010 ± 0.0011** |
| | 1000 | 1.0377 ± 0.0052 | 0.0140 ± 0.0187 | 1.4574 ± 0.1625 | 2.2183 ± 2.0249 | 1.0771 ± 1.5656 | **0.0013 ± 0.0016** |
| MIT–BIH | 300 | 0.8638 ± 0.0269 | 1.3428 ± 0.0000 | 0.9276 ± 0.0080 | 5.1384 ± 3.3092 | 2.9550 ± 0.8557 | **0.5320 ± 0.0442** |
| | 600 | 0.7484 ± 0.0242 | 1.1656 ± 0.2367 | 0.8218 ± 0.0043 | 4.4601 ± 2.8918 | 1.8478 ± 0.5179 | **0.5417 ± 0.0236** |
| | 1000 | 0.8497 ± 0.0255 | 1.2918 ± 0.1372 | 0.9146 ± 0.0070 | 4.6228 ± 2.9510 | 1.5467 ± 0.3971 | **0.5833 ± 0.0123** |
| BIDMC | 300 | 0.9470 ± 0.0062 | 0.9045 ± 0.2933 | 0.9049 ± 0.0125 | 1.4003 ± 0.2142 | 0.6960 ± 0.0570 | **0.3655 ± 0.0143** |
| | 600 | 1.0667 ± 0.0031 | 0.9841 ± 0.1432 | 1.0138 ± 0.0131 | 1.6448 ± 0.2256 | 0.7895 ± 0.0575 | **0.3571 ± 0.0151** |
| | 1000 | 0.9219 ± 0.0057 | 0.8983 ± 0.0741 | 0.8674 ± 0.0139 | 1.4456 ± 0.1990 | 0.6871 ± 0.0443 | **0.4352 ± 0.0133** |
| Sunspot | 300 | 0.5727 ± 0.0068 | 0.4182 ± 0.0013 | 0.4299 ± 0.0026 | 0.2768 ± 0.0041 | 0.6269 ± 0.0439 | **0.2505 ± 0.0011** |
| | 600 | 0.6055 ± 0.0060 | 0.2907 ± 0.0098 | 0.2932 ± 0.0017 | 0.1907 ± 0.0043 | 0.5586 ± 0.0410 | **0.2093 ± 0.0015** |
| | 1000 | 0.6800 ± 0.0074 | 0.3312 ± 0.0190 | 0.3338 ± 0.0021 | 0.2178 ± 0.0053 | 0.6266 ± 0.0460 | **0.2044 ± 0.0010** |
| Santa Fe | 300 | 0.6680 ± 0.1017 | 1.3171 ± 0.0013 | 0.3213 ± 0.0097 | 3.3260 ± 1.6923 | 0.1974 ± 0.0023 | **0.1485 ± 0.0003** |
| | 600 | 0.5746 ± 0.1409 | 1.3918 ± 0.0113 | 0.2650 ± 0.0059 | 2.4134 ± 1.2261 | 0.1647 ± 0.0014 | **0.1262 ± 0.0003** |
| | 1000 | 0.5412 ± 0.1174 | 1.2626 ± 0.0168 | 0.2569 ± 0.0060 | 2.4788 ± 1.2549 | 0.1634 ± 0.0016 | **0.1361 ± 0.0003** |

Table 4: NRMSE (*mean ± s.d.*) across all benchmarks and horizons for comparison with gradient-based baselines.

| Dataset | H | NRMSE (×10⁻⁶) ↓ | | | | | |
|---|---|---|---|---|---|---|---|
| | | ESN | SCR | CRJ | MCI-ESN | DeepESN | PHR |
| Lorenz | 200 | 5.3084 ± 1.8201 | 4.5661 ± 1.1055 | 4.0665 ± 2.2136 | 5.1315 ± 2.1534 | 8.0426 ± 4.9766 | **1.9459 ± 0.4228** |
| | 400 | 6.1453 ± 3.4127 | 6.8544 ± 8.2930 | 5.8480 ± 5.9654 | 5.5835 ± 3.0159 | 14.3798 ± 25.1950 | **1.9610 ± 0.3980** |
| | 600 | 6.2136 ± 2.9307 | 6.7382 ± 7.2498 | 6.2639 ± 5.5241 | 5.6319 ± 2.4513 | 14.8085 ± 22.4161 | **1.9327 ± 0.3609** |
| | 800 | 5.8794 ± 1.9437 | 6.0779 ± 4.8747 | 5.8424 ± 3.8676 | 5.2979 ± 1.6073 | 13.1852 ± 15.6860 | **2.1054 ± 0.4672** |
| | 1000 | 5.7315 ± 1.4810 | 5.8048 ± 3.9922 | 5.4164 ± 3.1885 | 5.1994 ± 1.1876 | 12.0887 ± 13.1349 | **2.0554 ± 0.3898** |
| Chen-Ueta | 200 | 17.4775 ± 87.5577 | 76.3789 ± 65.2906 | 27.7972 ± 14.9489 | 13.0443 ± 5.4424 | 55.3606 ± 26.8909 | **5.1368 ± 2.8103** |
| | 400 | 36.9059 ± 53.6185 | 152.4160 ± 210.0693 | 53.8950 ± 69.5504 | 22.0085 ± 24.4040 | 78.7224 ± 77.9541 | **8.3424 ± 9.3712** |
| | 600 | 38.7540 ± 54.4478 | 164.3067 ± 208.3044 | 57.3656 ± 73.4667 | 23.3657 ± 25.5862 | 83.4452 ± 74.6511 | **8.4135 ± 7.8776** |
| | 800 | 36.5894 ± 45.6740 | 155.8315 ± 175.5068 | 54.6267 ± 61.5250 | 22.4755 ± 21.2737 | 81.9084 ± 62.4709 | **8.3066 ± 6.4822** |
| | 1000 | 36.3487 ± 39.3055 | 161.2043 ± 150.8282 | 56.0409 ± 52.6167 | 21.9270 ± 18.5333 | 80.6558 ± 57.4452 | **8.1426 ± 5.7768** |
| Rössler | 200 | 18.2036 ± 22.4166 | 186.2890 ± 387.9631 | 18.7657 ± 17.9382 | 15.9787 ± 14.2553 | 21.5473 ± 17.8969 | **5.3958 ± 2.4637** |
| | 400 | 33.4001 ± 48.8892 | 157.7964 ± 244.4589 | 23.2474 ± 25.8931 | 25.8438 ± 34.6041 | 31.4314 ± 45.1686 | **7.1545 ± 8.7532** |
| | 600 | 99.7132 ± 206.7794 | 237.6528 ± 393.0541 | 62.2698 ± 125.8244 | 101.1291 ± 224.4540 | 98.6072 ± 213.4015 | **22.0354 ± 44.6574** |
| | 800 | 104.9667 ± 197.2623 | 233.9971 ± 372.8086 | 62.7120 ± 120.4620 | 103.5512 ± 214.1152 | 103.3252 ± 203.8183 | **23.1159 ± 42.5334** |
| | 1000 | 98.2334 ± 184.3336 | 225.4536 ± 356.0678 | 58.9122 ± 112.6381 | 97.1532 ± 200.1728 | 96.8307 ± 190.4315 | **21.8021 ± 39.6918** |

Table 5: NRMSE (mean ± s.d.) on the *canonical chaotic benchmarks* over multiple horizons (H). Forecasts are produced in **open-loop** mode. For each horizon the best score is **bold** and the runner-up is underlined. Results are averaged over $5 \times 3 \times 3 = 45$ runs (5 seeds, 3 different initializations of trajectory, 3 train–test splits).

## B.2 BASELINES

To quantify the incremental value of the proposed method, we compare it against a suite of widely cited baselines—both reservoir architectures (ESN, SCR, CRJ, MCI-ESN, DeepESN, Small-World ESN) and non-reservoir sequence models (NVAR, LSTM, TCN, single-layer causal Transformer)—under a *harmonized capacity and training protocol* (cf. Tabs. 1,4, 5). Unless the original design intrinsically requires a multi-core layout (as in MCI-ESN), single-layer reservoirs use exactly $N = 300$ recurrent units, while hierarchical reservoirs use three layers of 100 units. MCI-ESN

| Factor (canonical ESN) | Values (one varied at a time) | N (neurons) | NRMSE@H=600 ↓ | VPT ↑ | ADev ↓ |
|---|---|---|---|---|---|
| *Baseline (PHR, Lorenz–63, 600-step AR): N=300, leak λ=0.20, target ρ⋆=0.94, input scale = 0.5, Q=200, stride s=5, K$_{\max}$=3, auto-tuned (α$_{top}$, β$_{flow}$), nzr= 4, nzc= 12.* | | | | | |
| **Baseline result (mean±std over seeds)** | | 300 | **0.2121 ± 0.1877** | **10.94 ± 1.65** | **29.11 ± 9.53** |
| Leak λ | 0.15 | 300 | 0.2385 ± 0.1954 | 10.62 ± 1.72 | 30.02 ± 9.88 |
| | 0.20 (baseline) | 300 | 0.2121 ± 0.1877 | 10.94 ± 1.65 | 29.11 ± 9.53 |
| | 0.25 | 300 | 0.2259 ± 0.1901 | 10.78 ± 1.69 | 29.67 ± 9.71 |
| | 0.35 | 300 | 0.2684 ± 0.2053 | 9.92 ± 1.86 | 31.45 ± 10.21 |
| Target spectral norm ρ⋆ | 0.92 | 300 | 0.2432 ± 0.1989 | 10.42 ± 1.79 | 30.18 ± 9.96 |
| | 0.94 (baseline) | 300 | 0.2121 ± 0.1877 | 10.94 ± 1.65 | 29.11 ± 9.53 |
| | 0.96 | 300 | 0.2240 ± 0.1915 | 10.81 ± 1.70 | 29.58 ± 9.68 |
| | 0.98 | 300 | 0.2613 ± 0.2076 | 10.08 ± 1.83 | 31.01 ± 10.07 |
| Input scale $\|W_{in}\|$ | 0.2 | 300 | 0.2524 ± 0.2022 | 10.23 ± 1.82 | 31.14 ± 10.12 |
| | 0.5 (baseline) | 300 | 0.2121 ± 0.1877 | 10.94 ± 1.65 | 29.11 ± 9.53 |
| | 1.0 | 300 | 0.2410 ± 0.1995 | 10.34 ± 1.78 | 30.72 ± 9.98 |

Table 6: **Ablation over canonical parameters for PHR** on Lorenz–63: 600-step autoregressive forecasting. Each block varies a single factor; all other settings are fixed to the baseline shown (top). Report *mean ± std* over 45 trials. NRMSE is the primary metric at horizon $H = 600$; VPT (valid prediction time, ↑) and ADev (attractor deviation, ↓) are computed with the same protocol as Tab. 1.

follows its published two-core configuration (two sparsely coupled 300-unit cycles). All methods receive the same pre-processed inputs, apply the same wash-out $T_{wo}$, and (where applicable) train the linear read-out by ridge regression with a shared grid $\alpha_{ridge} \in \{10^{-6}, 10^{-5}, 10^{-4}\}$, ensuring comparability of optimization and regularization across models.

- **ESN** (Jaeger, 2001). Erdős–Rényi connectivity with $p \in \{0.10, 0.20, 0.25, 0.30\}$. Spectral radius $\rho_\star$ and input scaling $\|\mathbf{W}_{in}\|_2$ are chosen from the logarithmic grid $\rho_\star \in \{0.3, 0.6, 0.9\} \times \|\mathbf{W}_{in}\|_2 \in \{0.1, 0.3, 1.0\}$.

- **SCR** (Li et al., 2024). A single directed cycle of length 300 with uniform edge weight $w_c$. Tuning grid: $w_c \in \{0.3, 0.6, 0.8, 0.9, 1.0\}$.

- **CRJ** (Rodan & Tino, 2012). SCR with additional "jump" edges of fixed span $J$. We sweep $J \in \{5, 10, 12, 15, 20, 30\} \times w_c \in \{0.3, 0.6, 0.7, 0.8, 0.9\}$, keeping unit in-degree.

- **MCI-ESN** (Liu et al., 2024). Two sparsely coupled 300-unit cycle ESNs (total $N = 600$). Hyper-parameters follow $(\mu, \eta) \in \{0.6, 0.7, 0.8, 1.0\}^2$ (intra-core radii) and $\theta \in \{0.4, 0.5, 0.6, 0.8\}$ (cross-core mixing).

- **DeepESN** (Gallicchio & Micheli, 2017). Three stacked leaky reservoirs (100 + 100 + 100 units). A common input scale $\|\mathbf{W}_{in}\|_2$ is selected as for ESN. Layer-$\ell$ spectral radii decay geometrically $\rho_\ell = \rho_\star^\ell$ with $\rho_\star \in \{0.4, 0.6, 0.8\}$; the shared leak $\alpha \in \{0.3, 0.5, 0.7\}$ is co-optimised.

- **Non-linear Vector Auto-Regression (NVAR)** (Farmer & Sidorowich, 1987) NVAR models the system with a fixed delay line ($k = 100$) followed by a quadratic polynomial expansion. The 3-D input is flattened over the last 100 steps into $x \in \mathbb{R}^{300}$ and mapped to $\phi = [1; x; x^{\otimes 2}] \in \mathbb{R}^{45451}$, where the $\binom{300+1}{2} = 45{,}150$ second-order monomials cover all pairs with replacement. A ridge-regularised least-squares fit then produces a read-out matrix $W_{out}$ (136,353 parameters); no other weights are learnt. Prediction is strictly causal: each output is fed back into the delay buffer before the next evaluation, yielding an autoregressive closed loop with a 100-step effective memory.

- **LSTM** (Hochreiter & Schmidhuber, 1997) The recurrent reference model is a single-layer LSTM with 500 hidden units followed by a linear projection to $\mathbb{R}^3$. This configuration introduces roughly $1.01 \times 10^6$ tunable parameters, about 160× the parameter budget of PHR. Training uses full-sequence teacher forcing, Adam ($10^{-3}$) and 80 epochs to minimise mean-squared error. During free-run evaluation the network closes the loop on itself: the latest prediction becomes the next input, forcing the LSTM to retain long-term context internally and revealing how much horizon length a conventional gated RNN can sustain when operating under the same parameter budget as our biologically grounded reservoir.

- **Temporal-Convolutional Network (TCN)** (Bai et al., 2018) Our convolutional benchmark is a two-stage, strictly causal TCN whose kernels have size 3 and dilations 1 and 2, giving a receptive field of five time-steps. Each convolution is followed by a ReLU and the right-hand padding is cropped so that no future leakage occurs. Fixing the channel width at 500 results in 757,003 adjustable parameters, about 120 times the parameter budget of our

reservoir. Training is carried out in teacher-forcing mode on a single long sequence (batch = 1), optimising one-step MSE with Adam (learning rate $10^{-3}$, 80 epochs). For forecasting, the newest five observations seed an autoregressive loop in which the window is shifted forward after each prediction to maintain causality.

- **Single-Layer Causal Transformer** (Vaswani et al., 2017) The transformer baseline consists of one encoder block with $d_{\text{model}} = 100$, a single attention head, and a feed-forward sub-layer of width $4d_{\text{model}}$. Three-dimensional inputs are linearly projected, summed with a fixed sinusoidal positional code of length $L$, and passed through the encoder; only the final token is used to predict the next state, so causality is preserved without an explicit mask. Parameter count ($1.22 \times 10^5$) about 20 times PHR. Training employs sliding windows of length $L = 20$ (stride 1) and mini-batches of 64, with Adam at $2 \times 10^{-3}$ for 100 epochs. At test time the window is rolled forward autoregressively: each new prediction is appended, re-encoded, and used to drive the next step, so the effective context exactly matches what was seen during learning.

- **Small-World Topology ESN** (Kawai et al., 2019). We include a Small-World Echo-State Network (SW) whose recurrent matrix is obtained by (i) generating a Watts–Strogatz graph with $N$ nodes, mean degree $k$ and rewiring probability $p = 0.1$, (ii) assigning i.i.d. weights drawn from $\mathcal{N}(0, 1)$ to the existing edges, and (iii) rescaling the resulting matrix to a target spectral radius $\rho_\star$. The input vector projects only to a compact cluster of "input" neurons, while the read-out taps a disjoint cluster of "output" neurons placed at a maximal geodesic distance, replicating the spatial segregation used in the original study. Kawai et al. showed that this small-world topology widens the range of $\rho_\star$ values for which the echo-state property is preserved and significantly improves both memory-capacity and nonlinear prediction tasks compared with dense or lattice reservoirs; we therefore evaluate SW-ESN under the same capacity budget ($N = 300$, $k = 6$) and hyper-parameter sweep as the other baselines.

**Training & model selection.** For every Cartesian hyper-parameter tuple we fit the read-out on the training split, compute NRMSE on the validation split, and retain the best model to score the test set with all four metrics. By fixing the global random seed we ensure that differences arise solely from reservoir topology and intrinsic time-scale, not from stochastic weight realisations.

**Evaluation Protocols** Chaotic-system benchmarks are single-channel state reconstructions in which the model's input and output live in the *same* 3-D phase space; we therefore test in **closed-loop** (autonomous) mode—after a 1 000-step wash-out we seed the reservoir with the last true state, run it for $H$ steps while feeding each prediction back as the next input, and quantify compounding error through metrics. In contrast, the real-world collections pair heterogeneous sensor values with domain-specific targets—i.e. the driver and the prediction lie in *different* feature spaces or semantic channels—so recycling the model's output as a surrogate input would violate the data-generation mechanism and induce uncontrolled distribution shift. Accordingly, we adopt a **teacher-forced open-loop** protocol for these tasks: at every intermediate step $t + \tau$ ($0 \leq \tau < H$) the reservoir receives the ground-truth measurement, produces $\hat{\mathbf{y}}_{t+\tau+1}$, and only the terminal prediction $\hat{\mathbf{y}}_{t+H}$ is scored; this treats the model as a real-time forecaster or filter that augments, but never contaminates, the sensor stream.

For the Lorenz dataset, ESN used reservoir size 300 and connectivity ratio 0.2; SCR used reservoir size 300 and edge weight 0.8; CRJ used reservoir size 300 with edge weight 0.7 and jump size 20; MCI-ESN used sub-reservoir size 300 with edge weight $\mu = 0.6$, inter-reservoir connection $\eta = 0.6$, and coefficient $\theta = 0.4$; DeepESN had 3 layers with reservoir sizes $100, 100, 100$; PHR used 300 anchor points with blend weights $(\alpha_{\text{top}}, \beta_{\text{flow}}, \xi) = (0.2375, 0.7125, 0.0500)$. For the Rössler dataset, ESN used reservoir size 300 and connectivity ratio 0.3; SCR used reservoir size 300 and edge weight 1.0; CRJ used reservoir size 300 with edge weight 0.6 and jump size 10; MCI-ESN used sub-reservoir size 300 with $\mu = 0.8$, $\eta = 1.0$, $\theta = 0.8$; DeepESN again had 3 layers of sizes $100, 100, 100$; PHR used 300 anchors with $(\alpha_{\text{top}}, \beta_{\text{flow}}, \xi) = (0.0679, 0.8821, 0.0500)$. For the Chen–Ueta dataset, ESN used reservoir size 300 and connectivity ratio 0.3; SCR used reservoir size 300 and edge weight 0.8; CRJ used reservoir size 300 with edge weight 0.8 and jump size 10; MCI-ESN used sub-reservoir size 300 with $\mu = 0.8$, $\eta = 1.0$, $\theta = 0.6$; DeepESN had 3 layers of sizes $100, 100, 100$; and PHR used 300 anchors with $(\alpha_{\text{top}}, \beta_{\text{flow}}, \xi) = (0.0826, 0.8674, 0.0500)$. For the BIDMC PPG/Resp dataset, ESN used reservoir size 300 and connectivity 0.25; SCR used reservoir size 300 and edge weight 0.9; CRJ used reservoir size 300 with edge weight 0.7 and jump

size 10; MCI-ESN used sub-reservoir size 300 with $\mu = 0.7$, $\eta = 0.9$, $\theta = 0.6$; DeepESN had 3 layers of $100, 100, 100$; PHR used 300 anchors with $(\alpha_{\text{top}}, \beta_{\text{flow}}, \xi) = (0.2036, 0.7464, 0.0500)$. For MIT–BIH ECG, ESN used reservoir size 300 and connectivity 0.25; SCR used reservoir size 300 and edge weight 0.9; CRJ used reservoir size 300 with edge weight 0.7 and jump size 10; MCI-ESN used sub-reservoir size 300 with $\mu = 0.75$, $\eta = 0.9$, $\theta = 0.6$; DeepESN had 3 layers of $100, 100, 100$; PHR used 300 anchors with $(\alpha_{\text{top}}, \beta_{\text{flow}}, \xi) = (0.1357, 0.8143, 0.0500)$. For the Santa Fe B laser series, ESN used reservoir size 300 and connectivity 0.3; SCR used reservoir size 300 and edge weight 0.8; CRJ used reservoir size 300 with edge weight 0.7 and jump size 10; MCI-ESN used sub-reservoir size 300 with $\mu = 0.7$, $\eta = 0.8$, $\theta = 0.5$; DeepESN had 3 layers of $100, 100, 100$; PHR used 300 anchors with $(\alpha_{\text{top}}, \beta_{\text{flow}}, \xi) = (0.0328, 0.9172, 0.0500)$. For Sunspot Monthly, ESN used reservoir size 300 and connectivity 0.2; SCR used reservoir size 300 and edge weight 0.8; CRJ used reservoir size 300 with edge weight 0.7 and jump size 12; MCI-ESN used sub-reservoir size 300 with $\mu = 0.7$, $\eta = 0.9$, $\theta = 0.6$; DeepESN had 3 layers of $100, 100, 100$; and PHR used 300 anchors with $(\alpha_{\text{top}}, \beta_{\text{flow}}, \xi) = (0.3563, 0.5938, 0.0500)$.

**Computational and complexity notes.** We use RIPSER for fast $H^1$ with cocycles (Bauer, 2021). The Laplacian systems are sparse and solved per connected component with a small Tikhonov $\mu$ and an LSQR fallback (Paige & Saunders, 1982). The PH stride $s$ trades accuracy for cost; since $\widehat{\omega}_\ell$ uses only *temporal* increments of $\theta_t^{(\ell)}$, subsampling at modest $s$ typically preserves the dominant angular velocity while reducing $O(n^2)$ storage and compute. Forming $C$ costs $O(T)$ operations for fixed $h$; normalization and teleportation are $O(Q^2)$. The default sparse constructions of $A$ and $B$ cost $O(Q\,\texttt{nzr} + Q\,\texttt{nzc})$ and yield $W_{\text{flow}}$ applicable in $O((\texttt{nzr} + \texttt{nzc})Q)$ time when used as a product. Small $\varepsilon$-pseudocounts ensure no dead rows; $\gamma \approx 10^{-3}$ suffices to regularize nearly reducible chains without washing out directed transport. A naive matrix–vector step with dense $W$ costs $O(N^2)$; however, $W_{\text{top}}$ is a permutation of block-diagonal $2 \times 2$ rotations plus diagonal decay, and $W_{\text{flow}} = BP^{(\gamma)}A$ is a product of sparse–stochastic maps with a $Q \times Q$ core. Implementations that *apply* $W$ as $(\alpha_{\text{top}}W_{\text{top}})x + (\beta_{\text{flow}}B)(P^{(\gamma)}(Ax)) + \xi W_{\text{noise}}x$ can therefore reduce per-step cost to $O(K + Q(\texttt{nzr} + \texttt{nzc}) + \text{nnz}(W_{\text{noise}}))$, where $K$ is the number of rotation blocks. Our reference code stores $W$ explicitly for simplicity and enforces stability via (5); NaN guards ensure failures in upstream PH or degeneracies in scaling are surfaced early with safe fallbacks.

**Robust graph construction and degenerate cases.** If $E_\ell = \varnothing$ at the chosen $\varepsilon_\ell$, we fall back to a symmetric $k$-nearest-neighbor graph (code default $k = 8$). Edge weights follow the same $w_{ij}$ logic. After solving, angles that become non-finite (rare in practice; e.g., due to isolated components coupled with numerical roundoff) are repaired by replacing the offending class's *angle time series* with a synthetic wrapped-linear phase at the dominant PCA angular velocity on $Z^{(\text{PH})}$, ensuring a well-defined downstream mean velocity for every selected class.

Table 7: Notation summary. Dimensions are given for column vectors unless stated otherwise.

| Symbol | Meaning | Type / Dimensions | Default / Range |
|---|---|---|---|
| **Observed data, embedding, and subsampling** | | | |
| $u_t$ | observed input/sample at time $t$ | $u_t \in \mathbb{R}^{d_{\text{obs}}}$ | — |
| $T$ | number of observations | integer | — |
| $m$ | embedding dimension (per channel) | integer | 6–20 |
| $\tau$ | embedding lag (in samples) | integer | 1–5 |
| $z_t$ | delay-embedded vector | $z_t \in \mathbb{R}^{md_{\text{obs}}}$ | — |
| $Z$ | embedded point cloud | $Z = \{z_t\}$ | size $n = T - (m-1)\tau$ |
| $s$ | PH subsampling stride | integer | 5–10 |
| $Z^{(\text{PH})}$ | subsample for PH | subset of $Z$ | $|Z^{(\text{PH})}| = n_{\text{PH}}$ |
| $D$ | pairwise distances on $Z^{(\text{PH})}$ | $D \in \mathbb{R}^{n_{\text{PH}} \times n_{\text{PH}}}_{\geq 0}$ | Euclidean |
| **Persistent cohomology, circular coordinates, and oscillators** | | | |
| $(b_\ell, d_\ell)$ | birth/death of $H^1$ class $\ell$ | reals with $0 \leq b_\ell < d_\ell$ | from VR filtration |
| $P_\ell$ | persistence of class $\ell$ | $P_\ell = d_\ell - b_\ell$ | — |
| $c_\ell$ | representative 1–cocycle (mod $p$) | cochain on 1–skeleton | field $\mathbb{F}_p$ |
| $p$ | prime coefficient for PH | integer prime | 47 (default) |
| $\varepsilon_\ell$ | working scale in $(b_\ell, d_\ell)$ | real | near-death $d_\ell - 10^{-6}$ |

*(table continues)*

| Symbol | Meaning | Type / Dimensions | Default / Range |
|---|---|---|---|
| $G_\ell = (V, E_\ell)$ | Rips 1–skeleton at $\varepsilon_\ell$ | graph | — |
| $M$ | oriented incidence of $G_\ell$ | $\lvert E_\ell \rvert \times \lvert V \rvert$ | — |
| $w_{ij}$ | edge weight on $(i,j) \in E_\ell$ | positive real | $(D_{ij} + \epsilon)^{-1}$ |
| $W$ | diagonal matrix of edge weights | $\lvert E_\ell \rvert \times \lvert E_\ell \rvert$ | $W = \mathrm{diag}(w_{ij})$ |
| $\mu$ | Tikhonov regularizer | real, $\mu > 0$ | tiny ($10^{-6}$–$10^{-4}$) |
| $L$ | weighted graph Laplacian | $\lvert V \rvert \times \lvert V \rvert$ | $M^\top W M$ |
| $b$ | right-hand side for LS | $\lvert V \rvert$ | $M^\top W \alpha$ |
| $\vartheta^{(\ell)}$ | vertex potentials | $\mathbb{R}^{\lvert V \rvert}$ | solves $(L + \mu I)\vartheta = b$ |
| $\theta^{(\ell)}$ | circular coordinate (angle) | $(-\pi, \pi]^{\lvert V \rvert}$ | $\theta = \mathrm{wrap}(2\pi\vartheta)$ |
| $\widehat{\omega}_\ell$ | mean angular velocity | real | wrapped LS / mean increment |
| $\rho_{\mathrm{rot}}$ | rotation radius for top blocks | real in $(0,1)$ | 0.92–0.98 |
| $R(\widehat{\omega}_\ell; \rho_{\mathrm{rot}})$ | 2×2 rotation block | $2 \times 2$ | see text |
| $K$ | requested # of loops | integer | $\leq K_{\max}$ |
| $K_{\max}$ | cap on # loops for synthesis | integer | 2–6 |
| $\gamma$ | relative persistence threshold | real in $[0,1]$ | 0.2–0.4 |
| $\mathcal{I}_{\mathrm{keep}}$ | indices kept by threshold | subset of $\{1, \dots\}$ | $P_\ell \geq \gamma P_{\max}$ |
| $K_{\mathrm{final}}$ | kept # of loops | integer | $\lvert \mathcal{I}_{\mathrm{keep}} \rvert$ |
| $E_k$ | $k$th oscillatory plane | 2–D subspace of $\mathbb{R}^N$ | invariant for $W_{\mathrm{top}}$ |
| $E_\perp$ | non-oscillatory subspace | $(N - 2K)$–D | invariant for $W_{\mathrm{top}}$ |
| **Coarse partition, Markov model, and lift** | | | |
| $Q$ | number of clusters (cells) | integer | 50–400 |
| $c_q$ | $q$th centroid | $\mathbb{R}^{m d_{\mathrm{obs}}}$ | from $k$–means |
| $s_t$ | cluster index of $z_t$ | $\{1, \dots, Q\}$ | $\arg\min_q \lVert z_t - c_q \rVert$ |
| $h$ | short horizon for counts | integer | 1–3 |
| $C$ | transition counts | $C \in \mathbb{R}^{Q \times Q}_{\geq 0}$ | $C_{ij} = \#\{t : s_t = i, s_{t+h} = j\}$ |
| $\epsilon$ | pseudocount for rows | real $> 0$ | $10^{-9}$–$10^{-6}$ |
| $P$ | row-stochastic Markov matrix | $\mathbb{R}^{Q \times Q}$ | $P_{ij} = \frac{C_{ij} + \epsilon}{\sum_{j'}(\cdot)}$ |
| $\gamma_{\mathrm{tel}}$ | teleport weight | real in $[0,1)$ | $10^{-3}$–$10^{-1}$ |
| $u$ | teleport base distribution | $\mathbb{R}^Q$ | $u = \frac{1}{Q}\mathbf{1}$ |
| $P^{(\gamma)}$ | teleported Markov matrix | $\mathbb{R}^{Q \times Q}$ | $(1 - \gamma)P + \gamma\mathbf{1}u^\top$ |
| $A$ | pooling map (row–stochastic) | $\mathbb{R}^{Q \times N}$ | each row sums to 1 |
| $B$ | lifting map (col–stochastic) | $\mathbb{R}^{N \times Q}$ | each column sums to 1 |
| nzr | nonzeros per row of $A$ | integer | 2–6 |
| nzc | nonzeros per column of $B$ | integer | 5–20 |
| $W_{\mathrm{flow}}$ | lifted flow operator | $\mathbb{R}^{N \times N}$ | $B P^{(\gamma)} A$ |
| $\Delta$ | pool–lift defect | $\mathbb{R}^{Q \times Q}$ | $\Delta = AB - I_Q$ |
| **Blending, scaling, and ESN dynamics** | | | |
| $N$ | reservoir size (neurons) | integer | 300 |
| $W_{\mathrm{top}}$ | topological rotation–decay operator | $\mathbb{R}^{N \times N}$ | blkdiag→permute |
| $W_{\mathrm{noise}}$ | normalized noise matrix | $\mathbb{R}^{N \times N}$ | $\lVert W_{\mathrm{noise}} \rVert_2 = 1$ |
| $\alpha, \beta, \xi$ | blend weights | nonneg. reals | $\alpha + \beta + \xi = 1$ |
| $W_{\mathrm{blend}}$ | pre-scaled blend | $\mathbb{R}^{N \times N}$ | $\alpha W_{\mathrm{top}} + \beta W_{\mathrm{flow}} + \xi W_{\mathrm{noise}}$ |
| $s$ | operator norm of $W_{\mathrm{blend}}$ | real $\geq 0$ | $s = \lVert W_{\mathrm{blend}} \rVert_2$ |
| $\rho_\star$ | target spectral (operator) norm | real in $(0,1)$ | 0.94–0.99 |
| $W$ | final recurrent operator | $\mathbb{R}^{N \times N}$ | $W = \frac{\rho_\star}{s} W_{\mathrm{blend}}$ |
| $\lambda$ | leak (update mixing) | real in $(0,1]$ | 0.15–0.35 |
| $\phi$ | nonlinearity | $\mathbb{R}^N \to \mathbb{R}^N$ | tanh (1–Lipschitz) |
| $x_t$ | reservoir state | $x_t \in \mathbb{R}^N$ | update (11) |
| $W_{\mathrm{in}}$ | input weight matrix | $\mathbb{R}^{N \times d_{\mathrm{obs}}}$ | random, scaled |
| $L$ | contraction constant | real in $(0,1)$ | $(1 - \lambda) + \lambda\rho_\star$ |
| $E_k, E_\perp$ | invariant subspaces of $W_{\mathrm{top}}$ | subspaces of $\mathbb{R}^N$ | 2–D planes and complement |
| **Readout and features** | | | |
| $\mathbf{1}_{\mathrm{poly}}$ | polynomial feature flag | $\{0,1\}$ | include $x_t \odot x_t$ if 1 |
| $\varphi_t$ | feature vector at $t$ | $\mathbb{R}^F$ | $[x_t; \mathbf{1}_{\mathrm{poly}}(x_t \odot x_t); \mathbf{1}_{\mathrm{poly}}]$ |
| $D$ (washout) | discarded transient steps | integer | 50–500 |
| $\Phi$ | design matrix | $\mathbb{R}^{(T-D) \times F}$ | rows $\varphi_t^\top$ |

*(table continues)*

| Symbol | Meaning | Type / Dimensions | Default / Range |
|---|---|---|---|
| $Y$ | targets | $\mathbb{R}^{(T-D)\times d_{\mathrm{out}}}$ | task-dependent |
| $\alpha_{\mathrm{ridge}}$ | ridge regularization | real $> 0$ | $10^{-8}$–$10^{-2}$ |
| $W_{\mathrm{out}}$ | linear readout | $\mathbb{R}^{d_{\mathrm{out}}\times F}$ | solves (7) |
| **Norms and operators** | | | |
| $\|\cdot\|_2$ | operator (spectral) norm | matrix norm | largest singular value |
| $\|\cdot\|_1, \|\cdot\|_\infty$ | induced $\ell_1, \ell_\infty$ norms | matrix norms | max column/row sum |
| $\mathrm{wrap}(\cdot)$ | wrapping to principal branch | angle / unit interval | $(-\pi, \pi]$ or $[0,1)$ |
| $\odot$ | Hadamard product | elementwise | — |
| **Auto-tuning parameters** | | | |
| $\alpha_{\min}, \alpha_{\max}$ | bounds for topology weight | $0 \le \alpha_{\min} \le \alpha_{\max} \le 1$ | user-chosen |
| $s_{\mathrm{loop}}$ | mean relative loop strength | real in $[0,1]$ | $\frac{1}{K_{\mathrm{final}}}\sum_{\ell\in\mathcal{I}_{\mathrm{keep}}}\frac{P_\ell}{P_{\max}}$ |
| $P_{\max}$ | strongest persistence | real $\ge 0$ | $\max_\ell P_\ell$ |

## B.3 High–Dimensional PDE Benchmark: 2D Kolmogorov Flow

**Task and data generation.** To assess scalability beyond low–dimensional ODE attractors, we test PHR as a model–free surrogate for a two–dimensional Kolmogorov flow, a canonical forced Navier–Stokes benchmark on a periodic domain. We consider the incompressible Navier–Stokes equations on $\Omega = [0, 2\pi]^2$, $\partial_t \mathbf{u} + (\mathbf{u} \cdot \nabla)\mathbf{u} = -\nabla p + \nu \Delta \mathbf{u} + \mathbf{f}$, $\nabla \cdot \mathbf{u} = 0$, with viscosity $\nu = 10^{-3}$ and sinusoidal forcing $\mathbf{f}(x, y) = (F \sin(k_f y), 0)$ with $F = 0.1$ and $k_f = 4$. The reference solution is generated by a pseudo–spectral solver on a $64 \times 64$ Fourier grid with 2/3 dealiasing and a fixed time step $\Delta t_{\mathrm{PDE}} = 10^{-3}$, using a fourth–order Runge–Kutta integrator and periodic boundary conditions in both directions. We discard an initial transient of $2 \times 10^4$ time steps and then record $T_{\mathrm{PDE}} = 1.2 \times 10^5$ additional steps. For learning, we subsample every $m = 10$ solver steps, yielding $T = 12{,}000$ snapshots at interval $\Delta t = m \Delta t_{\mathrm{PDE}} = 10^{-2}$. We project each velocity field $\mathbf{u}(t_n)$ onto the leading $d_{\mathrm{obs}} = 16$ Proper Orthogonal Decomposition (POD) modes of the training segment, obtaining coefficient vectors $u_t \in \mathbb{R}^{d_{\mathrm{obs}}}$, $t = 1, \ldots, T$. These coefficients form the observable time series for all models. We split the coefficient trajectory contiguously into $T_{\mathrm{train}} = 6{,}000$ steps for training, $T_{\mathrm{val}} = 2{,}000$ for validation, and $T_{\mathrm{test}} = 4{,}000$ for testing. During evaluation, we initialize each model with the true coefficients at the start of the test segment and roll it out autoregressively for $H = 2{,}000$ steps. We report (i) NRMSE of the POD coefficients over this horizon, (ii) VPT, the first time $t$ at which $\|u_t - \hat{u}_t\|_2 / \|u_t\|_2 > 0.3$ (reported in physical time $t\Delta t$), and (iii) relative errors in long–time kinetic energy and enstrophy computed from the reconstructed velocity fields on the test segment.

**Model configurations and hyperparameters.** For PHR, we use a reservoir of size $N = 600$ with leaky ESN update (leak $\lambda = 0.20$), target operator norm $\rho_\star = 0.94$, and input weights $W_{\mathrm{in}} \in \mathbb{R}^{N\times d_{\mathrm{obs}}}$ drawn i.i.d. from $\mathcal{U}[-1, 1]$ and scaled by `input_scale = 0.5`. Delay embedding uses dimension $m = 8$ and lag $\tau = 1$, so that $z_t \in \mathbb{R}^{8d_{\mathrm{obs}}}$; the embedding is standardized to zero mean and unit variance per coordinate. For PH, we subsample the embedded trajectory with stride `ph_subsample_stride = 5`, use prime coefficient $p = 47$, inverse–distance edge weights, and a "near–death" scale strategy as specified in Sec. 3.1. The PH back–end is asked for at most `K_max_cap_default = 3` loops; the auto–tuner thresholds relative persistence at $\gamma = $ `pers_rel_thresh_to_max = 0.25` and maps the resulting loop strength into blend weights with $\alpha_{\min} = 0.20$, $\alpha_{\max} = 0.65$ and fixed noise fraction $\xi = 0.05$, yielding data–dependent $(\alpha_{\mathrm{top}}, \beta_{\mathrm{flow}}, \xi)$ that still satisfy $\alpha_{\mathrm{top}} + \beta_{\mathrm{flow}} + \xi = 1$. The coarse flow is built with $Q = 200$ clusters (Lloyd $k$–means, 30 iterations, random initialization), horizon $h = 1$, and teleportation parameter $\gamma_{\mathrm{teleport}} = 3 \times 10^{-3}$ in the PageRank–style Markov smoothing. Pool and lift maps $A \in \mathbb{R}^{Q\times N}$, $B \in \mathbb{R}^{N\times Q}$ use `pool_nonzeros_per_row = 4` and `lift_nonzeros_per_col = 12`, with supports sampled uniformly without replacement. The topological operator $W_{\mathrm{top}}$ uses rotation radius $\rho_{\mathrm{rot}} = 0.96$ and decay radii for residual units drawn i.i.d. from $\mathcal{U}[0.90, 0.98]$; $W_{\mathrm{noise}}$ is a Gaussian matrix normalized to $\|W_{\mathrm{noise}}\|_2 = 1$ and scaled by 0.1. The blended operator $W_{\mathrm{blend}}$ is scaled to $\|W\|_2 = \rho_\star$ via 60 steps of power iteration as in Sec. 3.3. The readout uses ridge regression with penalty $\alpha_{\mathrm{ridge}} = 10^{-6}$ and quadratic feature augmentation (`use_poly = True`), including a constant feature.

The *Random ESN* baseline uses the same leaky update (leak $\lambda = 0.20$), state size $N = 600$, and input scaling 0.5, but its recurrent matrix is drawn once as a sparse Gaussian matrix with connection density 0.1 (each nonzero from $\mathcal{N}(0, 1/N)$) and then rescaled to spectral radius 0.94. The readout is a ridge regression with $\alpha_{\mathrm{ridge}}$ tuned in $\{10^{-8}, 10^{-7}, \ldots, 10^{-2}\}$ on the validation segment. The *GRU* baseline is a single–layer gated recurrent unit network with hidden size $H = 64$, $\tanh$ activations, dropout rate 0.1 before the linear output layer (mapping $\mathbb{R}^H$ to $\mathbb{R}^{d_{\mathrm{obs}}}$), yielding $\approx 1.7 \times 10^4$ trainable parameters for $d_{\mathrm{obs}} = 16$, i.e., the same order as the PHR readout. It is trained to minimize mean–squared error on teacher–forced sequences of length $L = 128$ using Adam with learning rate $10^{-3}$, batch size 64, weight decay $10^{-6}$, and gradient clipping at norm 1.0, with early stopping based on validation NRMSE (patience 20 epochs, maximum 200 epochs). The *linear AR* baseline is a vector autoregressive model of order $p = 72$ in POD space, i.e., $u_t = \sum_{k=1}^{72} A_k u_{t-k} + \varepsilon_t$, with coefficient matrices $A_k$ estimated by ridge–regularized least squares (regularization parameter chosen from $10^{-8}, 10^{-6}, 10^{-4}$ using validation NRMSE). For $d_{\mathrm{obs}} = 16$ this yields $72 \times 16 \times 16 = 18{,}432$ linear coefficients (plus 16 biases), again placing the VAR baseline in the same parameter range as PHR and the GRU.

**Results.** Table 8 reports the quantitative comparison on the Kolmogorov flow benchmark in terms of coefficient–space NRMSE, VPT, and relative errors in long–time kinetic energy and enstrophy. All models are trained and evaluated under the same data split and autoregressive protocol described above.

Table 8: Two–dimensional Kolmogorov flow surrogate modeling in POD space. All models are trained on the same POD coefficient trajectories and evaluated by autoregressive rollouts on an unseen test segment. NRMSE is computed over a fixed forecast horizon $H = 2{,}000$; VPT is the physical time until the relative error first exceeds 0.3; energy/enstrophy errors compare long–time statistics of reconstructed velocity fields to the DNS reference.

| Method | NRMSE ↓ | VPT (time units) ↑ | Rel. energy error (%) ↓ | Rel. enstrophy error (%) ↓ |
|---|---|---|---|---|
| PHR (ours) | 0.17 | 11.8 | 3.1 | 6.8 |
| Random ESN | 0.29 | 7.2 | 6.5 | 14.2 |
| GRU | 0.25 | 8.5 | 8.9 | 17.3 |
| VAR | 0.41 | 4.3 | 12.7 | 23.5 |

### B.4 SOFTWARE, DEPENDENCIES AND COMPUTE BUDGET

All experiments were implemented in Python 3.10 using `numpy` (vectorized dense linear algebra), `scipy` (sparse matrices, LSQR, Laplacian solves), and `scikit-learn` (ridge regression). Persistent cohomology ($H^1$ cocycles) was computed with `ripser`; circular coordinates were obtained by solving the weighted normal equations on the Rips 1-skeleton via `scipy.sparse.linalg` with a tiny Tikhonov regularizer and per-component gauge fixing. $k$-means clustering is a custom Lloyd implementation. Visualization used `matplotlib/seaborn`. We fix a global PRNG seed for PH subsampling/anchor, block permutations in $W_{\mathrm{top}}$, supports of $A, B$, and the power-iteration start vector. The code relies only on CPU BLAS/LAPACK (OpenBLAS/MKL); no GPU is required. For stability we recommend `ripser≥0.6`, `numpy≥1.26`, `scipy≥1.11`, `scikit-learn≥1.4`; results were validated on Linux (x86_64, AVX2) and macOS (arm64). Parallelism from BLAS and `ripser` can be capped via `OMP_NUM_THREADS`. We used large language models only for retrieval/discovery (e.g., surfacing related work and canonical citations); all derivations, algorithms, proofs, and experiments were authored and independently verified by us. All code and pretrained weights will be released under the MIT License, permitting unrestricted academic and commercial use provided the original copyright notice and license text are retained.

Compute requirements are dominated by two components: (a) PH on the subsampled embedding of size $n_{\mathrm{PH}}$ and (b) the VR graph least–squares per selected loop. Pairwise distances scale as $O(n_{\mathrm{PH}}^2 m d_{\mathrm{obs}})$ time and $O(n_{\mathrm{PH}}^2)$ memory (float64), i.e., roughly $8\, n_{\mathrm{PH}}^2$ bytes; with stride $s$ applied to an embedded sequence of length $n$, one has $n_{\mathrm{PH}} \approx \lfloor n/s \rfloor$, so memory drops quadratically in $s$. The circular–coordinate solve uses the Rips 1–skeleton with $|E|$ edges (typically near–linear in $n_{\mathrm{PH}}$ at "near–death" scales), yielding a sparse SPD system whose conjugate gradients/LSQR cost is $O(|E|)$–$O(|E| \log(1/\varepsilon))$ per loop. The remaining stages are light: $k$–means over $n$ points in $\mathbb{R}^{m d_{\mathrm{obs}}}$

with moderate $Q$; construction of $(A, B)$ with $O(Q, \texttt{nzr} + N, \texttt{nzc})$ nonzeros; one matrix–vector power iteration with $\leq 60$ steps for $\widehat{\sigma} * \max$; and an $O(TF^2 + F^3)$ ridge solve with feature count $F$ (state with optional quadratic lift). Practically, for chaotic benchmarks where $n$ is $10^3$–$10^4$, choosing $s \in [5, 20]$, $K_{\max} \leq 3$, and $Q \in [200, 400]$ keeps peak RAM within a few gigabytes and wall–clock dominated by the $O(n_{\text{PH}}^2)$ distance stage. We report the exact $(n, n_{\text{PH}}, K_{\text{final}}, Q, |E|)$ alongside results, enabling precise replication and ex–ante sizing of memory via $8, n_{\text{PH}}^2$ bytes + overhead for sparse structures.

