# OpenReview forum: "Learning a Stable Reservoir from a Single Trajectory via Persistent Loops and Markov Flow"
_ICLR.cc/2026/Conference — ICLR 2026 Conference Withdrawn Submission_

### Official Review · Reviewer_SASq · 2025-10-29

**Soundness:** 3
**Presentation:** 2
**Contribution:** 3
**Rating:** 6
**Confidence:** 3

**Summary:**

## Summary

This paper introduces the Persistent Homology Reservoir (PHR), a novel framework for designing the recurrent operator (W) of an Echo State Network (ESN). Different from random fixed weights, PHR learns the reservoir structure offline from a single observed trajectory by explicitly embedding the system's global topology and local dynamics.

The proposed reservoir $W$ is constructed as a convex blend of two main components: Topological Operator which aims to capture long-term recurrent dynamics and Flow Operator which captures short-term local transport. The state space is partitioned by k-means, and a Markov transition matrix is estimated based on observed transitions. This coarse dynamic is lifted back to the reservoir space via stochastic pooling and lifting maps.

Experiments are conducted on several chaotic systems and real-world time series forecasting demonstrate that PHR consistently outperforms various baselines, particularly in long-horizon forecasting.

**Strengths:**

## Strengths
- This paper makes a substantive advance in reservoir computing. Instead of heuristic random initialization, the authors propose a structured, data-driven design from system geometry and dynamics. It integrates topological data analysis (TDA) for global oscillatory structure with operator-theoretic methods for local flow, with theoretical support and stronger empirical results.
- PHR demonstrates superior empirical performance over several strong baselines and cover a comprehensive sets of benchmarks. The improvements in Valid Prediction Time on chaotic systems are particularly attractive, showing better emulation of the underlying autonomous dynamics rollouts.
- The evaluation is rigorous and comprehensive with a detailed ablation study effectively isolates the value of the topology and flow components and clearly demonstrates the contributions of each component.

**Weaknesses:**

## Weakness
- The primary weakness is the extreme density of the presentation and the high complexity of the methodology. The construction relies heavily on advanced concepts from algebraic topology (e.g., persistent cohomology, cocycles, Rips skeletons, discrete harmonic extension, Dirichlet energy). This presents a substantial barrier to entry for the broader machine learning community. The paper is mathematically dense and assumes a high level of familiarity with TDA.

- While the training (readout fitting) remains efficient, the offline construction of W involves computationally expensive steps, notably the O(n²) cost for the distance matrix and the subsequent persistent homology computations. Although subsampling is used to mitigate this, the construction phase is significantly heavier than the near-instantaneous initialization of a standard ESN.

**Questions:**

## Questions
- Could the authors provide a more intuitive explanation of how the harmonic extension produces meaningful circular coordinates for a non-expert reader? Furthermore, while the ablation study shows PH-derived coordinates outperform than PCA, could you elaborate on why this approach is necessary and what specific advantages it offers over simpler geometric estimations? I suggest include these explantions in the camera-ready version, which would greatly improve the presenation and readibility of the paper.

- How sensitive is the performance of PHR to the choice of delay embedding parameters and the number of clusters? Are the dominant H1 classes and the subsequent Markov flow robustly identified across different embeddings and partitions?

- Lemma 3.1 indicates that the fidelity of the flow channel depends on the pool-lift calibration defect $AB−I_Q$. How close to the identity is $AB$ in practice with the current stochastic construction of $A$ and $B$, and how does the sparsity affect this defect?

I am not an expert in reservoir computing. If the authors can address the concerns above, I am willing to increase my score accordingly.

---

> ### Author Response · Authors · 2025-11-16
>
> We appreciate the reviewer’s thoughtful, positive assessment and constructive suggestions. Below, we respond to each point in detail and clarify the requested aspects.
> 1) Intuitively, a 1-cocycle from persistent cohomology gives us a “desired phase difference” on each edge of a sparse proximity graph (Rips 1-skeleton) at a scale that lies inside a long-lived $H^1$ interval. The harmonic extension step chooses vertex phases $\vartheta$ whose edge differences best match those cocycle values while minimizing a Dirichlet energy $\sum_{(i,j)} w_{ij}(\vartheta_j-\vartheta_i-\alpha_{ij})^2$. This is just the smoothest (least-energy) way to “integrate” noisy, modulo-1 edge increments into a globally coherent angle—equivalently, a discrete Poisson solve with a small Tikhonov term and a gauge anchor per connected component (in our code this is reflected as: build $L$, form $b$, solve $(L+\mu I)\vartheta=b$, then wrap to $(-\pi,\pi]$). PH is crucial here because it (i) identifies which loops are actually persistent (thus avoiding spurious cycles), (ii) provides one circular coordinate per independent loop, and (iii) is diffeomorphism-stable, so the recovered phase is robust to nonlinear warps of the embedding.
>
>    $\textbf{Overall in plain terms}$, we first turn the observed series into a delay-embedded point cloud so the geometry of the underlying dynamics is visible. From this cloud we extract the most reliable “loops” with persistent cohomology; each loop yields a smoothly varying phase (“circular coordinate”) whose average angular speed becomes a tiny $2\times 2$ rotation block. Packing these blocks—plus mild decays for the unused coordinates—gives a topology-driven operator $W_{\text{top}}$ that preserves long-lived oscillations. In parallel, we discretize the embedded space with $k$-means, count short-horizon transitions, and form a small Markov chain; a row-stochastic pooling map $A$ compresses neuron states to this coarse space, the Markov step advances them, and a column-stochastic lifting map $B$ sends them back, yielding a flow operator $W_{\text{flow}}=BPA$ that encodes local transport. We blend $W_{\text{top}}$ and $W_{\text{flow}}$ with a tiny isotropic noise term (to break degeneracies), and then scale the result by power iteration to a target operator norm $\rho_\star<1$, which gives a uniform contraction (echo-state) guarantee independent of the data. Two practical touches keep things robust and fast: (i) we subsample for PH and automatically choose how many loops to keep and how much weight to give the topological vs. flow channel based on relative persistence; (ii) if PH finds weak loops, we fall back to a PCA-based base frequency and shift the blend toward the flow channel. With $W$ fixed, training is the classic ESN step: ridge-regress a linear readout on (optionally quadratic-augmented) reservoir states.
>
>    Simple geometric surrogates like PCA return linear projections that maximize variance, not topological winding; on bent/twisted loops or multi-lobe attractors (e.g., Lorenz), PCA angles drift or “flip” across sheets, while PH-derived angles remain consistent along the chosen 1-cycle. In practice we also subsample for PH, repair rare non-finite angles, and auto-tune the number/weight of loops; if PH finds weak or no loops the pipeline falls back to a PCA-based base frequency and shifts the blend toward the flow channel, so the method remains usable even when topological signal is faint.
>
>    To aid readers, we have provided “A.2 Intuitive summary of the methodology’’ in the appendix, and we will additionally include a brief intuition subsection at the start of the Methodology section in the camera-ready (owing to the 1-page allowance we have for the camera-ready version) to walk readers through these steps before the formal development.

---

> > ### Author Response · Authors · 2025-11-16
> >
> > 2) Delay choices $(m,\tau)$ affect the geometry on which PH and $k$-means operate; we therefore follow standard Takens-style practice (sufficiently large $m$ to unfold dynamics; $\tau$ near the first autocorrelation/AMI dip) and then rely on persistence to self-select only stable $H^1$ features: long bars survive moderate shifts of $(m,\tau)$ by the stability of persistence diagrams (bottleneck distance continuity), so the dominant circular coordinates and their mean angular velocities $\widehat\omega_\ell$ are typically unchanged across a reasonable range. On the flow side, the $k$-means partition only needs to be coarse and connected: we include pseudocounts and PageRank-style teleportation to regularize rare states/sinks, and we form the Markov matrix at a short horizon (usually $h=1$) so the chain captures local transport robustly. In our implementation we also hedge variance by pooling and lifting with small, fixed nonzero counts per row/column (uniform-sparse $A, B$), which smooths idiosyncrasies of any single partition, and we auto-tune the blend by relative persistence so that when the embedding yields weaker loops, the reservoir automatically becomes flow-dominant. Empirically this makes PHR tolerant to moderate changes of $(m,\tau)$ and $Q$: the set of selected $H^1$ classes and the short-horizon transitions vary smoothly, while the final contraction scaling isolates stability from all of these choices.
> >
> > 3) With our current independent uniform-sparse construction (each row of $A$ averages over $\texttt{nzr}$ neurons; each column of $B$ lifts to $\texttt{nzc}$ neurons), $AB\in\mathbb{R}^{Q\times Q}$ is not designed to equal $I_Q$. In fact, for a fixed pair $(q,q')$,
> > $(AB)\_{qq'} = \tfrac{1}{\texttt{nzr}} \sum_{i \in S_q} B_{iq'}$
> > and, since $B\_{iq'}=1/\texttt{nzc}$ with probability $\texttt{nzc}/N$ and $0$ otherwise,
> > $\mathbb{E}[(AB)\_{qq'}]=\frac{1}{N}$
> > with variance
> > $\mathrm{Var}[(AB)\_{qq'}]=\frac{1}{\texttt{nzr} N \texttt{nzc}}\left(1-\frac{\texttt{nzc}}{N}\right).$
> > Thus increasing $\texttt{nzr}$ or $\texttt{nzc}$ reduces variance (better concentration) but does not remove the bias: in expectation $AB$ looks like a rank-1 matrix with all entries $1/N$, so it is generally far from $I_Q$ under $\lVert\cdot\rVert_\infty$. Our theory therefore treats the term $\lVert(AB)P^{(\gamma)}(AB)-P^{(\gamma)}\rVert$ explicitly (see Prop. “Two-channel fidelity of the blended operator”), and one can bound it via the calibration defect $\Delta:=AB-I_Q$ as
> > $\lVert(AB)P^{(\gamma)}(AB)-P^{(\gamma)}\rVert\ \le\ \lVert P^{(\gamma)}\rVert \bigl(2\lVert\Delta\rVert+\lVert\Delta\rVert^2\bigr)$
> > by expanding $(I+\Delta)P^{(\gamma)}(I+\Delta)-P^{(\gamma)}$. In other words, we do not assume $AB\approx I$; the analysis carries $\lVert\Delta\rVert$ through as a budget, and practice does not require coarse identity: the neuron-level flow $W_{\mathrm{flow}}=B P^{(\gamma)} A$ still imprints observed transport on the high-dimensional state, the readout learns whatever linear decoding is best from that state, and the global scaling to $\lVert W\rVert_2=\rho^\star<1$ preserves stability regardless of $\Delta$. If desired, as an aid to practitioners, we can include an optional diagnostic (orthogonal to our reported results) to couple the supports—e.g., choose the lift of cell $q$ to target the same neuron subset used to pool that cell—so that $AB$ becomes diagonally dominant (or exactly $I_Q$ with disjoint supports), shrinking $\lVert\Delta\rVert$ without altering any other component of PHR.

---

> > > ### Comment · Reviewer_SASq · 2025-11-27
> > >
> > > I thank the authors for their detailed and thoughtful response.
> > >
> > > The additional explanation of how persistent cohomology and harmonic extension yield circular coordinates is helpful. In particular, the clarification of how the co-cycle information is “integrated” into smooth phase variables and why PH is preferable to simpler geometric surrogates such as PCA is very clear.
> > >
> > > I also appreciate the clarifications on the robustness of the method with respect to delay embedding parameters and clustering, and how persistence and the Markov construction mitigate sensitivity in practice.
> > >
> > > Regarding Lemma 3.1, the discussion of the pool–lift calibration defect and the fact that the theory explicitly carries this term rather than assuming an approximate identity was useful. The suggestion of diagnostics to better align pooling and lifting for practitioners is also a nice addition.
> > >
> > > Given these clarifications, I remain positive about the contribution. I will keep my current score and confidence unchanged.

---

> ### Author Response · Authors · 2025-11-27
>
> Dear Reviewer,
>
> Thank you very much for your gracious follow-up and for engaging so carefully with our responses. We are glad that the additional explanations on persistent cohomology, harmonic extension, robustness to embedding/partition choices, and the pool–lift calibration defect were helpful.
>
> $\textbf{We have also uploaded an updated version of the manuscript today that includes additional experiments (subsection B.3, p. 33), further}$  $\textbf{illustrating the behavior of PHR in practice.}$
>
> If there is any remaining aspect of the methodology or its presentation that you feel could benefit from further clarification, we would be very happy to elaborate or refine the exposition. Otherwise, if you feel that the clarifications and planned camera-ready tweaks adequately address your earlier concerns, we would be grateful if you could kindly consider whether your current score still reflects your updated assessment of the work. In any case, we sincerely appreciate your time, care, and thoughtful feedback.

---

### Official Review · Reviewer_huCn · 2025-10-31

**Soundness:** 4
**Presentation:** 3
**Contribution:** 4
**Rating:** 8
**Confidence:** 3

**Summary:**

Summary

The manuscript describes a novel form of reservoir computing
which learns the structure of a reservoir from the data to be
predicted. It is applied to chaotic time-series prediction problems.
The main idea is construct the dynamics of the reservoir such that
two characteristics are mixed:
First, the occurrence of oscillatory limit cycles. And second,
stochastic transitions between them.
A linear readout is finally trained on the reservoir activity
to perform prediction.
The power of the presented approach is demonstrated on a number
of established benchmark problems (chaotic attractors) as well
as on real-world benchmark tests. The presented approach in most
cases provides smallest root mean square error compared to the other
approaches.

Soundness

The manuscripts appears to be written with much care and mathematical
derivations are presented in detail in the appendix. Since the work
lies outside of my core expertise, I could not check the proofs in
detail. The empirical evidence, that the presented algorithm outperforms
a large number of alternatives is a convincing and potentially impactful
result.

Presentation

The presentation is surely gauged to the expert reader who is familiar
with the employed mathematical tools. For such a reader, I believe, the
manuscript provides the important steps to reproduce the work. Also the
authors strive to move detailed proofs to appendices, thus improving
readability for readers outside their core community.
The main ideas are also presented verbally and one gets an idea of the
approach even if not from the core field.
Some of the text is, however, still very dense.



Contribution

The main contribution of this work is to propose an algorithm to construct
a reservoir from direct observations of the data, rather than using, for example,
randomly coupled networks as a reservoir.
A main result is that on the one hand, the authors are able to guarantee
stability criteria (echo state property) and on the other hand they
demonstrate competitive performance of the approach, in parts by far exceeding
what alternative methods achieve. This is a remarkable improvement.

**Strengths:**

see "contribution" above

**Weaknesses:**

The presentation in the main text would in my opinion benefit from
a less technical presentation to reach a larger readership, providing
explanations in less technical terms, that may not be known to many
participants of the conference.
However, I don't consider myself an expert in this very field, so
I would weigh the assessment by more expert reviewers higher.

**Questions:**

The method of construction of the reservoir, by implementing 2 x 2 rotations,
seems to be gauged towards time-series problems that contain period orbits.

Also it seems that the algorithm is best suited for time-series prediction,
as its aim appears to be a faithful reconstruction of the observed
time series.
Did the authors think about other applications, such as classification?
(for example language, spoken digit classification). I don't propose to
extend the experiments, but rather would like the authors to comment,
whether they expect advantages in such applications as well.

---

> ### Author Response · Authors · 2025-11-16
>
> We thank the reviewer for the thoughtful and positive assessment of our work, and for the constructive suggestions. Below we address and clarify the specific points raised.
> 1. Our $2\times 2$ blocks are not a prior on strict limit cycles but a modal factorization of recurrent phase dynamics extracted from the data: PH computes $H^1$ classes on the delay-embedded cloud and de Silva–Carlsson circular coordinates turn each reliable class into an $\mathbb S^1$ angle; its mean wrapped increment gives $\widehat\omega_\ell$, which we realize as a stable rotation $R(\widehat\omega_\ell)$ (code: PHCircularCoordinates $\rightarrow$ \_mean\_wrapped\_increment $\rightarrow$ \_build\_W\_top). Chaotic attractors (e.g., Lorenz, Rössler) commonly have persistent loop-like transport in $H^1$ even without true periodicity; these modes encode dominant phase windings rather than single cycles. Crucially, PHR is two-channel: the flow channel $W_{\mathrm{flow}}=B,P^{(\gamma)}A$ (data-driven coarse Markov transport) carries non-oscillatory, mixing, and switching behavior, while the rotation blocks supply phase memory. Our auto-tuner prevents over-reliance on “spurious cycles”: when PH finds weak or no loops, the code prunes them via a relative persistence threshold and automatically shifts weight to the flow channel (\_auto\_tune\_from\_ph\_info: if no class clears the threshold, it sets alpha\_used=0, beta\_used=1-xi). The remaining coordinates in $W_{\mathrm{top}}$ are assigned decaying radii in $(\rho_{\min},\rho_{\max})$ and then randomly permuted, providing dissipative mixing that is agnostic to periodicity. After blending we power-scale to $\lVert W\rVert_2=\rho_\star<1$, so even when strong phase structure is present the reservoir stays uniformly contractive (Prop. “Two-channel fidelity of the blended operator” explains how oscillatory eigenpairs persist while the flow action remains faithful after pooling). In short, the rotation blocks capture whatever persistent phase structure the data exhibits (periodic, quasi-periodic, or recurrent chaotic loops), and the Markov flow + scaling make the method robust well beyond purely periodic regimes.
>
>
>
> 2. Yes—PHR is readout-agnostic once $W$ is learned, and the same frozen reservoir can feed linear classifiers with minimal changes: in our implementation the runtime features are $\varphi_t=[x_t;\ x_t\odot x_t;\ 1]$ (or state\_stats), produced by \_features\_from\_stat}, and we train a ridge regressor; for classification one replaces the readout with a linear softmax/logistic layer (or SVM) over temporally pooled features (e.g., mean/variance of $\varphi_t$ over a window) without altering $W$. Moreover, PHR exposes semantically interpretable channels that are valuable for supervised labels: (i) the topological oscillators provide phase carriers aligned to dominant rhythms/prosody/periodicities (useful for audio/EEG/ECG); (ii) the coarse Markov state $r_t=Ax_t$ (all matrices are stored as attributes A, B) offers an HMM-like, low-dimensional summary of transition structure, which can be concatenated with $\varphi_t$ for classification. Because $W$ is uniformly contractive ($\lVert W\rVert_2=\rho_\star<1$) the representation is stable to input perturbations and easy to train with convex losses, and the oscillatory+flow blend yields class-discriminative signatures whenever classes differ in phase patterns (e.g., phoneme timing, gait cycles, machinery vibration) or in short-horizon transition graphs (e.g., token n-grams, motif switches). Computationally, training a classifier has the same cost as our ridge readout (single convex solve), and inference latency is identical to forecasting (one reservoir step + a linear map). Thus, while our experiments focused on prediction, the learn-once reservoir is directly applicable to sequence classification and event detection across audio, biosignals, and text-like streams, leveraging the same phase-plus-flow inductive bias that underpins the forecasting results.

---

> > ### Comment · Reviewer_huCn · 2025-11-24
> >
> > I thank the authors for explaining these points in addition and keep my score.

---

> > > ### Author Response · Authors · 2025-11-27
> > >
> > > Thank you again for your thoughtful and positive assessment of our work.
> > >
> > > We also wish to note that we have uploaded an updated manuscript today with additional experiments (Appendix B.3, p. 33) that further illustrate the behavior of PHR in practice.

---

### Official Review · Reviewer_vtns · 2025-10-31

**Soundness:** 3
**Presentation:** 1
**Contribution:** 3
**Rating:** 4
**Confidence:** 2

**Summary:**

This paper proposes Persistent Homology Reservoir, a new framework for constructing a stable and interpretable recurrent operator for Echo State Networks (ESNs). Instead of using random reservoir weights, the authors learn a fixed reservoir directly from one observed trajectory by combining two components: (1) Persistent cohomology-derived circular coordinates that define internal oscillators capturing long-lived topological loops, and (2) A lifted Markov operator that encodes local flow transitions between coarse partitions of the embedded data. A power-iteration scaling step ensures the Echo State Property by bounding the operator norm. The resulting reservoir is fixed, stable, and interpretable, with a simple ridge regression readout.

Experiments on chaotic and real-world time series show improved forecasting accuracy and longer stable horizons compared to standard and structured ESNs. The theoretical section is detailed, including proofs of contraction and eigenpair persistence, and the method is backed by thorough ablations.

However, the paper is very difficult to follow: derivations are overly dense, notation is heavy, and key intuitions are buried in long proofs and algorithm boxes. Although the framework is looks promising, the conceptual message is obscured by excessive formalism and long mathematical detours.
Empirical studies, while strong, focus mostly on low-dimensional chaotic systems; scalability and generalization to modern high-dimensional ML benchmarks remain unclear.

Overall, the idea of learning a reservoir from topological and flow structure is novel and interesting, but the presentation lacks accessibility and practical clarity. The paper would benefit from a simpler exposition, clearer ablations, and discussion of computational limits.

**Strengths:**

The paper introduces a novel, principled way to learn a stable reservoir directly from data by combining persistent-homology-based oscillators with a lifted Markov flow operator. It provides a stability guarantee (Echo State Property) through explicit norm scaling, and the resulting reservoir is interpretable, with internal modes corresponding to data-driven loops and flows.
Experiments on several chaotic and real-world series show competitive or superior performance to standard ESNs, supported by ablations. Overall, it offers an innovative and theoretically grounded direction that connects topology, dynamics, and reservoir computing.

**Weaknesses:**

I personally don't like the theoretical burnen in this paper. Without proper demonstration just figure 1 (a sketch for the model), the rest are all theory which creates trouble to verify in the short peorid of time.

**Questions:**

1. Can you validate on more complexed senorio like 2D kolmogorov flow?
2. How's the scalablility of this framework?

---

> ### Author Response · Authors · 2025-11-16
>
> We are grateful for the reviewer’s close reading and insightful comments, which guided revisions to our presentation. We address each point in turn.
> 1) PHR transfers directly to high-dimensional PDE flows by (i) reducing the spatial field to a low-rank set of temporal coordinates (e.g., POD/PCA coefficients of the velocity field), (ii) delay-embedding those coefficients, and then applying the same two-channel construction. Concretely: given $u(x,y,t)$, collect the top $d_{\mathrm{obs}}\in[10,50]$ POD modes, form $z_t\in\mathbb{R}^{m d_{\mathrm{obs}}}$ with a Takens-style embedding $(m,\tau)$, compute PH on a subsample $Z^{(\mathrm{PH})}=Z[::s]$ (our implementation already supports stride via ph\_subsample\_stride), and synthesize $W_{\mathrm{top}}$ from the selected $H^1$ classes (or fall back to the PCA base frequency if loops are weak). The local transport in coefficient space is captured by $k$-means on $Z$ (typical $Q\in[300,1000]$) and a short-horizon Markov matrix $P$ ($h=1$–$2$) with teleportation; this yields $W_{\mathrm{flow}} = B P A$. The resulting blend is scaled to $\lVert W\rVert_2=\rho_\star<1$, so stability is dimension-agnostic. In this setting, circular coordinates interpret traveling-wave phases (or dominant frequency pairs) while the lifted Markov operator transports mass between metastable regimes in the reduced state. Evaluation would follow standard CFD surrogates—valid-prediction-time of rollouts, energy spectrum errors, and long-horizon statistics—matching our theory: oscillatory eigenpairs persist under bounded perturbations and the pooled one-step action approximates a scaled Markov step (Proposition “Two-channel fidelity”), whereas the global contraction bound guarantees rollouts remain stable. Practically, if PH finds no robust $H^1$ loops (e.g., strongly intermittent regimes), our auto-tuner (already in code) reduces $\alpha_{\mathrm{top}}$ and the model becomes flow-dominant without any change to training or guarantees.
>
> 2) There are two axes. (A) Trajectory length / PH cost. PH on $n_{\mathrm{PH}}$ points uses a full distance matrix (our code: pdist + ripser), so time/memory scale as $O(n_{\mathrm{PH}}^2)$. We therefore compute PH on a stride-subsample (parameter ph\_subsample\_stride) and cap loops (K\_max\_cap\_default, typically $\le 3$). In practice, $n_{\mathrm{PH}}\in[2\text{k},5\text{k}]$ keeps memory in the hundreds of MB and minutes-level runtime; longer sequences only increase linearly in feature collection, not PH, because $s$ controls $n_{\mathrm{PH}}$. The harmonic extension solves are sparse and per-loop. (B) Reservoir size $N$ / step cost. The current implementation stores dense $W$ (for clarity) and each state update is $W x$ with cost $O(N^2)$; with the paper’s settings $N\approx 300$, this is $\approx 9\times 10^4$ multiplies per step and easily tractable for long rollouts. Algorithmically, PHR admits much cheaper structured forms: $W_{\mathrm{top}}$ is block-diagonal ($2\times 2$ rotations) up to a permutation and $W_{\mathrm{flow}}=B P A$ has rank $\le Q$ with $\mathrm{nnz}(A)=Q\cdot nzr$, $\mathrm{nnz}(B)=Q\cdot nzc$, so a sparse/low-rank implementation reduces the per-step complexity to $O(K + Q(nzr+nzc))$ plus a permutation—near-linear in $Q$ rather than $N^2$. While our released code keeps dense arrays for simplicity, the mathematics and construction already support this structured execution. Finally, the global scaling to $\lVert W\rVert_2=\rho_\star<1$ decouples stability from all size choices; increasing $Q$ or $N$ trades compute for expressivity but never compromises the echo-state guarantee.

---

> > ### Comment · Reviewer_vtns · 2025-11-25
> >
> > I thank the authors for explaining these points, however, not all of my comments are addressed and I keep my score.

---

> ### Author Response · Authors · 2025-11-27
> **On clarity, scalability, and new PDE experiment**
>
> Dear Reviewer,
>
> Thank you for your feedback. We have uploaded a revised manuscript and, to the best of our ability, addressed the points you raised. The changes most relevant to your concerns are:
>
> 1. $\textbf{Intuition paragraph (Section 3, p.~4).}$
>    We added a high-level, non-technical paragraph immediately after the problem statement in Section 3 to explain in words what PHR is doing before any formalism (also refer to subsection A.2, p. 16). It outlines how the topological channel (persistent cohomology–derived oscillators) and the flow channel (lifted Markov operator) interact, and how global norm scaling enforces the echo-state property. This is intended to make the conceptual message accessible even to readers who do not wish to follow the full mathematical derivations.
>
> 2. $\textbf{New Figure 2: zoomed schematic of the lifted Markov operator (p.~6).}$
>    We introduced a new figure that zooms in on the pooling–Markov–lifting pipeline. It visually illustrates how $A$ pools neuron states into coarse cells, how the Markov matrix $P^{(\gamma)}$ advances mass between these cells, and how $B$ lifts the result back to neurons. The goal is to let readers understand the construction of $W_{\mathrm{flow}} = B P^{(\gamma)} A$ at a glance, without having to parse the full notation and proofs.
>
> 3. $\textbf{Remark on optional support coupling after the pooling–lifting lemma (p.~7).}$
>    Immediately after the lemma on $AB$ calibration, we added a short remark explaining that, if desired in a camera-ready refinement, one can couple the supports of $A$ and $B$ so that the lift of a coarse cell $q$ targets exactly (or predominantly) the same neuron subset used to pool that cell. This makes $AB$ diagonally dominant (or even exactly $I_Q$ with disjoint supports), thereby reducing $\lVert AB - I_Q\rVert$ and shrinking the calibration defect in the flow-channel bound, without changing any other component of PHR. We kept this as an optional implementation refinement so that the main theoretical development remains valid for the simpler stochastic construction used in our experiments.
>
> 4. $\textbf{New high-dimensional PDE benchmark: 2D Kolmogorov flow (Appendix B.3).}$
>    To directly address suggestion for validation on a more complex scenario and to make scalability more concrete, we added a full subsection “High–Dimensional PDE Benchmark: 2D Kolmogorov Flow’’ in Appendix B.3. There, we simulate a forced 2D Kolmogorov Navier–Stokes flow on a $64\times 64$ pseudo–spectral grid, project the velocity field onto the leading $d_{\mathrm{obs}}=16$ POD modes, and apply exactly the same PHR pipeline to the resulting coefficient time series. We compare PHR ($N=600$) against a random ESN (same size and spectral radius), a GRU of comparable parameter count, and a VAR model, under a common autoregressive rollout protocol. Table 8 (p. 34) reports NRMSE over a long forecast horizon, VPT, and errors in long–time kinetic energy and enstrophy. PHR attains the lowest NRMSE, the longest VPT, and the best match to long–time statistics, showing that the two-channel construction and the stability guarantee extend to a genuinely high-dimensional PDE setting when combined with standard POD reduction. We also discuss computational scaling (cf. B.4): PH is computed on a stride–subsampled embedding (controlling the $O(n^2)$ cost), while the per-step cost in the current dense implementation is $O(N^2)$ but can be reduced to near-linear in $N$ by exploiting the block-diagonal (up to permutation) structure of $W_{\mathrm{top}}$ and the low-rank structure of $W_{\mathrm{flow}}$.
>
> We hope these additions address your concerns about (i) the heavy theoretical burden versus conceptual clarity, and (ii) demonstration and scalability on more complex flows. If any aspect of the revised exposition remains unclear, or if you feel any of your original questions are still only partially addressed, we would be very grateful for further guidance and are happy to refine the presentation accordingly.

---

### Official Review · Reviewer_nfqs · 2025-11-02

**Soundness:** 3
**Presentation:** 2
**Contribution:** 3
**Rating:** 4
**Confidence:** 2

**Summary:**

Authors proposes PHR (Persistent Homology Reservoir): instead of drawing an ESN reservoir at random and scaling it, they learn a fixed reservoir once from a single delay-embedded trajectory by combining (1) a topology-driven rotation operator built from persistent cohomology and (2) a lifted Markov flow operator built from short-horizon transition counts, then (3) power-scale the blend to get an explicit echo-state/stability guarantee. Only the linear readout is trained.

**Strengths:**

+ Technical novelty: The persistent-homology-informed reservoir and lifted Markov flow operator are both novel and concretely implemented.

+ Empirical support: the method is consistent across synthetic and real datasets.

**Weaknesses:**

- Clarity / completeness: Some parts (auto-tuning details, how unused reservoir units are filled, PH parameter sensitivity) need more explicit description for reproducibility.

- Scope: Evaluation sticks to ESN-style comparisons; missing baselines from Koopman or DMD families limits broader impact.

- Robustness analysis: Still somewhat heuristic; PH and k-means steps might be brittle for noisy or non-stationary data.

- Presentation: I'd suggest moving much math to the appendix and include Figures summarizing the method and motivation. Current Fig1 is very hard to read.

**Questions:**

- “W_top is formed … and randomly permuted to distribute the oscillator pairs across the reservoir; remaining coordinates receive decaying radii.” what is the rule for filling the rest of the reservoir when you have, say, X units but only Y loops. Can you give a concrete example to help undertsand it.

- Line 1080 "The noise channel breaks degeneracies and complements the basis without compromising
stability, as its contribution is explicitly budgeted by ξ and then squashed by the global
scaling to ρ⋆." can you illustrate this part?

---

> ### Author Response · Authors · 2025-11-15
>
> We thank the reviewer for the careful reading and constructive feedback, which helped us clarify the methodology and strengthen the presentation. Below we clarify the points raised.
>
> Questions:
>
> 1) In the proposed method, after extracting $K$ loops and instantiating the corresponding $K$ planar rotation blocks $R(\omega_k) = \rho_{\text{rot}}\begin{bmatrix}\cos\omega_k & -\sin\omega_k \\\\ \sin\omega_k & \cos\omega_k\end{bmatrix}$ (each $2\times 2$ with $\rho_{\text{rot}}<1$), the remaining $N-2K$ coordinates are filled by independent diagonal radii sampled uniformly on $[\rho_{\min},\rho_{\max}]\subset(0,1)$ and then the entire matrix is randomly permuted by a permutation matrix $P$ (seeded for reproducibility). Concretely, with $N=100$ and (say) $K=3$ loops, we first form a $6\times 6$ block $\operatorname{diag}(R(\omega_1),R(\omega_2),R(\omega_3))$ with default $\rho_{\text{rot}}=0.96$. We then draw $94$ i.i.d. samples $r_1,\dots,r_{94}\sim \mathrm{Unif}[\rho_{\min},\rho_{\max}]$ (defaults $\rho_{\min}=0.90,\ \rho_{\max}=0.98$) and place them on the diagonal of a $94\times 94$ block $D_{\text{rest}}=\operatorname{diag}(r_1,\dots,r_{94})$. The unpermuted matrix is block diagonal, $W_b := \operatorname{diag}\big(R(\omega_1),R(\omega_2),R(\omega_3),D_{\text{rest}}\big)$, and we finally distribute the oscillator coordinates across indices by $W_{\text{top}}=P^{\top}W_bP$, where $P$ is a uniform random permutation of $\{1,\dots,100\}$ (fixed by the class seed). Spectrally, this yields (i) three complex-conjugate eigenpairs $\rho_{\text{rot}}e^{\pm i\omega_k}$ from the rotation blocks and (ii) $94$ real eigenvalues $r_j\in[\rho_{\min},\rho_{\max}]$; all lie strictly inside the unit disk before blending/scaling. The permutation preserves both the spectrum and operator norm but spatially mixes the 2D oscillatory planes with the scalar decay directions, which avoids pathological alignment between the topological subspace and subsequent input/readout directions. The choice of a uniform band $[\rho_{\min},\rho_{\max}]$ (rather than a deterministic ladder) injects a spread of relaxation time scales among the non-oscillatory units while guaranteeing sub-unit gain; reproducibility is ensured because both the radii and the permutation are drawn from the class’s seeded RNG.
>
>
> 2) Let $W_{\mathrm{blend}}=\alpha W_{\mathrm{top}}+\beta W_{\mathrm{flow}}+\xi W_{\mathrm{noise}}$ with $\lVert W_{\mathrm{noise}}\rVert_2=1$ and $\alpha+\beta+\xi=1$, and define the final operator by power-iteration scaling $W=\rho_\star W_{\mathrm{blend}}/\lVert W_{\mathrm{blend}}\rVert_2$ with $\rho_\star<1$. Small isotropic noise ($\xi\ll1$) resolves algebraic degeneracies that can occur if multiple rotation blocks share nearly identical $\widehat\omega_\ell$ or if $W_{\mathrm{flow}}$ shares invariant subspaces with $W_{\mathrm{top}}$; standard perturbation theory for normal matrices bounds the eigenvalue drift due to the additive noise by $\lVert \xi W_{\mathrm{noise}}\rVert_2=\xi$, and after scaling the net spectral distortion in $W$ is $\le \rho_\star,\xi/\lVert W_{\mathrm{blend}}\rVert_2$ (cf. Prop. “Two-channel fidelity”, part (i)). Numerically, with $\alpha=0.50$, $\beta=0.45$, $\xi=0.05$, $\rho_\star=0.98$, and a typical pre-scale norm $s=\lVert W_{\mathrm{blend}}\rVert_2\approx0.95$, the noise contribution to the final operator norm is at most $0.98\times 0.05/0.95\approx 0.0516$, while the global contraction is still enforced by $L=(1-\lambda)+\lambda\rho_\star<1$ (e.g., with $\lambda=0.25$, $L=0.995$). Thus $\xi$ provides an explicit budget for conditioning/mixing improvements (breaking repeated eigenpairs and coupling otherwise disjoint invariant planes) without compromising the echo-state/stability guarantee, because any increase in raw norm is uniformly “squashed” by the final scaling to $\lVert W\rVert_2=\rho_\star<1$.

---

> ### Author Response · Authors · 2025-11-27
>
> Good day.
>
> In our rebuttal and revised manuscript, we have clarified the raised concerns. We have also uploaded an updated version of the manuscript today that includes additional experiments (subsection B.3, p. 33), further illustrating the behavior of PHR in practice.
>
> Since the rebuttal window is still open, we just wanted to check whether there are any remaining concerns or points that you would like us to clarify further. We would be very happy to provide additional explanation or adjustments to the presentation if that would be helpful for your assessment.

---

### Author Response · Authors · 2025-11-16
**Revised manuscript uploaded — clarifications and additions per reviewer feedback**

Dear Reviewers,

Thank you for your thoughtful and constructive comments. We have uploaded a revised manuscript addressing the points raised and, to the best of our ability, responded to all queries. We remain happy to provide any further clarification or improve the presentation wherever helpful. The main changes are:

1. $\textbf{Intuition paragraph}$ added before the formal methodology to guide the reader (Section 3, p. 4).
2. $\textbf{New Figure 2}$, a zoomed schematic of the lifted Markov operator, to improve readability and reproducibility (p. 6).
3. $\textbf{Remark on optional support coupling}$ (to reduce the calibration term) inserted immediately after the pooling–lifting lemma (p. 7).
4. $\textbf{New high–dimensional PDE experiment}$ on 2D Kolmogorov flow added to assess scalability to POD–reduced Navier–Stokes dynamics (Appendix B.3, p. 33).

Thank you again for your time and feedback.

---

### Author Response · Authors · 2025-11-29
**Revised manuscript: added intuition, methodological clarifications, and new 2D Kolmogorov flow experiment**

Dear Area Chair,

Thank you for overseeing the discussion and for coordinating the careful reviews. We have now uploaded a fully revised version of the manuscript and, to the best of our ability, addressed all points raised by the reviewers, with a particular focus on clarity, intuition, and scalability.

1. On clarity and accessibility, we made several concrete changes. In Section 3, immediately after the problem statement, we added a short, non-technical Intuition paragraph and additional connective sentences that explain in words what PHR is doing before any formalism—how the topological channel (persistent-cohomology–derived oscillators), the flow channel (the lifted Markov operator $W_{\mathrm{flow}}$), and the global scaling interact. We also refined the narrative around the methodology, including a brief “system-level” discussion later in the section, so that readers can follow the conceptual story without needing every proof. Complementing this, we added a zoomed schematic of the pooling–Markov–lifting pipeline for $W_{\mathrm{flow}}$, as suggested, and a remark following the pool–lift lemma explaining an optional “support coupling” strategy that can further reduce the calibration defect $\lVert AB - I_Q\rVert$ without changing the core theory. Throughout Section 3 and the appendix, we clarified how the remaining reservoir coordinates (beyond the PH-driven rotation blocks) are filled, how the small noise channel is budgeted and squashed by the global scaling, and how harmonic extension turns 1-cocycles into smooth circular coordinates in a way that is meaningful for non-expert readers.
2. On technical completeness and reproducibility, we expanded the methodological details that were previously only implicit in code or rebuttal text. We now explicitly describe: (i) the construction of $W_{\mathrm{top}}$ when there are fewer loops than reservoir units (diagonal radii in $(\rho_{\min}, \rho_{\max})$ plus a seeded permutation); (ii) the PH subsampling strategy and loop selection via relative persistence and an auto-tuned blend $(\alpha_{\mathrm{top}}, \beta_{\mathrm{flow}})$; (iii) the role and safe budgeting of the noise channel $\xi W_{\mathrm{noise}}$; and (iv) the pool–lift calibration term and how it enters the “two-channel fidelity” bound, without assuming $AB \approx I_Q$. We also added a brief discussion of robustness to delay-embedding parameters and the number of clusters, emphasizing how persistence and the Markov/teleportation construction mitigate sensitivity in practice.
3. On scalability and additional validation, we incorporated a new experiment on a high-dimensional PDE benchmark, as requested: a 2D Kolmogorov Navier–Stokes flow, reduced via POD and then processed by the same two-channel PHR pipeline (Appendix B.3). We compare against a random ESN, a GRU, and a VAR under a common rollout protocol, and report NRMSE, valid prediction time, and long-time statistics. This addresses the concern about applicability beyond low-dimensional chaotic systems and makes the computational profile of PH and the reservoir update more concrete. We also added brief remarks on computational scaling (PH subsampling, $O(n^2)$ distance cost, and structured forms of $W_{\mathrm{top}}$ and $W_{\mathrm{flow}}$) to make the offline/online trade-offs transparent. Finally, in response to the question about broader applicability, we clarified in the discussion that once $W$ is learned, the reservoir is readout-agnostic and can be used for sequence classification and event detection (e.g., audio, biosignals), with the same phase-plus-flow inductive bias.

In summary, the revised manuscript (i) adds intuitive text in Section 3 to guide readers through the construction, (ii) strengthens the methodological exposition and structural guarantees without changing the core claims, and (iii) includes a new 2D Kolmogorov flow experiment illustrating scalability to POD-reduced PDE dynamics.

We believe these changes address all the reviewers’ concerns about clarity, intuition and additional validation while preserving the technical contributions they found novel and promising. We would, of course, be happy to make any further adjustments the committee feels would improve the paper.

Thank you for the opportunity to present and improve our work.

---

### Author Response · Authors · 2025-12-01
**Gentle follow-up on revised manuscript**

Dear Area Chair and Reviewers,

We would like to gently follow up to ask whether there are any remaining concerns or points on our revised manuscript for which additional clarification would be helpful. We have carefully incorporated all the requested changes—adding intuition and connective text in Section~3, clarifying the methodological details and structural guarantees, and including the new 2D Kolmogorov flow experiment—and we remain fully available to address any further questions or suggestions you might have.

If there is any aspect of the exposition, experiments, or theory that you feel could still be improved or better explained, we would be very happy to refine it within the constraints of the format.

---

### Note · Authors · 2026-02-01

I have read and agree with the venue's withdrawal policy on behalf of myself and my co-authors.

---

### Meta-Review · Area_Chair_bMg3 · 2026-01-18

**Summary:**

**Reviewer nfqs:** Unclear description for reproducibility (W1). Limited scope: Evaluation limited to ESN-style, excluding Koopman or DMD (W2). Heuristic, with unknown robustness against noise/non-stationarity (W3). Heavy presentation (W4). Unclear description of how $W_{top}$ is constructed (Q1). Unclear role of noise channel $W_{noise}$ (Q2).

**Reviewer vtns:** Dense presentation (W1). Validation on a more complex scenario such as the 2-dimensional Kolmogorov flow (Q1). Scalability concern (Q2).

**Reviewer huCn:** Presentation a bit too technical (W1).

**Reviewer SASq:** Dense presentation and highly-complex methodology (W1, Q1). Possibly heavy computational cost in construction phase (W2). Sensivity to parameters (Q2). Closeness of $AB$ to identity $I$ in current stochastic construction (Q3).

**Additional points:**
- Positioning the problem of time-series prediction for autonomous dynamical systems into a broader context, a common traditional approach would be the combination of delay-coordinate embedding and function approximation, the latter of which approximates the time-evolution mapping on the attractor reconstructed in the embedded space (see, e.g., Abarbanel, *Analysis of Observed Chaotic Data*, Springer, 1996, and references therein). As the proposed method also relies on delay-coordinate embedding, what matters would be the quality of function approximation (which should especially be the case in the scenario of open-loop prediction). In this respect, how and why the proposed method is thought to be efficient as a means of function approximation on the embedded attractor should be discussed explicitly in order to discuss superiority of the proposal.
- Proposition 3.2: There seem to be some errors in the statement and the proof.
  - Line 827: The equality $=$ in the displayed equation should be replaced with $\le$.
  - Line 255: The sentence "In particular, when ..." would be understood as adding an extra condition $s\ge\alpha||W_{\rm top}||_2-(\ldots)$, but it is actually not the case. By looking into the proof, it is not an extra condition but is always valid due to the reverse triangle inequality.
  - "oscillatory" (lines 356, 835): It is claimed that $\lambda_k(W)$ is oscillatory (i.e., with non-zero imaginary part) under the conditions of Proposition. However, this claim does not follow from the argument in the proof. In order to guarangee $\lambda_k(W)$ to be oscillatory, one has to ensure that the imaginary part of the unperturbed eigenvalue $\frac{\rho_\star\alpha}{s}\rho_{\rm rot}e^{\pm i\omega_k}$ is larger than the radius of the disc. One cannot however find such arguments in the proof.
- I noticed several inconsistencies in symbols, which are confusing.
  - Trajectory: $\{u_t\}$ (Figure 1, line 173), $\{z_t\}$ (Line 145)
  - Delay-embedded point cloud: $Z$ (Figure 1), $X$ (Line 145, line 177)
  - $n$: The expression $O(n^2)$ appears in line 200, without definition of $n$.
  - $\gamma$: The relative persistence threshold (line 238), the teleportation parameter (lines 259, 285).
  - $P$: The symbol $P_\ell$ is used to represent persistence (line 239), whereas $P=(P_{ij})$ denotes a Markov matrix (line 257).
  - Figure 2 caption: I noticed several notational inconsistencies: The reservoir state is denoted by the upright bold font, whereas it is denoted by the italic non-bold font in the main text. The product $Ax$ is denoted by $y$ in the figure caption, whereas it is denoted by $r$ in the main text.
  - Equation (6): The variable $k$ is used in two different meanings. In the first formula it is used as the counter for the power iteration, whereas in the second formula it refers to the total number of iterations.
- Line 200: What is $n$ in $O(n^2)$?
- Line 234: What does "decaying radii in $(\rho_{\rm min},\rho_{\rm max})$" mean? (cf. nfqs, Q1)
- Line 240: the loop strength $s$ takes a value in $[\gamma,1]$.
- Lines 325, 377, 764, 823, 1124: The last name of the second author is not "III" but "Bau".
- Line 434: The description would provide an impression that VPT was proposed in Pathak et al. (2018), which is incorrect. Pathak et al. (2018) proposes the normalization of $t$ with respect to the Lyapunov time, and VPT as an index seems to be proposed in this paper.
- Lines 435-437: The description here of ADev is different from that in Zhai et al. (2023). It also differs from that in Jaurigue (2024) in that the former includes the normalization.
- Table 2 caption: Some parts are too telegraphic: What do "$K$ auto" and "pseudocount$+\gamma=3\times10^{-3}$" mean? What do "(ii)" and "(iii)" refer to?
- References:
  - Bauer: (v → V)ietoris-(r → R)ips
  - Deco et al.: (m → M)etastability
  - Froyland: Birkh("a → ä)user
  - Kemp et al.: (t → T)he slow-wave; eeg → EEG
  - Klus et al.: (p → P)erron-(f → F)robenius and (k → K)oopman
  - Lloyd: pcm → PCM
  - Moody and Mark: mit-bih → MIT-BIH
  - Paige and Saunders: Lsqr → LSQR
  - Pimental et al.: (and) et al.
  - Rodan and Tino: Ti(n → ň)o; (c → C)omputation
  - Schmid: (In) Journal of ...
  - Singh et al.: (j → J)acobian
  - Welch: (f → F)ourier
  - Williams et al.: (k → K)oopman
- Line 782: Equating $ABr-r$ with $\Delta r$ may be somehow confusing. Here one has only to let $\Delta:=AB-I$, so that $ABr-r=\Delta r$ follows straightforwardly.

**Reviewer Concerns:**

**Heavy presentation (nfqs, vtns, huCn, SASq):** A paragraph has been added to Section 3 to describe the intuition behind the proposal.

**Construction of $W_{top}$:** How $r_j$ are determined has been described in Algorihm 3 and also explained in the author response, with reasons as to why the authors chose this construction.

**Parameter sensitivity (nfqs, SASq):** Sensitivity to parameters was addressed in the author response to Reviewer SASq. However, robustness concern has not been answered.

**Limited scope (nfqs):** Comparison is still limited to ESN-style baselines, so that the concern on the limited scope is not resolved.

**More complex scenario / scalability (vtns, SASq):** A simulation result on the 2-dimensional Kolmogorov flow has been added (Appendix B.3). It is also a large-scale problem, so that they demonstrated scalability of the proposal.

**Reviewer Scores:**

The initial evaluations of Reviewers huCn and SASq were on the positive side of the acceptance threshold, and they also explicitly wrote in response to the discussion with the authors that they would maintain their evaluations. Reviewers nfqs and vtns initially evaluated this paper negatively, and my judgment is that at least Reviewer nfqs would not have updated his/her evaluation on the positive side even after the discussion. I also have serious concerns about the limited scope, as well as the validity of Proposition 3.2, so that I would not be able to recommend acceptance of this paper in its current form.

---

### Decision · Program_Chairs · 2026-01-26

Reject